# Revisiting Evaluation Metrics for Semantic Segmentation: Optimization and Evaluation of Fine-grained Intersection over Union

**Zifu Wang**[1]* **Maxim Berman**[2] **Amal Rannen-Triki**[3] **Philip H.S. Torr**[4] **Devis Tuia**[5]

**Tinne Tuytelaars**[1] **Luc Van Gool**[1,6,7] **Jiaqian Yu**[8] **Matthew B. Blaschko**[1]

[1] ESAT-PSI, KU Leuven, Leuven, Belgium
[2] Google, Zürich, Switzerland
[3] Google DeepMind, London, United Kingdom
[4] University of Oxford, Oxford, United Kingdom
[5] EPFL, Lausanne, Switzerland
[6] ETH Zürich, Zürich, Switzerland
[7] INSAIT, Sofia, Bulgaria
[8] Samsung Research, Beijing, China

## Abstract

Semantic segmentation datasets often exhibit two types of imbalance: *class imbalance*, where some classes appear more frequently than others and *size imbalance*, where some objects occupy more pixels than others. This causes traditional evaluation metrics to be biased towards *majority classes* (e.g. overall pixel-wise accuracy) and *large objects* (e.g. mean pixel-wise accuracy and per-dataset mean intersection over union). To address these shortcomings, we propose the use of fine-grained mIoUs along with corresponding worst-case metrics, thereby offering a more holistic evaluation of segmentation techniques. These fine-grained metrics offer less bias towards large objects, richer statistical information, and valuable insights into model and dataset auditing. Furthermore, we undertake an extensive benchmark study, where we train and evaluate 15 modern neural networks with the proposed metrics on 12 diverse natural and aerial segmentation datasets. Our benchmark study highlights the necessity of not basing evaluations on a single metric and confirms that fine-grained mIoUs reduce the bias towards large objects. Moreover, we identify the crucial role played by architecture designs and loss functions, which lead to best practices in optimizing fine-grained metrics. The code is available at https://github.com/zifuwanggg/JDTLosses.

## 1 Introduction

Every metric reflects a certain property of the result and choosing the right one is important to emphasize those that we care about. However, this choice may not be easy, as many metrics come with certain biases [34]. For semantic segmentation, the overall pixel-wise accuracy (Acc) is believed unsuitable due to its bias towards majority classes, especially given that semantic segmentation datasets typically have long-tailed class distributions [18]. The mean pixel-wise accuracy (mAcc) is introduced to counter this issue by averaging pixel-wise accuracy across all classes, and it eventually became the official evaluation metric in PASCAL VOC 2007 [16], which was one of the earliest challenges that showcased a segmentation leaderboard. However, mAcc does not account for false positives, leading to over-segmentation of the results. Consequently, in PASCAL VOC 2008 [17],

---

*Correspondence to: zifu.wang@kuleuven.be

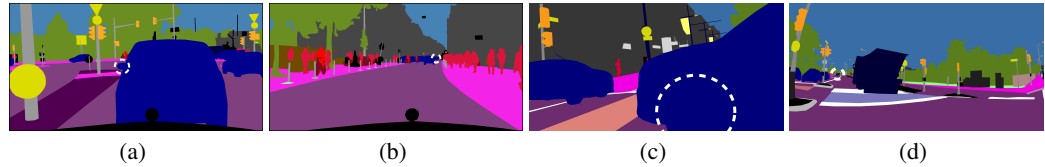



(a)    (b)    (c)    (d)



Figure 1: An illustration of size imbalance. The class `Car` is depicted in blue and the regions are highlighted within white dashed circles. (a) A car whose side mirror takes up around 2,000 pixels (from Cityscapes). (b) A car that takes up around 1,000 pixels (from Cityscapes). (c) A car whose front wheel takes up around 600,000 pixels (from Mapillary Vistas). (d) A car that takes up around 1,000 pixels (from Mapillary Vistas).

the official metric was updated to the per-dataset mean intersection over union ($\text{mIoU}^{\text{D}}$). Since then, PASCAL VOC [18] has consistently used $\text{mIoU}^{\text{D}}$, and this metric has been adopted by a multitude of segmentation datasets [38, 9, 39, 59, 5, 11, 12, 15, 43].

In $\text{mIoU}^{\text{D}}$, true positives, false positives and false negatives are accumulated across all pixels over the whole dataset and therefore this metric suffers several notable shortcomings [10, 9]:

- It is biased towards large objects, which can pose a significant issue given that most semantic segmentation datasets inherently have a considerable size imbalance (see Figure 1). This bias is particularly concerning in safety-aware applications with critical small objects such as autonomous driving [4, 9, 39, 11, 43] and medical imaging [37, 22, 49, 3].
- It fails to capture valuable statistical information about the performance of methods on individual images or instances, which impedes a comprehensive comparison.
- It is challenging to link the results from user studies to those obtained with a per-dataset metric, since humans generally assess the quality of segmentation outputs at the image level.

In this paper, we advocate fine-grained mIoUs for semantic segmentation. In line with previous studies [10, 55, 28, 31], we calculate image-level metrics. When instance-level labels are available [9, 39, 59, 5], we propose to compute approximated instance-level scores. These fine-grained metrics diminish the bias toward large objects. Besides, they yield a wealth of statistical information, which not only allows for a more robust and comprehensive comparison, but also leads to the design of corresponding worst-case metrics, further proving their importance in safety-critical applications. Furthermore, we show that these fine-grained metrics facilitate detailed model and dataset auditing.

Despite the clear advantages of fine-grained mIoUs over $\text{mIoU}^{\text{D}}$, these metrics are seldom considered in recent studies within the segmentation community. We bridge this gap with a large-scale benchmark, where we train and evaluate 15 modern neural networks on 12 natural and aerial segmentation datasets, spanning a variety of scenarios including (i) street scenes [4, 9, 39, 11, 43], (ii) "thing" and "stuff" [18, 38, 59, 5], (iii) aerial scenes [12]. Our benchmark study emphasizes the peril of depending solely on a single metric and verifies that fine-grained mIoUs mitigate the bias towards large objects. Moreover, we conduct an in-depth analysis of neural network architectures and loss functions, leading to best practices for optimizing these metrics.

In sum, we believe no evaluation metric is perfect. Thus, we advocate a comprehensive evaluation of segmentation methods and encourage practitioners to report fine-grained metrics and corresponding worst-case metrics, as a complement to $\text{mIoU}^{\text{D}}$. We believe this is particularly crucial in safety-critical applications, where the bias towards large objects can precipitate catastrophic outcomes.

## 2    Fine-grained mIoUs

In this section, we review the traditional per-dataset mean intersection over union, followed by our proposed fine-grained image-level and instance-level metrics.

**$\text{mIoU}^{\text{D}}$.** Intersection over union (IoU) is defined as

$$\text{IoU} = \frac{\text{TP}}{\text{TP} + \text{FP} + \text{FN}}, \tag{1}$$

where TP, FP and FN represent true positives, false positives and false negatives, respectively. Per-dataset IoU computes these numbers by accumulating all pixels over the whole dataset. In particular, for a class $c$, it is defined as

$$\text{IoU}_c^{\text{D}} = \frac{\sum_{i=1}^{I} \text{TP}_{i,c}}{\sum_{i=1}^{I} (\text{TP}_{i,c} + \text{FP}_{i,c} + \text{FN}_{i,c})}, \tag{2}$$

where $I$ is the number of images in the dataset; $\text{TP}_{i,c}$, $\text{FP}_{i,c}$ and $\text{FN}_{i,c}$ are the number of true-positive, false-positive and false-negative pixels for class $c$ in image $i$, respectively. The per-dataset mean IoU is then calculated as

$$\text{mIoU}^{\text{D}} = \frac{1}{C} \sum_{c=1}^{C} \text{IoU}_c^{\text{D}}, \tag{3}$$

where $C$ denotes the number of classes for the dataset.

Compared to pixel-wise accuracy (Acc) and mean pixel-wise accuracy (mAcc), $\text{mIoU}^{\text{D}}$ aims to mitigate class imbalance and to take FP into account. It is widely used in the evaluation of semantic segmentation models, but it has several shortcomings. Most notably, it is biased towards large objects in the dataset, due to dataset-level accumulations of TP, FP and FN. To address this issue, we can resort to fine-grained mIoUs at per-image ($\text{mIoU}^{\text{I}}$, $\text{mIoU}^{\text{C}}$) and per-instance ($\text{mIoU}^{\text{K}}$) levels, as detailed below.

**$\text{mIoU}^{\text{I}}$.** Following [10, 55, 28, 31], we compute per-image scores. In particular, for each image $i$ and class $c$, a per-image-per-class score is calculated as

$$\text{IoU}_{i,c} = \frac{\text{TP}_{i,c}}{\text{TP}_{i,c} + \text{FP}_{i,c} + \text{FN}_{i,c}}. \tag{4}$$

Then the per-image IoU for image $i$ is defined as

$$\text{IoU}_i^{\text{I}} = \frac{\sum_{c=1}^{C} \mathbb{1}\{\text{IoU}_{i,c} \neq \texttt{NULL}\}\text{IoU}_{i,c}}{\sum_{c=1}^{C} \mathbb{1}\{\text{IoU}_{i,c} \neq \texttt{NULL}\}} \tag{5}$$

where we only sum over $\text{IoU}_{i,c}$ that is not $\texttt{NULL}$ (discuss below). Averaging these per-image scores yields

$$\text{mIoU}^{\text{I}} = \frac{1}{I} \sum_{i=1}^{I} \text{IoU}_i^{\text{I}}. \tag{6}$$

Using image-level metrics presents a challenge: not all classes appear in every image, leading to $\texttt{NULL}$ values and potential ambiguities. In multi-class segmentation, we denote the per-image-per-class value as $\texttt{NULL}$ if the class is missing from the ground truth, regardless of it appearing in the prediction or not. In contrast, for binary segmentation, the value is set to 0 instead of $\texttt{NULL}$ when the prediction includes the foreground class, but the ground truth does not. More detail of the definition is in Appendix D.

**$\text{mIoU}^{\text{C}}$.** As the left panel of Figure 2 shows, due to the presence of $\texttt{NULL}$ values, averaging these per-image-per-class scores by class first and then by image ($\text{mIoU}^{\text{I}}$) is different from averaging them by image first and then by class. A drawback of $\text{mIoU}^{\text{I}}$ is that it is biased towards classes that appear in more images[2]. To address this bias, we first average these per-image-per-class scores by image and define the per-image IoU for class $c$ as

$$\text{IoU}_c^{\text{C}} = \frac{\sum_{i=1}^{I} \mathbb{1}\{\text{IoU}_{i,c} \neq \texttt{NULL}\}\text{IoU}_{i,c}}{\sum_{i=1}^{I} \mathbb{1}\{\text{IoU}_{i,c} \neq \texttt{NULL}\}}. \tag{7}$$

Similar to $\text{mIoU}^{\text{D}}$, we then average these per-class values to obtain

$$\text{mIoU}^{\text{C}} = \frac{1}{C} \sum_{c=1}^{C} \text{IoU}_c^{\text{C}}. \tag{8}$$

---

[2]This is different from Acc that is biased towards classes that take up more pixels.

| | $I_1$ | $I_2$ | $I_3$ | $I_4$ | $I_5$ |
|---|---|---|---|---|---|
| $C_1$ | 0.85 | 0.94 | 0.56 | 0.73 | 0.68 |
| $C_2$ | 0.42 | NULL | 0.59 | 0.88 | NULL |
| $C_3$ | NULL | 0.79 | NULL | NULL | NULL |

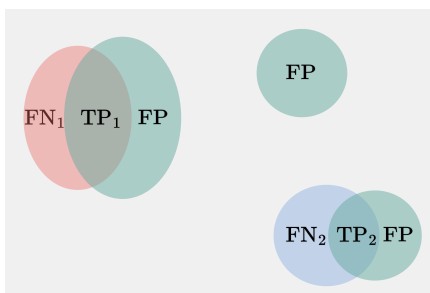

Figure 2: **Left:** mIoU$^I$ first averages per-image-per-class scores by class and then by image. mIoU$^C$ first averages them by image and then by class. mIoU$^I$ is biased towards classes that appear in more images, e.g. $C_1$. **Right:** Given two ground-truth instances 1 (red) and 2 (blue), and the prediction (green), we can compute TP$_1$, TP$_2$, FN$_1$, FN$_2$. We propose to approximate FP$_1$ and FP$_2$ by distributing FP proportional to the size of each instance.

**mIoU$^K$.** IoU is known for its scale-variance. However, this property only makes sense at an instance level. When instance-level annotations are available, for example, for "thing" classes in panoptic segmentation [27] ("thing" classes are with characteristic shapes and identifiable instances such as a truck; see a more detailed discussion of "thing" and "stuff" in Appendix B.1), we can compute a even more fine-grained metric.

Specifically, for an image $i$ and a class $c$, we have a binary mask $H \times W$ from the prediction of a semantic segmentation network and an instance-level annotation $H \times W \times K_{i,c}$ from panoptic segmentation, where $K_{i,c}$ is the number instances of class $c$ that appear in image $i$. The key insight, as illustrated in the right panel of Figure 2, is to realize that TP and FN for each instance can be computed exactly, and we are only left with image-level FP. We propose to approximate instance-level FP by distributing image-level FP proportional to the size of each instance, with other variants discussed in Appendix E. Therefore, for class $c$ in image $i$, we compute the per-instance IoU as

$$\text{IoU}_{i,c,k} = \frac{\text{TP}_{i,c,k}}{\text{TP}_{i,c,k} + \text{FN}_{i,c,k} + \frac{S_{i,c,k}}{\sum_{k=1}^{K_{i,c}} S_{i,c,k}}\text{FP}_{i,c}}, \tag{9}$$

such that FP$_{i,c}$ is the total number of false-positive pixels of class $c$ in image $i$, and $S_{i,c,k} = \text{TP}_{i,c,k} + \text{FN}_{i,c,k}$ is the size of instance $k$. Having these per-instance IoUs, we can again calculate a per-class score as

$$\text{IoU}_c^K = \frac{\sum_{i=1}^{I} \sum_{k=1}^{K_{i,c}} \text{IoU}_{i,c,k}}{\sum_{i=1}^{I} K_{i,c}}, \tag{10}$$

With a slightly abuse of notation, we have

$$\text{mIoU}^K = \frac{1}{C} \sum_{c=1}^{C} \text{IoU}_c^K, \tag{11}$$

such that for classes with instance-level labels, we calculate IoU$_c^K$ according to Eq. (10), and for classes without instance-level labels, such as "stuff" classes (classes that are amorphous and do not have identifiable instances such as sky, see Appendix B.1), we compute it as in Eq. (7).

To conclude, similar to mIoU$^D$, we average the scores on a per-class basis for mIoU$^C$ and mIoU$^K$, therefore mitigating *class imbalance*. Additionally, mIoU$^I$ and mIoU$^C$ reduce *size imbalance* from dataset-level to image-level, and mIoU$^K$ further computes the score at an instance level. On the other hand, scores of mIoU$^I$ are finally averaged on a per-image basis; therefore it will be biased towards classes that appear in more images. However, mIoU$^I$ leads to a single averaged score per image (while mIoU$^C$ has $I$ scores per class), making per-image analysis (e.g. image-level histograms and worst-case images) feasible as we will present in section 5.1.

# 3   Worst-case Metrics

It is essential for safety-critical applications to evaluate a model's worst-case performance. Adopting fine-grained mIoUs allows us to compute worst-case metrics, where only images/instances that a model obtains the lowest scores are considered. In this section, we focus on $\text{mIoU}^C$ as an example. However, the same idea can be applied to both $\text{mIoU}^I$ and $\text{mIoU}^K$.

For each class $c$, we first sort $\text{IoU}_{i,c}$ by images such that $\text{IoU}_{1,c} \leq ... \leq \text{IoU}_{I_c,c}$, where $I_c = \sum_{i=1}^{I} \mathbb{1}\{\text{IoU}_{i,c} \neq \text{NULL}\}$. We then only compute the average of those scores that fall below the $q$-th quantile:

$$\text{IoU}_c^{C^q} = \frac{1}{\max(1, \lfloor I_c \times q\% \rfloor)} \sum_{i=1}^{\max(1, \lfloor I_c \times q\% \rfloor)} \text{IoU}_{i,c}. \tag{12}$$

For the ease of comparison, we want to derive a single metric that considers different $q\%$. Thus, we partition the range into 10 quantile thresholds $\{10, 20, \dots, 90, 100\}$. We then average the scores corresponding to these thresholds:

$$\text{IoU}_c^{C^{\bar{q}}} = \frac{1}{10} \sum_{q \in \{10, \dots, 100\}} \text{IoU}_c^{C^q}. \tag{13}$$

Consequently, the scores with lower values will be given more weights to the final metric. Additionally, we consider $\text{IoU}_c^{C^1}$ and $\text{IoU}_c^{C^5}$ to evaluate a model's performance on extremely hard cases. Having these per-class scores, we can average them to obtain the mean metrics: $\text{mIoU}^{C^{\bar{q}}}$, $\text{mIoU}^{C^5}$ and $\text{mIoU}^{C^1}$.

# 4   Advantages of Fine-grained mIoUs

In summary, we advocate these fine-grained mIoUs as a complement of $\text{mIoU}^D$ because they present several notable benefits:

- **Reduced bias towards large objects.** Unlike $\text{mIoU}^D$, fine-grained metrics compute TP, FP and FN at an image or an instance level, thus decreasing the bias towards large objects. To further illustrate this on real datasets, we can compute the ratio of the size of the largest object to that of the smallest object for each class, evaluated at both dataset ($r_c^D$) and image levels ($r_c^I$):

$$r_c^D = \frac{\max_{i,k} S_{i,c,k}}{\min_{i,k} S_{i,c,k}}, \quad r_c^I = \max_i \frac{\max_k S_{i,c,k}}{\min_k S_{i,c,k}}. \tag{14}$$

  These ratios can serve as an indicator of size imbalance at either the dataset or image level. We average $\log \frac{r_c^D}{r_c^I}$ for "thing" and "stuff" classes separately, and note that $r_c^I = 1$ for all "stuff" classes. In Figure 5 (Appendix B.1), we present the numbers for Cityscapes, Mapillary Vistas, ADE20K, and COCO-Stuff. The size imbalance is considerably reduced at the image level, especially for "stuff" classes. As a result, fine-grained mIoUs can be less biased towards large objects.

- **A wealth of statistical information.** For example, fine-grained metrics enable to compute worst-case metrics at image and instance levels as discussed in the previous section. We can also plot a histogram for $\text{mIoU}^I$, from which we can extract various statistical information and identify worst-case images. These worst-case images can be instrumental in examining the specific scenarios in which a model underperforms, i.e. *model auditing*. Additionally, employing fine-grained scores facilitates statistical significance testing. Altogether, this yields a more robust and comprehensive comparison between different methods.

- **Dataset auditing.** Images that consistently yield a low image-level score across various models can be examined. These low scores could potentially be attributed to mislabeling. Furthermore, the presence of discrepancies between image- and object-level labels can result in an abnormal $\text{mIoU}^K$ values (NaN) because the denominator will become zero. We present the identified mislabels in Appendix B.3.

# 5 Experiments

We provide a large-scale benchmark study, where we train 15 modern neural networks from scratch and evaluate them on 12 datasets that cover various scenes. More details of 15 models and 12 datasets considered in our benchmark study can be found in Appendix A and Appendix B, respectively. The total amount of compute time takes around 1 NVIDIA A100 year.

In our benchmark, we emphasize the importance of training models from scratch (training details are in Appendix C) instead of relying on publicly available pretrained checkpoints. This choice is motivated by several factors:

- **Analysis.** By training models from scratch, we aim to analyze how specific training strategies can lead to high results on fine-grained metrics. This provides insights into the training process and enables a better understanding of the factors affecting performance.

- **Completeness.** While there are pretrained models available for popular datasets like Cityscapes and ADE20K, there is a scarcity of checkpoints for other datasets. Training models from scratch ensures a more comprehensive evaluation across different datasets, providing a more complete perspective on model performance.

- **Fairness.** Publicly available pretrained checkpoints are often trained with different hyperparameters and data preprocessing procedures. Training models from scratch ensures fairness in comparing different architectures and training strategies, as they all start from the same initialization and undergo the same training process.

- **Performance.** We leverage recent training techniques [33, 50], which have been shown to improve results. By training models from scratch with these techniques, we aim to achieve better performance compared to other publicly available checkpoints. We provide a comparison of our results with those of MMSegmentation [8] in Table 1 and Table 2.

- **Statistical significance.** Publicly available pretrained checkpoints often stem from a single run, which may lack statistical significance. By training models from scratch, we can perform multiple runs and obtain statistically significant results. This ensures a more reliable evaluation and avoids potential misinterpretations.

Table 1: Comparing the performance of DeepLabV3+-ResNet101 with MMSegmentation. All results are mIoU$^D$. Red: the best in a column.

| Dataset | Cityscapes | ADE20K | VOC | Context |
|---|---|---|---|---|
| MMSegmentation | 80.97 | 45.47 | 78.62 | 53.20 |
| Ours | 80.91 ± 0.17 | 46.32 ± 0.33 | 81.07 ± 0.28 | 56.49 ± 0.10 |

Table 2: Comparing the performance on ADE20K with MMSegmentation. All results are mIoU$^D$. Red: the best in a column.

| Model | DL3+-R101 | DL3+-MB2 | UPN-R101 | UPN-Conv | Seg-MiTB4 | PSP-R101 |
|---|---|---|---|---|---|---|
| MMSegmentation | 45.47 | 34.02 | 43.82 | 48.71 | 48.46 | 44.39 |
| Ours | 46.32 ± 0.33 | 36.85 ± 0.34 | 45.89 ± 0.11 | 51.08 ± 0.42 | 50.23 ± 0.26 | 45.69 ± 0.33 |

## 5.1 Main Results

Complete results, including tabular entries, image-level histograms and worst-case images are in Appendix G. Besides, we rank 15 models by their averaged ranking across 10 datasets, excluding Nighttime Driving and Dark Zurich. As depicted in Figure 3, we contrast the rank of mIoU$^C$ with (a) mIoU$^D$, (b) mIoU$^I$, (c) mIoU$^{C^{\bar{q}}}$, (d) mIoU$^{C^5}$. Furthermore, we count the number of times each worst-case image appears for each model, and present the count of three most common worst-case images for each dataset in Table 13 (Appendix G). The key findings are summarized below:

**No model achieves the best result across all metrics and datasets.** UPerNet-ConvNeXt usually obtains the highest score for most of the metrics, possibly due to the ImageNet-22K pretraining of ConvNeXt, while other backbones are pretrained on ImageNet-1K. However, UPerNet-MiTB4 and SegFormer-MiTB4 outperform UPerNet-ConvNeXt on many worst-case metrics. Since different metrics focus on a certain property of the result, it is important to have a comprehensive comparison using various metrics.

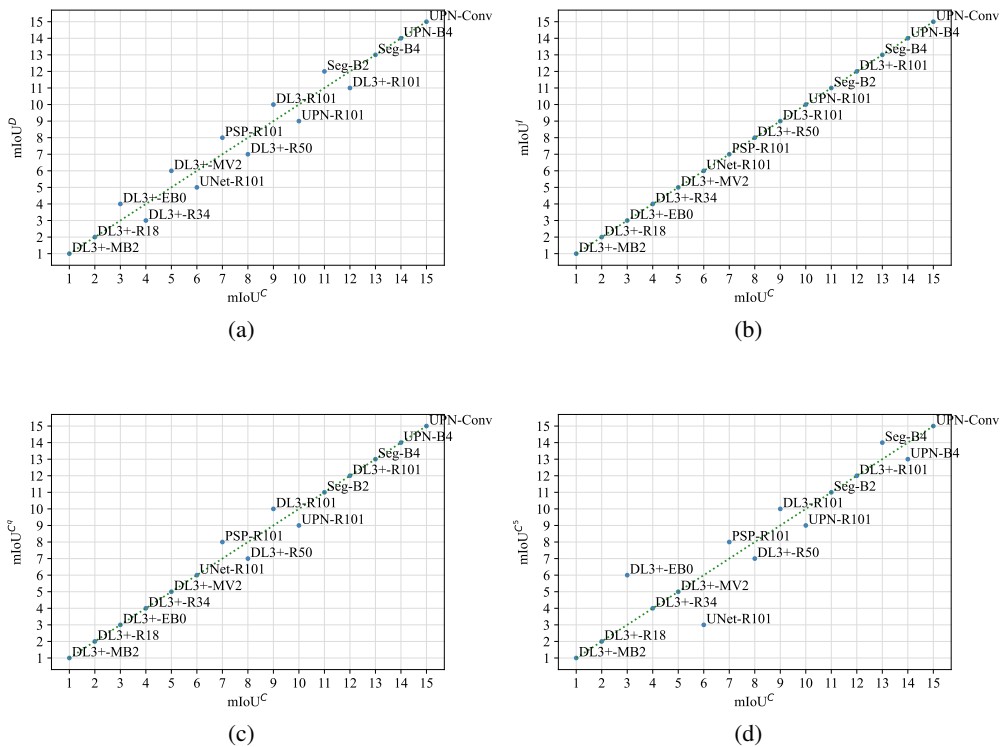

Figure 3: Contrasting the rank of mIoU$^C$ with (a) mIoU$^D$, (b) mIoU$^I$, (c) mIoU$^{C^{\bar{q}}}$, (d) mIoU$^{C^5}$. Models below the green dashed line have a higher rank of mIoU$^C$ than that of the counterpart.

**Models do not overfit to a particular metric or a specific range of image-level values.** As shown in Figure 3, we witness only minor local rank discrepancies when comparing mIoU$^C$ with mIoU$^D$, and notably, no rank differences arise when comparing mIoU$^C$ with mIoU$^I$. In general, we observe a monotonic trend. This proves that models are not tailored to favor a particular metric. The rank differences between mIoU$^C$ and mIoU$^{C^{\bar{q}}}$ are also minimal. This implies that the advancements on mIoU$^C$ are global and models do not overfit to a specific range of image-level values. However, we identify a larger rank drop of UNet-ResNet101 in mIoU$^{C^5}$, indicating that UNet may suffer from more challenging examples.

**The performance on fine-grained mIoUs is relatively low.** mIoU$^D$, mIoU$^C$ and mIoU$^K$ are all averaged on a per-class basis. Their values typically follow the ranking: mIoU$^D \geq$ mIoU$^C \geq$ mIoU$^K$. The more fine-grained a metric is, the more challenging the problem it presents, since it is less merciful to individual mistakes. Nevertheless, we observe that the results for mIoU$^C$ and mIoU$^K$ are usually very similar. This observation indicates that mIoU$^C$ can serve as a reliable proxy for instance-wise metrics when instance-level labels are unavailable, e.g. PASCAL VOC/Context, etc.

**Improvements in fine-grained mIoUs do not always extend to corresponding worst-case metrics.** The worst-case metrics generally fall substantially below their corresponding mean metrics. For example, on PASCAL Context, the value of mIoU$^C$ ranges from 46.00% to 60.17%, yet all models have 0% mIoU$^{C^1}$. The presence of extremely challenging examples poses obstacles to the application of neural networks in safety-critical tasks. It would be an interesting future research to explore ways of enhancing a model's performance on these hard examples.

**Certain images consistently yield low image-level IoU values across various models.** As presented in Table 13, most models agree on worst-case images across multiple datasets. Several challenging factors, such as difficult lighting conditions and the presence of small objects, can contribute to low IoU scores (as seen in Mapillary Vistas, depicted in Figures 19 - 21). Besides, mislabeling in the dataset may also cause models to produce an image-level IoU of 0 (see Appendix B.3).

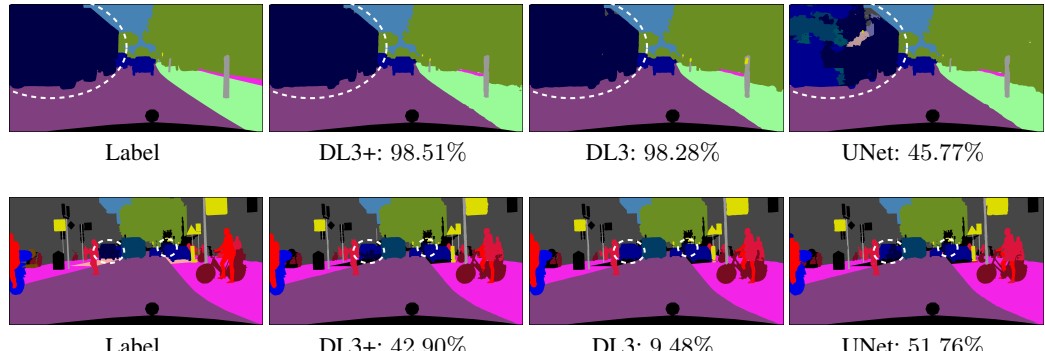

Figure 4: From left to right: ground-truth label, prediction and per-image IoU of class `Truck` with DeepLabV3+, DeepLabV3 and UNet using ResNet101 backbone, respectively. The class `Truck` is represented in dark blue and highlighted within white dashed circles.

Table 3: Comparing DeepLabV3+, DeepLabV3 and UNet on Cityscapes using ResNet-101 backbone. Red: the best in a column. Green: the worst in a column.

| Method | $\text{IoU}^{\text{D}}_{\text{Truck}}$ | $\text{mIoU}^{\text{D}}$ | $\text{IoU}^{\text{C}}_{\text{Truck}}$ | $\text{mIoU}^{\text{C}}$ | $\text{mIoU}^{\text{I}}$ |
|---|---|---|---|---|---|
| DeepLabV3+ | $84.94 \pm 0.97$ | $80.90 \pm 0.16$ | $55.26 \pm 0.79$ | $70.17 \pm 0.16$ | $76.21 \pm 0.05$ |
| DeepLabV3 | $81.48 \pm 1.43$ | $80.07 \pm 0.06$ | $53.87 \pm 0.94$ | $69.09 \pm 0.06$ | $75.26 \pm 0.03$ |
| UNet | $65.58 \pm 0.59$ | $77.12 \pm 0.17$ | $53.00 \pm 0.61$ | $68.78 \pm 0.09$ | $75.36 \pm 0.06$ |

**Image-level histograms vary across datasets for the same architecture, but are similar across architectures for the same dataset.** The same architecture yields significantly different shapes of image-level histograms across various datasets. We believe it has to do with the coverage of a dataset (the fraction of classes appear in each image, discussed in Appendix B.2): a dataset with low coverage (e.g. ADE20K and COCO-Stuff) often results in a spread-out histogram with fat tails. On the other hand, although it is possible that a model that has a medium score on all images obtains the same $\text{mIoU}^{\text{I}}$ as a model that performs exceedingly well on some images and very poorly on others, we find that different architectures generally produce similar histograms for the same dataset.

**Fine-grained mIoUs exhibit less bias towards large objects.** For example, in Cityscapes, the class `Truck` has high dataset-level size imbalance (see Figure 4). As demonstrated in Table 3, compared with DeepLabV3-ResNet101, UNet-ResNet101 obtains a low value on $\text{IoU}^{\text{D}}_{\text{Truck}}$ (15% lower) and $\text{mIoU}^{\text{D}}$ (3% lower), since it has a limited reception field and struggles with large objects such as trucks that are close to the camera. On the other hand, it is capable of identifying small objects. As a result, its performance on $\text{IoU}^{\text{C}}_{\text{Truck}}$, $\text{mIoU}^{\text{C}}$ and $\text{mIoU}^{\text{I}}$ is similar to that of DeepLabV3-ResNet101.

**Architecture designs and loss functions are vital for optimizing fine-grained mIoUs.** We have seen from previous findings that architecture designs play an important role in optimizing fine-grained mIoUs. The influence of loss functions is also significant. We leave discussions of architecture designs in section 5.2.1 and loss functions in section 5.2.2. These insights contribute to the development of best practices for optimizing fine-grained mIoUs.

## 5.2 Best Practices

In this section, we explore best practices for optimizing fine-grained mIoUs. In particular, we underscore two key design choices: (i) the aggregation of multi-level features is as vital as incorporating modules that increase the receptive field, and (ii) the alignment of the loss function with the evaluation metrics is paramount.

### 5.2.1 Architecture Designs

Our study reveals the importance of aggregating multi-scale feature maps for optimizing fine-grained metrics. DeepLabV3 and PSPNet seek to increase the receptive field through the ASPP module [6] and the PPM module [58], respectively. The large receptive field helps to capture large objects, resulting in high values on $\text{mIoU}^{\text{D}}$. However, these methods lack the fusion of multi-scale feature

maps and lose the ability to perceive multi-scale objects due to aggregated down-sampling. On the other hand, UNet does not incorporate a separate module to increase the receptive field and therefore underperforms on $mIoU^D$. Nevertheless, it performs comparably to DeepLabV3 and PSPNet on fine-grained metrics. The aggregation of multi-scale feature maps in UNet is critical for detecting small objects and achieving high values on fine-grained metrics[3].

This observation is further confirmed by comparing DeepLabV3 with its successor, DeepLabV3+. The sole architectural difference between them is the incorporation of a feature aggregation branch in DeepLabV3+. Although they produce similar results on $mIoU^D$, DeepLabV3+ typically outperforms DeepLabV3 on $mIoU^C$.

It is encouraging to see that state-of-the-art methods, such as DeepLabV3+, UPerNet, and SegFormer, all incorporate modules to increase the receptive field and to aggregate multi-level feature maps. Such design choices align well with the objective of fine-grained mIoUs. We also advocate for the design of more specialized modules for perceiving small objects which can further improve the performance on fine-grained mIoUs.

### 5.2.2 Loss Functions

Aligning loss functions with evaluation metrics is a widely accepted practice in semantic segmentation [47, 2, 56, 14, 57, 50, 51]. The conventional approach involves combining the cross-entropy loss (CE) and IoU losses [2, 56, 50, 51], which extend discrete $mIoU^D$ with continuous surrogates. With the introduction of fine-grained mIoUs, it is straightforward to adjust the losses to optimize for these new metrics. Specifically, we explore the Jaccard metric loss (JML) [50] and its variants for optimizing $mIoU^I$ and $mIoU^C$. From a high-level, JML relaxes set counting (intersection and union) with simple $L^1$ norm functions. More details of JML are in Appendix F.

In Table 4, we compare DeepLabV3+-ResNet101 trained with various loss functions on Cityscapes and ADE20K, respectively. Specifically, (D, I, C) represents the fraction of $mIoU^D$, $mIoU^I$ and $mIoU^C$ in JML. For example, we consider the CE baseline: (0, 0, 0) and variants of JML to only optimize $mIoU^D$: (1, 0, 0) and a uniform mixture of three: $(\frac{1}{3}, \frac{1}{3}, \frac{1}{3})$. The main takeaways are as follows:

**The impact of loss functions is more pronounced when evaluated with fine-grained metrics.** Specifically, we notice improvements on $mIoU^C$ over CE reaching nearly 3% on Cityscapes and 7% on ADE20K. We also observe substantial increments on $mIoU^I$. Roughly speaking, Acc $\geq mIoU^D \geq mIoU^C$ in terms of class/size imbalance, while CE directly optimizes Acc. Therefore, it becomes more crucial to choose the correct loss functions when evaluated with fine-grained metrics.

**There exist trade-offs among different JML variants.** For example, compared with other JML variants, (1, 0, 0) achieves the highest $mIoU^D$ in both datasets, but have low values on $mIoU^I$ and $mIoU^C$. We use (0, 0.5, 0.5) as the default setting in our benchmark, but alternate ratios may be considered depending on the evaluation metrics.

A notable observation is that CE surpasses (0, 1, 0) in terms of $mIoU^D$ on Cityscapes. Interestingly, many segmentation codebases, including SMP [25] and MMSegmentation [8], calculate the Jaccard/Dice loss on a per-GPU basis. When training on GPUs with limited memory, the per-GPU batch size is often small. As a result, the loss function tends to mirror the behavior of (0, 1, 0), potentially leading to sub-optimal results as measured by $mIoU^D$.

Table 4: Comparing different loss functions using DeepLabV3+-ResNet101 on Cityscapes and ADE20K. (D, I, C) represents the fraction of $mIoU^D$, $mIoU^I$ and $mIoU^C$ in JML, respectively. Red: the best in a column. Green: the worst in a column.

| (D, I, C) | Cityscapes | | | ADE20K | | |
|---|---|---|---|---|---|---|
| | $mIoU^D$ | $mIoU^I$ | $mIoU^C$ | $mIoU^D$ | $mIoU^I$ | $mIoU^C$ |
| (0, 0, 0) | $80.67 \pm 0.36$ | $74.57 \pm 0.09$ | $67.61 \pm 0.12$ | $45.01 \pm 0.13$ | $53.73 \pm 0.09$ | $39.66 \pm 0.13$ |
| (1, 0, 0) | $81.22 \pm 0.27$ | $75.34 \pm 0.09$ | $68.97 \pm 0.19$ | $47.21 \pm 0.26$ | $57.74 \pm 0.11$ | $46.69 \pm 0.25$ |
| (0, 1, 0) | $80.29 \pm 0.28$ | $76.28 \pm 0.02$ | $69.97 \pm 0.09$ | $45.83 \pm 0.19$ | $58.37 \pm 0.03$ | $45.55 \pm 0.25$ |
| (0, 0, 1) | $80.75 \pm 0.24$ | $76.21 \pm 0.08$ | $70.49 \pm 0.19$ | $46.55 \pm 0.54$ | $58.74 \pm 0.09$ | $46.97 \pm 0.23$ |
| $(\frac{1}{3}, \frac{1}{3}, \frac{1}{3})$ | $81.13 \pm 0.10$ | $76.19 \pm 0.05$ | $70.08 \pm 0.15$ | $46.92 \pm 0.25$ | $58.51 \pm 0.02$ | $46.77 \pm 0.07$ |
| $(0, \frac{1}{2}, \frac{1}{2})$ | $80.91 \pm 0.17$ | $76.21 \pm 0.05$ | $70.18 \pm 0.16$ | $46.32 \pm 0.33$ | $58.79 \pm 0.04$ | $46.52 \pm 0.30$ |

---

[3]This might explain the efficacy of UNet for medical datasets. Typically, these datasets exhibit small lesion areas but lack extremely large objects that necessitate a large receptive field.

# 6 Discussion

## 6.1 Limitation

We propose three fine-grained mIoUs: $\text{mIoU}^I$, $\text{mIoU}^C$, $\text{mIoU}^K$, along with their associated worst-case metrics. These metrics bring significant advantages over traditional $\text{mIoU}^D$. However, they are not without shortcomings, and users should be aware of these when employing them in research or applications. Specifically, $\text{mIoU}^I$ and $\text{mIoU}^C$ fall short when it comes to addressing instance-level imbalance. Meanwhile, $\text{mIoU}^K$ serves merely as an approximation of the per-instance metric, rather than an exact representation. Beisdes, conducting image-level analyses, such as image-level histograms and worst-case images, proves challenging with $\text{mIoU}^C$ and $\text{mIoU}^K$. Although such capabilities are accommodated by $\text{mIoU}^I$, it has an inherent bias towards classes that are more frequently represented across images.

Additionally, it is important to note that our conclusions are based on a specific training setting, including choices of optimizer, loss function, and other hyperparameters. There is potential for varied findings when altering this setting, especially when it comes to the choice of the loss function.

Finally, our experiments primarily focus on natural and aerial datasets, excluding medical datasets from our analysis. Besides, while our core emphasis is on addressing size imbalance in relation to IoU, the concept of calculating segmentation metrics in a fine-grained manner could be extended to other metrics, such as mAcc, the Dice score, the calibration error [19], etc. We plan to address these limitations in future work.

## 6.2 Suggestion

While $\text{mIoU}^I$ might show a bias towards more common classes, it remains invaluable for users to analyze the dataset with image-level information, as well as provide a holistic evaluation of various segmentation methods. Given that $\text{mIoU}^C$ is a good proxy of $\text{mIoU}^K$, and considering many datasets lack instance-level labels, we believe $\text{mIoU}^C$ adequately serves as an instance-level approximation and could be used in most cases.

Regarding the worst-case metrics, we examine three variants: $\text{mIoU}^{C^{\bar{q}}}$, $\text{mIoU}^{C^5}$ and $\text{mIoU}^{C^1}$. We recommend the use of $\text{mIoU}^{C^{\bar{q}}}$ in applications where worst-case outcomes are prioritized. We provide evaluations using both $\text{mIoU}^{C^5}$ and $\text{mIoU}^{C^1}$ for the sake of comprehensiveness, but believe that adopting either one is sufficient. While utilizing $q = 1$ might be suitable for larger datasets like ADE20K to emphasize these exceptionally difficult examples, it could introduce significant variance for smaller datasets such as DeepGlobe Land. For the latter, a value of $q = 5$ or even larger might be more appropriate. With these considerations, dataset organizers can choose the metric that best fits their context.

# 7 Conclusion

In conclusion, this study provides crucial insights into the limitations of traditional per-dataset mean intersection over union and advocates for the adoption of fine-grained mIoUs, as well as the corresponding worst-case metrics. We argue that these metrics provide a less biased and more comprehensive evaluation, which is vital given the inherent class and size imbalances found in many segmentation datasets. Besides, these fine-grained metrics furnish a wealth of statistical information that can guide practitioners in making more informed decisions when evaluating segmentation methods for specific applications. Furthermore, they offer valuable insights for model and dataset auditing, aiding in the rectification of corrupted labels.

Our extensive benchmark study, covering 12 varied natural and aerial datasets and featuring 15 modern neural network architectures, underscores the importance of not relying solely on a single metric and affirms the potential of fine-grained metrics to reduce the bias towards large objects. It also sheds light on the significant roles played by architectural designs and loss functions, leading to best practices in optimizing fine-grained metrics.

## Acknowledgments and Disclosure of Funding

We acknowledge support from the Research Foundation - Flanders (FWO) through project numbers G0A1319N and S001421N, and funding from the Flemish Government under the Onderzoeksprogramma Artificiële Intelligentie (AI) Vlaanderen programme. The resources and services used in this work were provided by the VSC (Flemish Supercomputer Center), funded by the FWO and the Flemish Government.

M.B.B. is entitled to stock options in Mona.health, a KU Leuven spinoff.

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

# Contents

# A Models

We select segmentation methods that are included in MMSegmentation [8] and backbones that are supported by timm [52] for our benchmark study. An overview of 15 selected models are presented in Table 5. Specifically, DeepLabV3-ResNet101 and PSPNet-ResNet101 are the only models that leverage an extra module to increase the reception field, but does not utilize multi-scale features. Conversely, UNet-ResNet101 is the only model that does not have an extra module to increase the reception field, but merges hierarchical features. A brief introduction of each segmentation method and backbone is below.

Table 5: An overview of 15 selected models. FLOPs (G) is computed with an input size of $512 \times 1024$. Inference latency (ms) is measured with the same input size on a NVIDIA A100. Training memory is estimated using a ground-truth size of $8 \times 19 \times 512 \times 1024$ (`batch_size, num_classes, H, W`), also on a NVIDIA A100. RF: if the model adopts an extra module to increase the reception field except aggregated downsampling. MS: if the model utilizes multi-scale features. Yes*: SegFormer does not adopt an extra module to increase the reception field but it has a large reception field enabled by the MiT backbones.

| Method | Backbone | #Params (M) | FLOPs (G) | Latency (ms) | Memory (GB) | RF | MS |
|---|---|---|---|---|---|---|---|
| DeepLabV3+ [7] | ResNet101 [21] | 62.58 | 528.32 | 26.47 | 42.68 | Yes | Yes |
| DeepLabV3+ [7] | ResNet50 [21] | 43.59 | 372.74 | 17.92 | 29.72 | Yes | Yes |
| DeepLabV3+ [7] | ResNet34 [21] | 22.57 | 194.52 | 8.11 | 10.80 | Yes | Yes |
| DeepLabV3+ [7] | ResNet18 [21] | 12.47 | 109.90 | 5.50 | 8.69 | Yes | Yes |
| DeepLabV3+ [7] | EfficientNetB0 [46] | 4.65 | 34.52 | 10.16 | 23.96 | Yes | Yes |
| DeepLabV3+ [7] | MobileNetV2 [44] | 2.86 | 25.05 | 6.45 | 18.64 | Yes | Yes |
| DeepLabV3+ [7] | MobileViTV2 [36] | 5.73 | 52.59 | 26.81 | 21.32 | Yes | Yes |
| UPerNet [53] | ResNet101 [21] | 85.40 | 517.95 | 20.20 | 22.63 | Yes | Yes |
| UPerNet [53] | ConvNeXt [32] | 117.21 | 254.43 | 30.64 | 22.87 | Yes | Yes |
| UPerNet [53] | MiTB4 [54] | 91.22 | 545.34 | 43.60 | 38.97 | Yes | Yes |
| SegFormer [54] | MiTB4 [54] | 63.17 | 163.01 | 36.08 | 36.53 | Yes* | Yes |
| SegFormer [54] | MiTB2 [54] | 26.52 | 87.16 | 17.65 | 21.73 | Yes* | Yes |
| DeepLabV3 [6] | ResNet101 [21] | 87.10 | 714.73 | 26.20 | 33.99 | Yes | No |
| PSPNet [58] | ResNet101 [21] | 67.96 | 532.31 | 23.90 | 33.98 | Yes | No |
| UNet [42] | ResNet101 [21] | 95.06 | 448.25 | 22.78 | 28.16 | No | Yes |

## A.1 Segmentation Methods

**UNet [42]** is a classic encoder-decoder architecture for semantic segmentation. The encoder is a feature extractor and the decoder consists of a series of up-convolution operations followed by concatenation with the high-resolution features from the encoder path, and two regular convolutions. A crucial feature of UNet is the skip connections between layers at the same level in the encoder and the decoder. These connections allow the network to use both the context information from the low-level features and the spatial information from the high-resolution features, thereby enabling precise localization.

**DeepLabV3 [6]** attaches the Atrous Spatial Pyramid Pooling (ASPP) module to the output of the feature extractor. ASPP applies atrous (dilated) convolutions at a variety of dilation rates in parallel, allowing the model to handle objects of diverse sizes in the image.

**DeepLabV3+ [7]** builds upon DeepLabV3 and retains the use of the ASPP module. Its primary enhancement lies in incorporating an encoder-decoder structure where the decoder upsamples the low-level features from the encoder and combines them with high-level features to generate detailed segmentations.

**PSPNet [58]** adopts the Pyramid Pooling Module (PPM) to capture different levels of details in the scene. The idea behind this module is to apply pooling operations at varying scales and concatenate the output alongside the original feature map.

**UPerNet [53]** applies an encoder-decoder structure. The encoder takes inspiration from the Feature Pyramid Network (FPN) [29], creating a pyramid of features extracted at multiple scales. The decoder merges multi-level feature maps and adopts PPM at the lowest FPN level to bring useful global prior representations.

**SegFormer [54]** features a lightweight decoder that solely consists of MLP layers for upsampling and fusing multi-level features. It avoids computationally intensive components, and the key for this simplified design is the large receptive field provided by the hierarchical transformer encoder.

## A.2 Backbones

**ResNet [21]** is a type of CNN that uses shortcut connections, or "skip connections". These shortcut connections enable the network to learn an identity function, ensuring that the output of the block is at least as good as its input, and hence the term "residual". This identity function makes it easier for the network to learn and allows gradients to flow directly through the network during backpropagation, which can alleviate the problem of vanishing gradients in deep networks.

**ConvNeXt [32]** is a pure convolutional model, inspired by the design of vision transformers [13]. It consists of several macro designs (stage ratios, patchified stem, etc), micro-designs (replacing ReLU with GELU [23], substituting BN [26] with LN [1], etc), modern training techniques (AdamW [33], label smoothing [45], etc) and other designs (large kernel sizes, inverted bottlenecks, etc) to modernize ResNets.

**EfficientNet [46]** is a family of CNN architectures innovatively designed for model scaling. Efficient-Net's central insight is recognizing the interactive nature of the network width, depth, and resolution. Therefore, they need to be scaled together to maintain model efficiency. The key is to adopt a neural architecture search [60] approach to find the best scaling combinations.

**MobileNetV2 [44]** is designed to be a lightweight, efficient network architecture optimized for speed and size, making it particularly well-suited for mobile and embedded vision applications. It adopts depth-wise separable convolutions from MobileNetV1 [24] to reduce computational costs. Additionally, it leverages linear bottleneck layers to prevent nonlinearities from destroying too much information and inverted residuals blocks to facilitate gradient propagation across multiple layers.

**MiT [54]** is a type of vision transformer [13] specifically optimized for semantic segmentation tasks. One key aspect of MiT is its hierarchical structure. The model comprises several stages, forming a multi-scale structure that is effective for semantic segmentation. Another noteworthy feature of MiT is its relatively lightweight nature compared to some other vision Transformers, making it more practical for real-world deployment.

**MobileViTV2 [36]** is a lightweight and low-latency hybrid network for mobile vision tasks. It follows the macro-architecture of MobileViT [35], which adopt a hybrid design of MobileNetV2 blocks [44] and self-attention operations [48]. Its main innovation is a separable self-attention that reduces the computational complexity of self-attention from $O(n^2)$ to $O(n)$.

# B Datasets

Our benchmark include 12 datasets that cover various domains including: (i) street scenes [4, 9, 39, 11, 43], (ii) "thing" and "stuff" [18, 38, 59, 5], (iii) aerial scenes [12]. The number of images in the dataset ranges from 50 (e.g. Nighttime Driving and Dark Zurich) to more than a hundred thousand (e.g. COCO-Stuff); the number of classes varies from binary (e.g. DeepGlobe Road and DeepGlobe Building) to over a hundred (e.g. ADE20K and COCO-Stuff). Table 6 presents an overview of 12 selected datasets. Following is a brief introduction to each.

**Cityscapes [9]** is a large-scale dataset that focuses on semantic understanding of urban street scenes. It contains a diverse set of stereo video sequences recorded in street scenes from 50 different cities, with high quality pixel-level annotations of 5,000 frames (2,975 for training, 500 for validation and

Table 6: An overview of 12 selected datasets.

| Dataset | Scene | #Train | #Val | #Classes | Coverage (%) |
|---|---|---|---|---|---|
| Cityscapes [9] | Urban | 2,975 | 500 | 19 | $63.22 \pm 18.88$ |
| Nighttime Driving [11] | Urban | - | 50 | 19 | $34.42 \pm 24.50$ |
| Dark Zurich [43] | Urban | - | 50 | 19 | $57.99 \pm 16.28$ |
| Mapillary Vistas [39] | Urban | 18,000 | 2,000 | 65 | $30.18 \pm 22.23$ |
| CamVid [4] | Urban | 469 | 232 | 11 | $89.53 \pm 9.18$ |
| ADE20K [59] | "Thing" & "stuff" | 20,210 | 2,000 | 150 | $5.63 \pm 50.62$ |
| COCO-Stuff [5] | "Thing" & "stuff" | 118,287 | 5,000 | 171 | $4.92 \pm 54.21$ |
| PASCAL VOC [18] | "Thing"-only | 10,582 | 1,449 | 21 | $11.59 \pm 28.54$ |
| PASCAL Context [38] | "Thing" & "stuff" | 4,996 | 5,104 | 60 | $9.60 \pm 41.77$ |
| DeepGlobe Land [12] | Aerial | 643 | 160 | 6 | $63.33 \pm 27.09$ |
| DeepGlobe Road [12] | Aerial | 4,981 | 1,245 | 2 | $100 \pm 0$ |
| DeepGlobe Building [12, 15] | Aerial | 8,476 | 2,117 | 2 | $88.97 \pm 23.30$ |

1,525 for testing). In addition, there are 20,000 coarsely annotated frames also included in the dataset. In this study, we utilize the finely annotated training and validation images exclusively.

**Nighttime Driving [11] and Dark Zurich [43]** are datasets that provide images of various road scenes captured during the nighttime. They are designed to facilitate research in autonomous driving, and specifically addresses the challenges associated with low-light and nighttime conditions. They incorporate fine pixel-level Cityscapes [9] labels.

**Mapillary Vistas [39]** is a large-scale street-level imagery dataset used primarily for semantic segmentation tasks. It is one of the most diverse publicly available datasets of street-level imagery, collected from numerous cities around the world under a variety of weather and lighting conditions. While images are mainly taken from mobile devices, in general the dataset spans a wide range of different camera types, including head- or car-mounted ones. This diversity in viewpoints is useful for training more robust machine learning models that can generalize well to novel scenarios.

**CamVid [4]** is one of the pioneering datasets used for semantic segmentation. The dataset comprises of footage captured from the perspective of a driving automobile, providing a vision of the road scene ahead. It has been collected over different times of the day, offering an excellent selection of varying lighting conditions and appearances.

**ADE20K [59]** is a widely used dataset for scene parsing. One unique feature of the ADE20K dataset is its high diversity in object classes. The dataset has annotations for over 150 object categories, and includes a diverse set of scenes from both indoor and outdoor environments. This diversity makes the dataset challenging and well-suited for developing and benchmarking algorithms for semantic segmentation.

**COCO-Stuff [5]** is an extension of the original COCO [30] dataset which is renowned for its diverse set of high-quality, real-world images containing complex everyday scenes of common objects in their natural context. It augments the COCO dataset by adding pixel-level annotations for "stuff" classes. Thus, it contains the same images as the original COCO dataset and is well-suited for semantic segmentation tasks.

**PASCAL VOC [18]** is composed of images collected from the Flickr photo-sharing web-site. The labels include a diverse set of classes, including various types of vehicles, animals, and indoor objects. The original dataset contains 1,464 images for training, 1,449 images for validation, and a further 1,456 images for testing. Extra annotations are provided in [20], leading to a total of 10,582 training images.

**PASCAL Context [38]** is an extension of the popular PASCAL VOC dataset [18] where only object classes of interest were annotated, leaving a large portion of pixels marked as "background" or "unlabeled". In the PASCAL Context dataset, the goal is to provide a comprehensive understanding of the scene, thus all pixels in an image are assigned a label, giving the "context" of the objects.

**DeepGlobe [12]** contains high-resolution satellite images from around the globe, covering various geographical locations, features, and phenomena. It offers three distinct satellite image understanding datasets: land cover classification, road extraction, and building detection. With this rich assortment of large-scale satellite imagery, DeepGlobe aids in propelling advancements in geospatial analysis.

## B.1 "Thing" and "Stuff"

The differences between "thing" and "stuff" encompass various aspects [5]:

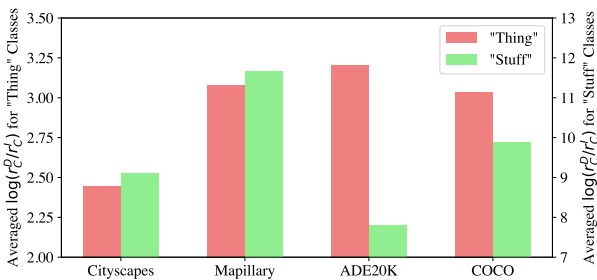

Figure 5: Results of averaged $\log \frac{r_c^D}{r_c^I}$ for "thing" and "stuff" classes on Cityscapes, Mapillary Vistas, ADE20K and COCO-Stuff.

- Shape: "thing" has characteristic shapes (e.g. person, car, phone), whereas "stuff" is amorphous (e.g. sky, road, water).
- Size: "thing" occurs in characteristic sizes with little variance, whereas "stuff" is highly variable in size.
- Parts: "thing" possess identifiable parts, whereas "stuff" does not. For example, a fragment of sky is still considered sky, but a wheel alone does not constitute a car.
- Instances: "stuff" is typically uncountable and has no clearly defined instances.
- Texture: "stuff" is typically highly textured.

The characterization of "thing" and "stuff" is usually a decision made by the dataset creator. However, many datasets do not define which classes are "thing" or "stuff" (e.g. Cityscapes, Mapillary Vistas, ADE20K, etc). In this work, for datasets that provide instance-level annotations (Cityscapes, Mapillary Vistas, ADE20K, COCO-Stuff), we treat classes that have instance-level labels as "thing" classes, and as "stuff" classes otherwise.

## B.2 Dataset Coverage

For each dataset, we calculate the proportion of distinct classes present in each image to the total number of classes within that dataset. We define the *coverage* of a dataset as the mean and the normalized standard deviation over all images. A dataset exhibiting a high mean and a low normalized standard deviation is considered to have high coverage, implying low variations in contexts.

Urban-scene datasets, such as Cityscapes, Mapillary Vistas, and CamVid, generally demonstrate high coverage. In contrast, "thing" and "stuff" datasets like ADE20K, COCO-Stuff, and PASCAL Context typically exhibit lower coverage as they incorporate a wide range of indoor and outdoor scenes. Datasets with low coverage often showcase more spread-out image-level histograms (see Appendix G) with fat tails, indicating a diverse range of image-level IoU values.

## B.3 Dataset Auditing

Images that consistently yield a low image-level score across different models can be inspected. These low scores could potentially be attributed to mislabeling within the images. For example, in COCO-Stuff, instances in "000000064574" and "000000053529" are erroneously labeled as "stuff" (as shown in Figures 31 - 33). This mislabeling leads to most models registering an image-level IoU of 0.

Additionally, the presence of inconsistencies between image- and object-level labels can lead to `NaN` values during the computation of mIoU$^K$, because the denominator in Eq. (9) will become zero. This provides a way to identify mislabels within the dataset. As demonstrated in Table 7, we find these mislabels exist in Cityscapes and ADE20K. Specifically, only a single mislabeled image is in the validation set (highlighted in red), and only a single mislabeled object is "stuff" (highlighted in green).

Table 7: Identified mislabels in Cityscapes and ADE20K. ID: object ID. Reason: 1 means that the object is present in the object-level label but absent in the image-level label; 2 means that the object is included in the image-level label but missing from the object-level label. Red: the image is in the validation set. Green: the object is "stuff".

| Dataset | File | Class | ID | Reason |
|---------|------|-------|-----|--------|
| Cityscapes | strasbourg_000000_029577 | Terrain | 22 | 2 |
| Cityscapes | hanover_000000_053604 | Rider | 25000 | 2 |
| Cityscapes | aachen_000022_000019 | Car | 26000 | 2 |
| ADE20K | ADE_val_00000899 | Door | 3407624 | 2 |
| ADE20K | ADE_train_00000939 | Door | 3407624 | 1 |
| ADE20K | ADE_train_00019938 | Door | 3407624 | 1 |
| ADE20K | ADE_train_00005928 | Seat | 14745351 | 1 |
| ADE20K | ADE_train_00018960 | Seat | 14745351 | 1 |
| ADE20K | ADE_train_00018960 | Seat | 15859465 | 2 |
| ADE20K | ADE_train_00015355 | Seat | 14745351 | 1 |
| ADE20K | ADE_train_00001176 | Seat | 14745351 | 1 |
| ADE20K | ADE_train_00010454 | Seat | 14745351 | 1 |
| ADE20K | ADE_train_00008027 | Seat | 14745351 | 1 |
| ADE20K | ADE_train_00016771 | Seat | 13361409 | 2 |
| ADE20K | ADE_train_00013611 | Seat | 14745351 | 1 |
| ADE20K | ADE_train_00013611 | Seat | 16116992 | 2 |
| ADE20K | ADE_train_00002139 | Seat | 14745351 | 1 |
| ADE20K | ADE_train_00005924 | Seat | 15204121 | 2 |
| ADE20K | ADE_train_00016746 | Seat | 14745351 | 1 |
| ADE20K | ADE_train_00008022 | Seat | 14745351 | 1 |
| ADE20K | ADE_train_00008022 | Seat | 14410752 | 2 |
| ADE20K | ADE_train_00019174 | Seat | 13034752 | 2 |
| ADE20K | ADE_train_00019174 | Seat | 16383749 | 2 |
| ADE20K | ADE_train_00019174 | Seat | 14673932 | 2 |
| ADE20K | ADE_train_00019174 | Seat | 13696785 | 2 |
| ADE20K | ADE_train_00019177 | Seat | 12909092 | 2 |
| ADE20K | ADE_train_00013522 | Seat | 14745351 | 1 |
| ADE20K | ADE_train_00001354 | Booth | 13369599 | 1 |
| ADE20K | ADE_train_00001387 | Booth | 13369599 | 1 |
| ADE20K | ADE_train_00001387 | Booth | 11469055 | 2 |
| ADE20K | ADE_train_00001392 | Booth | 13369599 | 1 |
| ADE20K | ADE_train_00001392 | Booth | 11796735 | 2 |
| ADE20K | ADE_train_00018451 | Booth | 13369599 | 1 |
| ADE20K | ADE_train_00015050 | Booth | 13369599 | 1 |
| ADE20K | ADE_train_00001573 | Booth | 13369599 | 1 |
| ADE20K | ADE_train_00001573 | Booth | 11469055 | 2 |
| ADE20K | ADE_train_00001573 | Booth | 13769215 | 2 |
| ADE20K | ADE_train_00001573 | Booth | 13893861 | 2 |
| ADE20K | ADE_train_00001573 | Booth | 11993599 | 2 |
| ADE20K | ADE_train_00001573 | Booth | 15275519 | 2 |
| ADE20K | ADE_train_00012787 | Booth | 13369599 | 1 |
| ADE20K | ADE_train_00012787 | Booth | 14943216 | 2 |
| ADE20K | ADE_train_00012787 | Booth | 11999999 | 2 |
| ADE20K | ADE_train_00012787 | Booth | 12715263 | 2 |
| ADE20K | ADE_train_00012713 | Booth | 13369599 | 1 |
| ADE20K | ADE_train_00014369 | Booth | 13369599 | 1 |
| ADE20K | ADE_train_00005268 | Booth | 13369599 | 1 |
| ADE20K | ADE_train_00001386 | Booth | 13369599 | 1 |
| ADE20K | ADE_train_00016930 | Booth | 13369599 | 2 |

## C  Implementation Details

**Training details.** We use AdamW [33] with a weight decay of 0.01. The learning rate starts from 1e-6 and linearly warms up during the first 1% iterations to the initial learning rate which is 6e-5 for ResNet101/50, ConvNeXt, MiTB4/B2, and 6e-4 for ResNet34/18, EfficientNetB0, MobileNetV2, MobileViTV2. The learning rate is then decayed in a "poly" policy with an exponent of 1. The number of training iterations and the crop size for each dataset is summarized in Table 8. All models are trained with a combination of the cross-entropy loss and the Jaccard metric loss (JML) [50], balancing with weights of 0.25 and 0.75, respectively. JML optimizes for $mIoU^I$ and $mIoU^C$ with equal weights, as discussed in Section 5.2.2. Data augmentations including (i) random scaling in the range of [0.5, 2], and (ii) random horizontal flipping with a probability of 0.5.

All results are based on three independent runs and are presented in the format of mean±standard deviation. We do not apply any inference-time augmentation such as multi-scale inference and flipping.

**Dataset splits.** For Nighttime Driving and Dark Zurich, we evaluate models trained on Cityscapes on their test sets, following MMSegmentation [8]. For CamVid, we merge "train" and "val" into the training set, and "test" as the test set. For DeepGlobe, since it does not provide an official data split, we select the first 80% (sorted by indexes) of training images as the training set and the rest as the test set. For all other datasets, we adhere to the official training/validation splits.

In general, our implementations (models, dataset preprocessing, training recipes) closely follow MMSegmentation [8], except that we adopt a smaller number of training iterations to save the computational costs and utilize recent training techniques [50, 33] for superior performance. As demonstrated in Table 1 and Table 2, we either match or exceed the results of MMSegmentation with less training iterations (see Table 8). Also note that our training hyper-parameters are not optimized for mIoU$^{\text{D}}$ (see Table 4).

Table 8: The number of training and warmup iterations and the crop size for each dataset. MMSegmentation: the number of training iterations in MMSegmentation [8].

| Dataset | MMSegmentation | #Iterations | Warmup | Crop Size |
|---|---|---|---|---|
| Cityscapes [9] | 80,000 | 40,000 | 400 | $512 \times 1024$ |
| Nighttime Driving [11] | - | - | - | $512 \times 1024$ |
| Dark Zurich [43] | - | - | - | $512 \times 1024$ |
| Mapillary Vistas [39] | - | 160,000 | 1,600 | $512 \times 1024$ |
| CamVid [4] | - | 20,000 | 200 | $512 \times 512$ |
| ADE20K [59] | 160,000 | 80,000 | 800 | $512 \times 512$ |
| COCO-Stuff [5] | 160,000 | 80,000 | 800 | $512 \times 512$ |
| PASCAL VOC [18] | 40,000 | 40,000 | 400 | $512 \times 512$ |
| PASCAL Context [38] | 80,000 | 40,000 | 400 | $512 \times 512$ |
| DeepGlobe Land [12] | - | 20,000 | 200 | $512 \times 512$ |
| DeepGlobe Road [12] | - | 40,000 | 400 | $512 \times 512$ |
| DeepGlobe Building [12, 15] | - | 40,000 | 400 | $512 \times 512$ |

## D  `NULL` **in Fine-grained mIoUs**

For each image, based on the presence or absence of a class in both the prediction and the ground truth, we can identify four cases:

1. The class is present in both the prediction and the ground truth.

2. The class is absent in both the prediction and the ground truth (true negatives).

3. The class is absent in the prediction but present in the ground truth (false negatives).

4. The class is present in the prediction but is absent in the ground truth (false positives).

In multi-class segmentation, for cases 1 and 3, we compute the per-image-per-class score as usual. In particular, the score for case 3 is 0. For cases 2 and 4, we define the score as NULL. In binary segmentation, if it is viewed as 2-class segmentation, then we can refer to the definition above. Otherwise, if it is considered as foreground-background segmentation, the definition for cases 1 and 3 remains consistent with the multi-class setting. For case 2, we define the score as 1 and for case 4, we define the score as 0. A summary is shown in Table 9.

Notably, in the multi-class setting, our definition differs from Csurka et al. [10]. In their definition, the score is 0 for case 4. Our modification aims at preventing the metric from overly reacting to minor false positives, e.g. a few pixels are mispredicted as a car absent from the ground truth. In safety-critical settings, false negatives should have higher importance, e.g. overlooking a car present in the ground truth. Overemphasizing such minor false positives might inadvertently diminish the significance of false negatives. A concerning scenario could be when the ground truth contains only a small number of classes, yet each unrelated class has a few pixels erroneously present in the prediction. It is pivotal to note that in our definition, these mispredicted pixels still incur penalties—they are false negatives with respect to the corresponding ground-truth classes.

Besides, we do not want to discriminate between false positives solely based on whether the predicted class is present in the ground truth or not. Generally speaking, mispredicting `Road` as either `Car` or `Truck` should incur a similar penalty. However, according to the definition of [10], if `Car` is present in the ground truth and `Truck` is not, mispredicting `Road` as `Car` may only result in a mild penalty, whereas mispredicting `Road` as `Truck` will incur a full penalty.

To further illustrate the differences between the definition of [10] and ours, consider an example image composed of just 4 pixels, with a total of 6 classes in the dataset, as presented in Table 10. We compare the two ways of computing the score, as detailed in Table 11. When calculating the score as prescribed by [10], false positives that appear in the prediction but not in the ground truth incur a considerable penalty. Consequently, the mean IoU value for this image is only 0.25 under their definition, but is 0.5 under ours.

For a more empirical comparison, in Table 12, we present the two ways for calculating mIoU$^C$ using DeepLabV3+-ResNet101. It is evident that, when the score is derived following [10], there is a more aggressive penalty, resulting in a generally lower value.

In the end, although we present our definition with the belief that it offers advantages over [10] in certain scenarios, we understand that each metric will have its pros and cons. It is essential to note that our aim is not to assert that our metrics are the best or only way, but to highlight this distinction. Ultimately, end users could select the version most aligned with their specific requirements.

Table 9: The definition of IoU$_{i,c}$ in multi-class and binary segmentation.

| Case | Multi-class | Binary |
|------|-------------|--------|
| 1 | IoU$_{i,c}$ | IoU$_{i,c}$ |
| 2 | NULL | 1 |
| 3 | 0 | 0 |
| 4 | NULL | 0 |

Table 10: Predicted and ground-truth classes of an example image composed of just 4 pixels, with a total number of 6 classes in the dataset.

| Pred | 0 | 2 | 1 | 3 |
|------|---|---|---|---|
| GT | 0 | 0 | 1 | 1 |

Table 11: The two ways of computing IoU$_{i,c}$ for the example image.

| [10] | 0.5 | 0.5 | 0 | 0 | NULL | NULL |
|------|-----|-----|------|------|------|------|
| Ours | 0.5 | 0.5 | NULL | NULL | NULL | NULL |

Table 12: The two ways of computing mIoU$^C$ using DeepLabV3+-ResNet101.

| Definition | Cityscapes | Mapillary | ADE20K | COCO-Stuff | VOC | Context | Land |
|------------|-----------|-----------|--------|------------|-----|---------|------|
| [10] | $56.14 \pm 0.43$ | $23.95 \pm 0.19$ | $20.23 \pm 0.05$ | $20.98 \pm 0.09$ | $62.30 \pm 0.58$ | $25.92 \pm 0.07$ | $40.14 \pm 1.39$ |
| Ours | $70.18 \pm 0.16$ | $46.23 \pm 0.12$ | $46.52 \pm 0.30$ | $41.98 \pm 0.04$ | $79.07 \pm 0.18$ | $55.89 \pm 0.07$ | $53.76 \pm 0.40$ |

# E  Other Per-instance mIoUs

Recall that we have instance-level TP$_{i,c,k}$, FN$_{i,c,k}$ and image-level FP$_{i,c}$. We can solve a minimization problem:

$$\min \sum_k \frac{\text{TP}_{i,c,k}}{\text{TP}_{i,c,k} + \text{FN}_{i,c,k} + \text{FP}_{i,c,k}} \tag{15}$$

$$\text{s.t.} \sum_k \text{FP}_{i,c,k} = \text{FP}_{i,c}. \tag{16}$$

Alternatively, we can replace the minimization with maximization. Denote the resulting instance-level values as IoU$_{\min c,i,k}$ and IoU$_{\max c,i,k}$, respectively. They provide a lower/upper bound on the value of IoU$_{i,c,k}$:

$$\text{IoU}_{\min c,i,k} \leq \text{IoU}_{i,c,k} \leq \text{IoU}_{\max c,i,k}. \tag{17}$$

Compared to distributing image-level FP according to the size of each instances (as we defined in Section 2), IoU$_{\min c,i,k}$ and IoU$_{\max c,i,k}$ may yield results misaligned with our intuition. For example, consider a case with two instances where $\text{TP}_1 = 1, \text{TP}_2 = 10, \text{FN}_1 = \text{FN}_2 = 0, \text{FP}_1 + \text{FP}_2 = 2$. Then we have

$$\frac{1}{1 + \text{FP}_1} + \frac{10}{10 + \text{FP}_2}, \tag{18}$$

which is minimized when $\text{FP}_1 = 2, \text{FP}_2 = 0$. Thus, IoU$_{\min c,i,k}$ will distribute FP to the smaller instance when there is an instance imbalance.

Consider another example with two instances such that $\text{TP}_1 = \text{TP}_2 = 10, \text{FN}_1 = \text{FN}_2 = 0, \text{FP}_1 + \text{FP}_2 = 10$. Then we have

$$\frac{10}{10 + \text{FP}_1} + \frac{10}{10 + \text{FP}_2}, \tag{19}$$

which is maximized when $\text{FP}_1 = 10, \text{FP}_2 = 0$ or $\text{FP}_1 = 0, \text{FP}_2 = 10$. Therefore, IoU$_{\max c,i,k}$ will distribute FP to one instance when instances are of the equal size.

# F   The Jaccard Metric Loss

IoU is defined over sets that are discrete and therefore cannot be directly optimized by neural networks. Given two sets $x, y \in \{0, 1\}^p$, the Jaccard metric loss (JML) [50] rewrites IoU as a function of set difference:

$$\text{IoU} = \frac{|x \cap y|}{|x \cup y|} = \frac{|x| + |y| - |x \Delta y|}{|x| + |y| + |x \Delta y|}, \tag{20}$$

and realizes that when $x, y \in [0, 1]^p$, set operations can be relaxed with norm functions:

$$|x| = \|x\|_1, |y| = \|y\|_1, |x \Delta y| = \|x - y\|_1. \tag{21}$$

Taking these together, the loss is defined as:

$$\text{JML} = 1 - \frac{\|x + y\|_1 - \|x - y\|_1}{\|x + y\|_1 + \|x - y\|_1}. \tag{22}$$

JML has the property that it is identical to the widely used soft Jaccard loss (SJL) [40, 41] when $y \in \{0, 1\}^p$, and it is a metric on $[0, 1]^p$. Therefore, it is compatible with soft labels, e.g. $y = 0.5$, while SJL is not [50].

JML extends discrete IoU with a continuous surrogate. It was used to optimize mIoU$^D$, but it is straightforward to adjust it to optimize fine-grained mIoUs by simply modifying how IoU values are aggregated during the loss computation.

# G   Results

In this section, we provide detailed tabular results, image-level histograms, and examples of the worst-case images for each dataset. In Table 13, we present the count of three most common worst-case images for each dataset.

Table 13: The count of three most common worst-case images for each dataset.

| Dataset | #1 | #2 | #3 |
|---|---|---|---|
| Cityscapes | 9 | 6 | 0 |
| Nighttime Driving | 6 | 2 | 1 |
| Dark Zurich | 4 | 4 | 2 |
| Mapillary Vistas | 10 | 2 | 1 |
| CamVid | 5 | 3 | 2 |
| ADE20K | 3 | 3 | 2 |
| COCO-Stuff | 11 | 2 | 2 |
| PASCAL VOC | 4 | 4 | 3 |
| PASCAL Context | 3 | 3 | 2 |
| DeepGlobe Land | 13 | 2 | 0 |
| DeepGlobe Road | 9 | 3 | 1 |
| DeepGlobe Building | 11 | 2 | 2 |

## G.1 Cityscapes

Table 14: Results: Cityscapes. Red: the best for each metric. Green: the worst for each metric.

| Method | Backbone | $mIoU^I$ / $mIoU^C$ / $mIoU^K$ / $mIoU^D$ | $mIoU^{I\bar{q}}$ / $mIoU^{C\bar{q}}$ / $mIoU^{K\bar{q}}$ / mAcc | $mIoU^{I5}$ / $mIoU^{C5}$ / $mIoU^{K5}$ / Acc | $mIoU^{I1}$ / $mIoU^{C1}$ / $mIoU^{K1}$ |
|---|---|---|---|---|---|
| DeepLabV3+ | ResNet101 | $76.21 \pm 0.05$ | $70.32 \pm 0.06$ | $59.17 \pm 0.38$ | $51.62 \pm 0.71$ |
| | | $70.18 \pm 0.16$ | $52.64 \pm 0.19$ | $17.13 \pm 0.13$ | $5.00 \pm 0.00$ |
| | | $69.77 \pm 0.16$ | $51.40 \pm 0.19$ | $16.11 \pm 0.07$ | $5.00 \pm 0.00$ |
| | | $80.91 \pm 0.17$ | $88.33 \pm 0.10$ | $96.38 \pm 0.01$ | |
| DeepLabV3+ | ResNet50 | $75.33 \pm 0.12$ | $69.31 \pm 0.19$ | $57.90 \pm 0.31$ | $51.30 \pm 0.91$ |
| | | $69.10 \pm 0.30$ | $51.65 \pm 0.42$ | $16.69 \pm 0.40$ | $4.67 \pm 0.47$ |
| | | $68.72 \pm 0.22$ | $50.39 \pm 0.28$ | $15.58 \pm 0.25$ | $4.67 \pm 0.47$ |
| | | $79.34 \pm 0.36$ | $86.85 \pm 0.42$ | $96.10 \pm 0.06$ | |
| DeepLabV3+ | ResNet34 | $75.24 \pm 0.07$ | $69.19 \pm 0.12$ | $57.50 \pm 0.55$ | $50.39 \pm 2.30$ |
| | | $68.58 \pm 0.17$ | $50.96 \pm 0.20$ | $17.33 \pm 0.48$ | $5.00 \pm 0.00$ |
| | | $68.06 \pm 0.20$ | $49.45 \pm 0.24$ | $15.74 \pm 0.46$ | $5.00 \pm 0.00$ |
| | | $78.61 \pm 0.14$ | $86.54 \pm 0.12$ | $96.18 \pm 0.04$ | |
| DeepLabV3+ | ResNet18 | $74.23 \pm 0.04$ | $68.07 \pm 0.06$ | $56.12 \pm 0.36$ | $49.65 \pm 0.73$ |
| | | $67.44 \pm 0.11$ | $49.80 \pm 0.22$ | $16.37 \pm 0.28$ | $4.67 \pm 0.47$ |
| | | $66.91 \pm 0.18$ | $48.35 \pm 0.25$ | $15.32 \pm 0.17$ | $5.00 \pm 0.00$ |
| | | $77.07 \pm 0.17$ | $85.31 \pm 0.38$ | $95.96 \pm 0.03$ | |
| DeepLabV3+ | EfficientNetB0 | $74.60 \pm 0.04$ | $68.53 \pm 0.10$ | $57.38 \pm 0.36$ | $50.59 \pm 1.21$ |
| | | $67.60 \pm 0.04$ | $50.07 \pm 0.10$ | $17.43 \pm 0.15$ | $5.67 \pm 0.47$ |
| | | $66.98 \pm 0.03$ | $48.45 \pm 0.06$ | $16.04 \pm 0.21$ | $5.67 \pm 0.47$ |
| | | $77.40 \pm 0.13$ | $85.65 \pm 0.12$ | $96.07 \pm 0.04$ | |
| DeepLabV3+ | MobileNetV2 | $73.60 \pm 0.09$ | $67.41 \pm 0.07$ | $55.49 \pm 0.47$ | $49.18 \pm 0.87$ |
| | | $66.79 \pm 0.09$ | $49.09 \pm 0.09$ | $15.43 \pm 0.31$ | $5.00 \pm 0.00$ |
| | | $66.38 \pm 0.10$ | $47.75 \pm 0.08$ | $14.58 \pm 0.21$ | $5.00 \pm 0.00$ |
| | | $75.40 \pm 0.43$ | $84.28 \pm 0.44$ | $95.78 \pm 0.03$ | |
| DeepLabV3+ | MobileViTV2 | $75.06 \pm 0.06$ | $69.12 \pm 0.12$ | $58.02 \pm 0.38$ | $51.53 \pm 0.96$ |
| | | $68.47 \pm 0.25$ | $51.18 \pm 0.25$ | $17.80 \pm 0.15$ | $6.00 \pm 0.00$ |
| | | $67.99 \pm 0.27$ | $49.80 \pm 0.28$ | $16.47 \pm 0.14$ | $6.33 \pm 0.47$ |
| | | $78.17 \pm 0.39$ | $86.56 \pm 0.19$ | $96.10 \pm 0.02$ | |
| UPerNet | ResNet101 | $75.73 \pm 0.04$ | $69.66 \pm 0.07$ | $58.09 \pm 0.22$ | $51.61 \pm 0.95$ |
| | | $69.24 \pm 0.17$ | $51.66 \pm 0.19$ | $17.45 \pm 0.08$ | $4.67 \pm 0.47$ |
| | | $68.78 \pm 0.10$ | $50.36 \pm 0.12$ | $16.47 \pm 0.09$ | $5.00 \pm 0.82$ |
| | | $79.89 \pm 0.10$ | $87.22 \pm 0.09$ | $96.22 \pm 0.04$ | |
| UPerNet | ConvNeXt | $77.13 \pm 0.13$ | $71.28 \pm 0.19$ | $60.41 \pm 0.40$ | $53.86 \pm 0.81$ |
| | | $71.22 \pm 0.22$ | $53.87 \pm 0.33$ | $19.48 \pm 0.44$ | $6.00 \pm 0.00$ |
| | | $70.75 \pm 0.17$ | $52.56 \pm 0.27$ | $17.98 \pm 0.39$ | $5.67 \pm 0.47$ |
| | | $81.81 \pm 0.11$ | $88.30 \pm 0.07$ | $96.62 \pm 0.02$ | |
| UPerNet | MiTB4 | $76.52 \pm 0.16$ | $70.82 \pm 0.23$ | $60.50 \pm 0.64$ | $56.09 \pm 1.67$ |
| | | $70.64 \pm 0.11$ | $53.55 \pm 0.20$ | $19.79 \pm 0.33$ | $6.33 \pm 0.47$ |
| | | $70.02 \pm 0.13$ | $51.88 \pm 0.22$ | $17.96 \pm 0.33$ | $6.33 \pm 0.47$ |
| | | $80.99 \pm 0.20$ | $88.10 \pm 0.23$ | $96.51 \pm 0.04$ | |
| SegFormer | MiTB4 | $76.78 \pm 0.07$ | $71.05 \pm 0.13$ | $60.12 \pm 0.43$ | $55.15 \pm 1.36$ |
| | | $70.68 \pm 0.16$ | $53.56 \pm 0.14$ | $19.95 \pm 0.13$ | $6.33 \pm 0.47$ |
| | | $70.14 \pm 0.19$ | $52.10 \pm 0.18$ | $18.22 \pm 0.14$ | $6.33 \pm 0.47$ |
| | | $80.75 \pm 0.19$ | $87.96 \pm 0.17$ | $96.55 \pm 0.01$ | |
| SegFormer | MiTB2 | $75.66 \pm 0.07$ | $69.86 \pm 0.09$ | $58.79 \pm 0.31$ | $53.77 \pm 0.55$ |
| | | $69.21 \pm 0.16$ | $51.86 \pm 0.15$ | $18.75 \pm 0.41$ | $6.33 \pm 0.47$ |
| | | $68.63 \pm 0.14$ | $50.41 \pm 0.12$ | $17.41 \pm 0.36$ | $6.33 \pm 0.47$ |
| | | $79.96 \pm 0.05$ | $87.40 \pm 0.00$ | $96.36 \pm 0.01$ | |
| DeepLabV3 | ResNet101 | $75.26 \pm 0.03$ | $69.35 \pm 0.07$ | $58.44 \pm 0.23$ | $51.10 \pm 0.68$ |
| | | $69.10 \pm 0.07$ | $51.60 \pm 0.12$ | $16.91 \pm 0.09$ | $4.67 \pm 0.47$ |
| | | $68.51 \pm 0.12$ | $50.06 \pm 0.15$ | $15.79 \pm 0.05$ | $4.67 \pm 0.47$ |
| | | $80.07 \pm 0.06$ | $87.77 \pm 0.13$ | $96.25 \pm 0.02$ | |
| PSPNet | ResNet101 | $75.18 \pm 0.07$ | $69.27 \pm 0.11$ | $58.83 \pm 0.29$ | $53.18 \pm 1.10$ |
| | | $69.05 \pm 0.15$ | $51.53 \pm 0.21$ | $16.94 \pm 0.08$ | $5.00 \pm 0.00$ |
| | | $68.58 \pm 0.14$ | $50.18 \pm 0.18$ | $15.84 \pm 0.09$ | $5.00 \pm 0.00$ |
| | | $80.08 \pm 0.13$ | $87.62 \pm 0.06$ | $96.18 \pm 0.05$ | |
| UNet | ResNet101 | $75.36 \pm 0.06$ | $69.43 \pm 0.07$ | $58.16 \pm 0.20$ | $52.19 \pm 0.94$ |
| | | $68.79 \pm 0.10$ | $51.28 \pm 0.08$ | $16.59 \pm 0.16$ | $5.00 \pm 0.00$ |
| | | $68.31 \pm 0.20$ | $49.96 \pm 0.22$ | $15.66 \pm 0.17$ | $5.00 \pm 0.00$ |
| | | $77.12 \pm 0.18$ | $85.30 \pm 0.04$ | $96.05 \pm 0.00$ | |

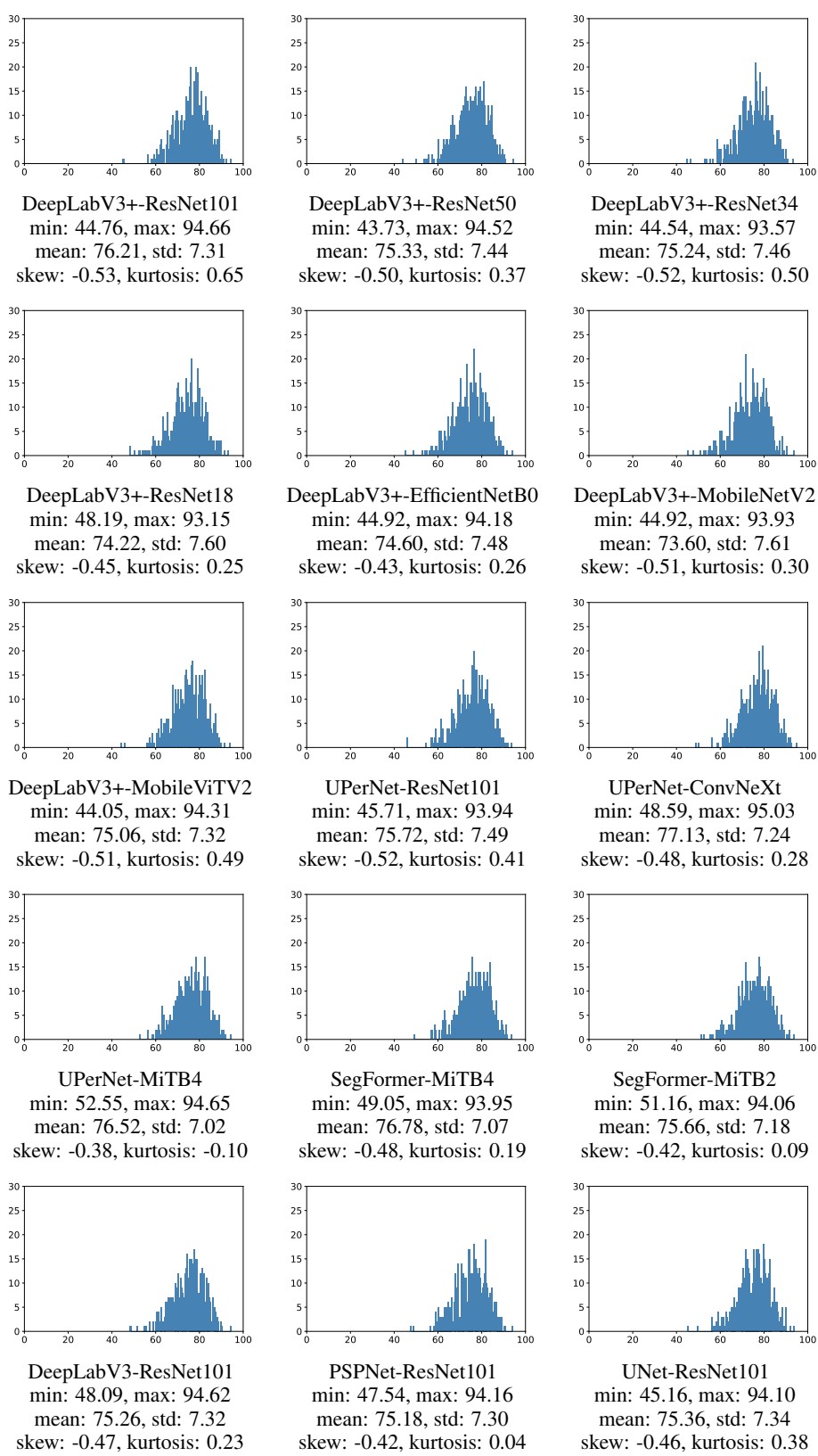

Figure 6: Histograms: Cityscapes.

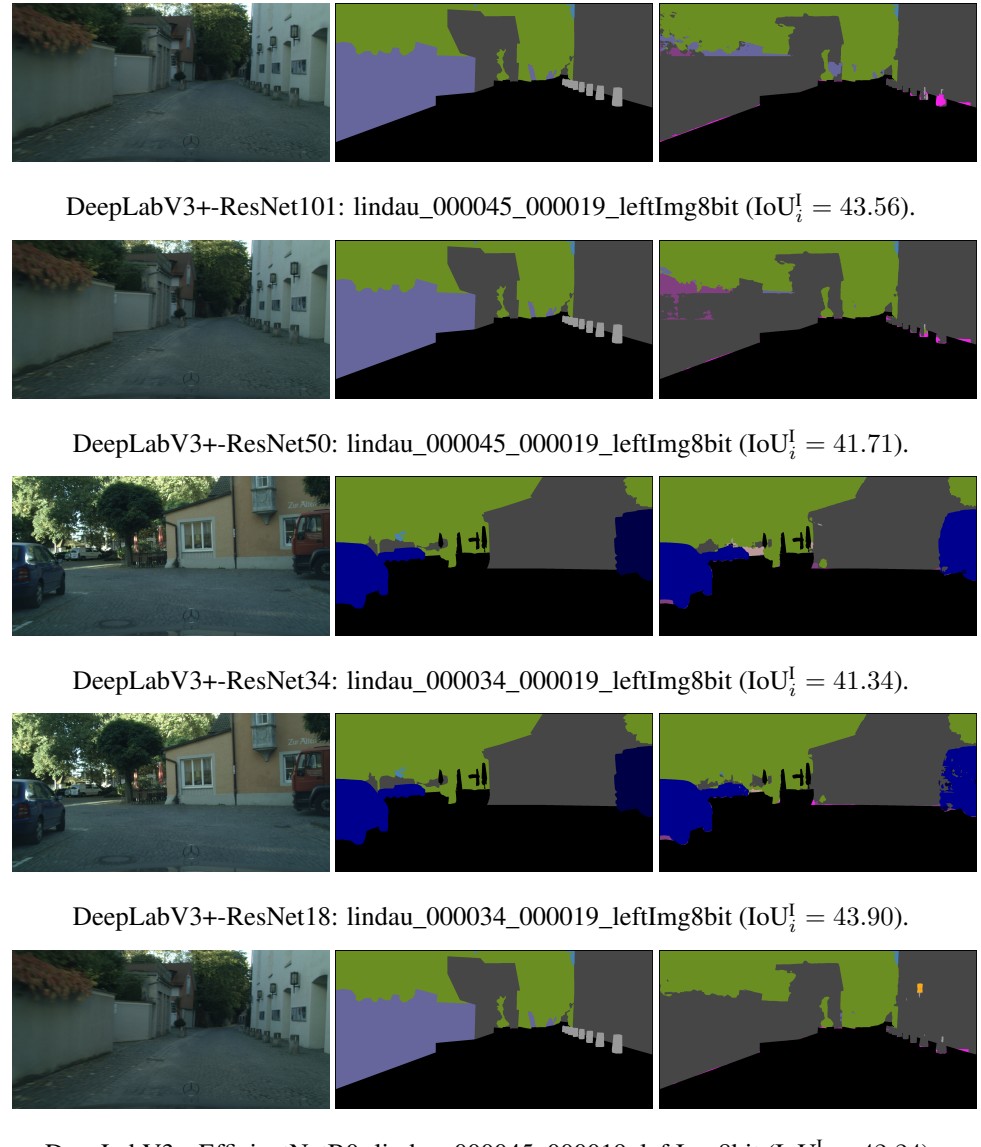

DeepLabV3+-ResNet101: lindau_000045_000019_leftImg8bit (IoU$_i^{\mathrm{I}}$ = 43.56).

DeepLabV3+-ResNet50: lindau_000045_000019_leftImg8bit (IoU$_i^{\mathrm{I}}$ = 41.71).

DeepLabV3+-ResNet34: lindau_000034_000019_leftImg8bit (IoU$_i^{\mathrm{I}}$ = 41.34).

DeepLabV3+-ResNet18: lindau_000034_000019_leftImg8bit (IoU$_i^{\mathrm{I}}$ = 43.90).

DeepLabV3+-EfficientNetB0: lindau_000045_000019_leftImg8bit (IoU$_i^{\mathrm{I}}$ = 42.24).

Figure 7: Worst-case images: Cityscapes 1 / 3. Left: image. Middle: label. Right: prediction.

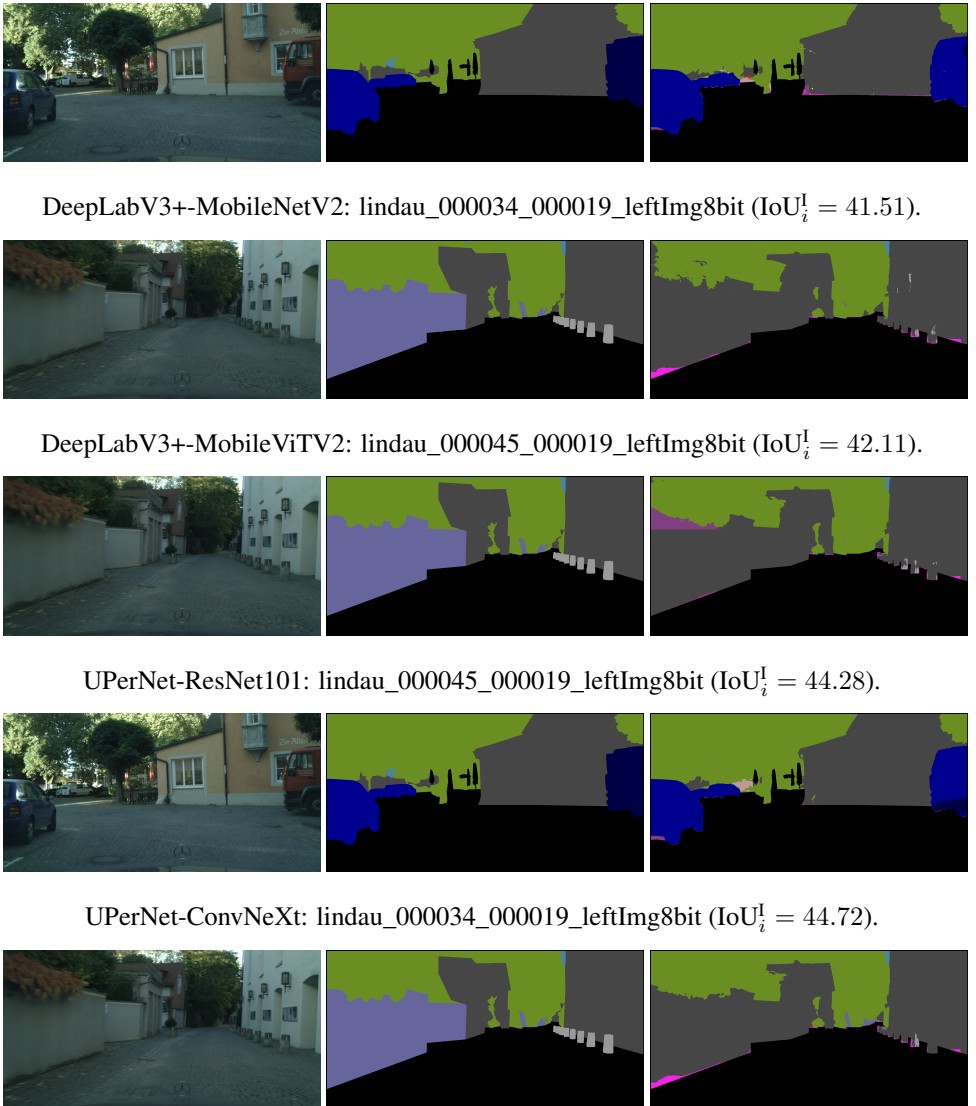

DeepLabV3+-MobileNetV2: lindau_000034_000019_leftImg8bit ($\text{IoU}_i^{\text{I}} = 41.51$).

DeepLabV3+-MobileViTV2: lindau_000045_000019_leftImg8bit ($\text{IoU}_i^{\text{I}} = 42.11$).

UPerNet-ResNet101: lindau_000045_000019_leftImg8bit ($\text{IoU}_i^{\text{I}} = 44.28$).

UPerNet-ConvNeXt: lindau_000034_000019_leftImg8bit ($\text{IoU}_i^{\text{I}} = 44.72$).

UPerNet-MiTB4: lindau_000045_000019_leftImg8bit ($\text{IoU}_i^{\text{I}} = 44.25$).

Figure 8: Worst-case images: Cityscapes 2 / 3. Left: image. Middle: label. Right: prediction.

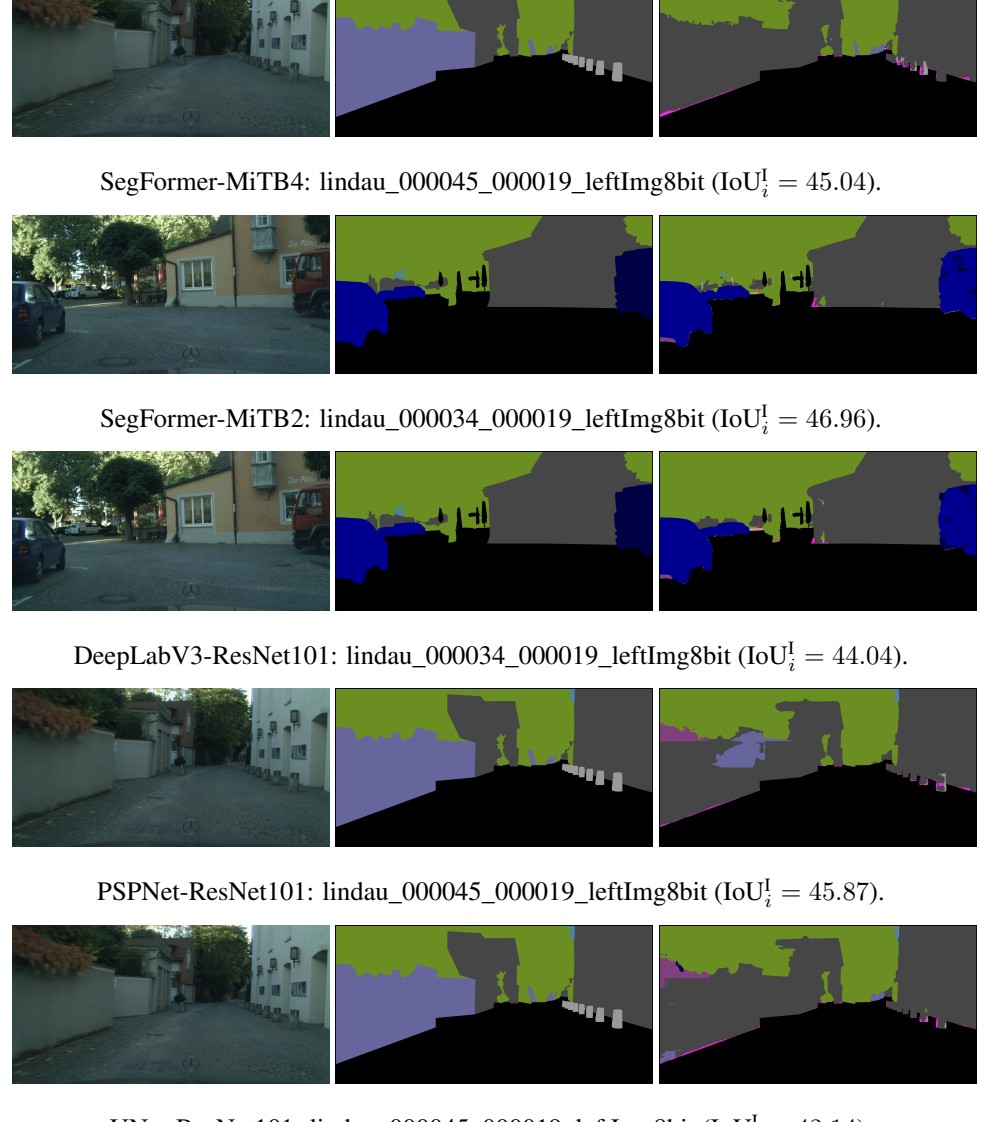

SegFormer-MiTB4: lindau_000045_000019_leftImg8bit ($IoU_i^I = 45.04$).

SegFormer-MiTB2: lindau_000034_000019_leftImg8bit ($IoU_i^I = 46.96$).

DeepLabV3-ResNet101: lindau_000034_000019_leftImg8bit ($IoU_i^I = 44.04$).

PSPNet-ResNet101: lindau_000045_000019_leftImg8bit ($IoU_i^I = 45.87$).

UNet-ResNet101: lindau_000045_000019_leftImg8bit ($IoU_i^I = 42.14$).

Figure 9: Worst-case images: Cityscapes 3 / 3. Left: image. Middle: label. Right: prediction.

## G.2  Nighttime Driving

Table 15: Results: Nighttime Driving. Red: the best for each metric. Green: the worst for each metric.

| Method | Backbone | $mIoU^{I}$ 
 $mIoU^{C}$ 
 $mIoU^{D}$ | $mIoU^{I^{\bar{q}}}$ 
 $mIoU^{C^{\bar{q}}}$ 
 mAcc | $mIoU^{I5}$ 
 $mIoU^{C5}$ 
 Acc | $mIoU^{I1}$ 
 $mIoU^{C1}$ 
 |
|---|---|---|---|---|---|
| DeepLabV3+ | ResNet101 | $43.02 \pm 2.28$ 
 $33.15 \pm 3.84$ 
 $25.93 \pm 1.68$ | $31.63 \pm 2.07$ 
 $17.61 \pm 2.88$ 
 $49.29 \pm 2.43$ | $13.68 \pm 3.54$ 
 $5.08 \pm 2.03$ 
 $69.50 \pm 0.80$ | $10.89 \pm 4.59$ 
 $4.33 \pm 1.70$ 
 |
| DeepLabV3+ | ResNet50 | $38.46 \pm 0.99$ 
 $29.18 \pm 0.86$ 
 $24.28 \pm 0.90$ | $28.14 \pm 1.77$ 
 $13.89 \pm 0.53$ 
 $45.69 \pm 2.02$ | $14.04 \pm 3.20$ 
 $2.48 \pm 1.02$ 
 $58.49 \pm 2.42$ | $13.38 \pm 3.21$ 
 $2.00 \pm 1.41$ 
 |
| DeepLabV3+ | ResNet34 | $28.85 \pm 1.17$ 
 $20.49 \pm 2.05$ 
 $14.35 \pm 0.88$ | $18.30 \pm 0.30$ 
 $8.24 \pm 0.98$ 
 $37.58 \pm 3.42$ | $4.97 \pm 1.04$ 
 $1.02 \pm 0.84$ 
 $53.31 \pm 4.11$ | $3.81 \pm 1.03$ 
 $0.67 \pm 0.94$ 
 |
| DeepLabV3+ | ResNet18 | $29.66 \pm 0.96$ 
 $19.27 \pm 1.44$ 
 $14.20 \pm 0.15$ | $20.25 \pm 0.23$ 
 $8.33 \pm 0.56$ 
 $35.49 \pm 2.45$ | $7.22 \pm 0.99$ 
 $0.82 \pm 0.32$ 
 $57.58 \pm 2.08$ | $5.36 \pm 0.38$ 
 $0.33 \pm 0.47$ 
 |
| DeepLabV3+ | EfficientNetB0 | $34.06 \pm 2.34$ 
 $23.66 \pm 1.42$ 
 $19.45 \pm 1.79$ | $23.63 \pm 2.02$ 
 $10.04 \pm 1.22$ 
 $40.71 \pm 1.76$ | $7.55 \pm 2.75$ 
 $0.69 \pm 0.40$ 
 $61.71 \pm 4.52$ | $5.71 \pm 3.39$ 
 $0.00 \pm 0.00$ 
 |
| DeepLabV3+ | MobileNetV2 | $20.10 \pm 1.19$ 
 $14.96 \pm 1.44$ 
 $12.67 \pm 1.48$ | $10.23 \pm 1.29$ 
 $4.78 \pm 0.69$ 
 $31.86 \pm 3.21$ | $1.11 \pm 0.71$ 
 $0.04 \pm 0.05$ 
 $33.22 \pm 3.82$ | $0.68 \pm 0.38$ 
 $0.00 \pm 0.00$ 
 |
| DeepLabV3+ | MobileViTV2 | $43.04 \pm 1.12$ 
 $31.88 \pm 1.38$ 
 $25.88 \pm 0.81$ | $32.93 \pm 1.37$ 
 $16.49 \pm 0.82$ 
 $43.99 \pm 1.50$ | $17.47 \pm 2.35$ 
 $4.32 \pm 1.03$ 
 $71.18 \pm 1.90$ | $17.00 \pm 2.73$ 
 $3.67 \pm 1.25$ 
 |
| UPerNet | ResNet101 | $39.39 \pm 4.58$ 
 $28.96 \pm 2.15$ 
 $24.24 \pm 1.76$ | $29.09 \pm 4.26$ 
 $14.08 \pm 1.63$ 
 $47.37 \pm 1.62$ | $14.49 \pm 2.73$ 
 $1.99 \pm 1.11$ 
 $63.55 \pm 5.89$ | $13.20 \pm 2.86$ 
 $1.00 \pm 0.82$ 
 |
| UPerNet | ConvNeXt | $62.54 \pm 1.21$ 
 $49.58 \pm 0.99$ 
 $42.10 \pm 0.21$ | $53.40 \pm 0.94$ 
 $33.40 \pm 1.20$ 
 $63.38 \pm 0.66$ | $42.33 \pm 0.68$ 
 $11.94 \pm 1.42$ 
 $82.65 \pm 0.33$ | $41.86 \pm 0.49$ 
 $11.33 \pm 1.25$ 
 |
| UPerNet | MiTB4 | $61.27 \pm 1.15$ 
 $49.89 \pm 1.02$ 
 $41.59 \pm 0.85$ | $52.21 \pm 1.16$ 
 $33.23 \pm 0.83$ 
 $63.42 \pm 1.22$ | $39.71 \pm 1.16$ 
 $13.09 \pm 0.67$ 
 $83.34 \pm 0.60$ | $39.10 \pm 1.17$ 
 $12.33 \pm 0.94$ 
 |
| SegFormer | MiTB4 | $61.45 \pm 1.53$ 
 $50.12 \pm 2.94$ 
 $42.23 \pm 1.66$ | $51.21 \pm 1.76$ 
 $33.41 \pm 2.47$ 
 $62.04 \pm 2.92$ | $37.90 \pm 3.07$ 
 $12.77 \pm 1.48$ 
 $82.95 \pm 1.14$ | $36.07 \pm 3.43$ 
 $12.00 \pm 1.63$ 
 |
| SegFormer | MiTB2 | $50.89 \pm 2.45$ 
 $41.33 \pm 2.03$ 
 $31.40 \pm 2.73$ | $40.91 \pm 2.29$ 
 $25.21 \pm 1.32$ 
 $51.54 \pm 2.54$ | $24.63 \pm 1.78$ 
 $9.61 \pm 0.16$ 
 $79.19 \pm 1.00$ | $23.47 \pm 1.18$ 
 $9.00 \pm 0.00$ 
 |
| DeepLabV3 | ResNet101 | $42.21 \pm 3.33$ 
 $30.71 \pm 2.58$ 
 $26.84 \pm 2.30$ | $31.12 \pm 3.79$ 
 $14.88 \pm 2.27$ 
 $46.24 \pm 2.53$ | $16.23 \pm 3.87$ 
 $2.46 \pm 1.22$ 
 $68.97 \pm 6.15$ | $15.68 \pm 3.54$ 
 $1.67 \pm 1.25$ 
 |
| PSPNet | ResNet101 | $40.48 \pm 1.86$ 
 $30.22 \pm 1.98$ 
 $23.65 \pm 1.17$ | $29.59 \pm 1.97$ 
 $14.81 \pm 2.10$ 
 $45.63 \pm 2.74$ | $13.87 \pm 0.55$ 
 $2.57 \pm 1.78$ 
 $61.03 \pm 4.07$ | $11.37 \pm 1.37$ 
 $1.67 \pm 1.70$ 
 |
| UNet | ResNet101 | $36.86 \pm 2.73$ 
 $25.72 \pm 2.38$ 
 $21.65 \pm 1.17$ | $25.72 \pm 2.82$ 
 $12.31 \pm 1.70$ 
 $45.33 \pm 2.14$ | $9.31 \pm 3.44$ 
 $2.56 \pm 1.36$ 
 $65.95 \pm 3.74$ | $8.20 \pm 3.30$ 
 $1.67 \pm 1.25$ 
 |

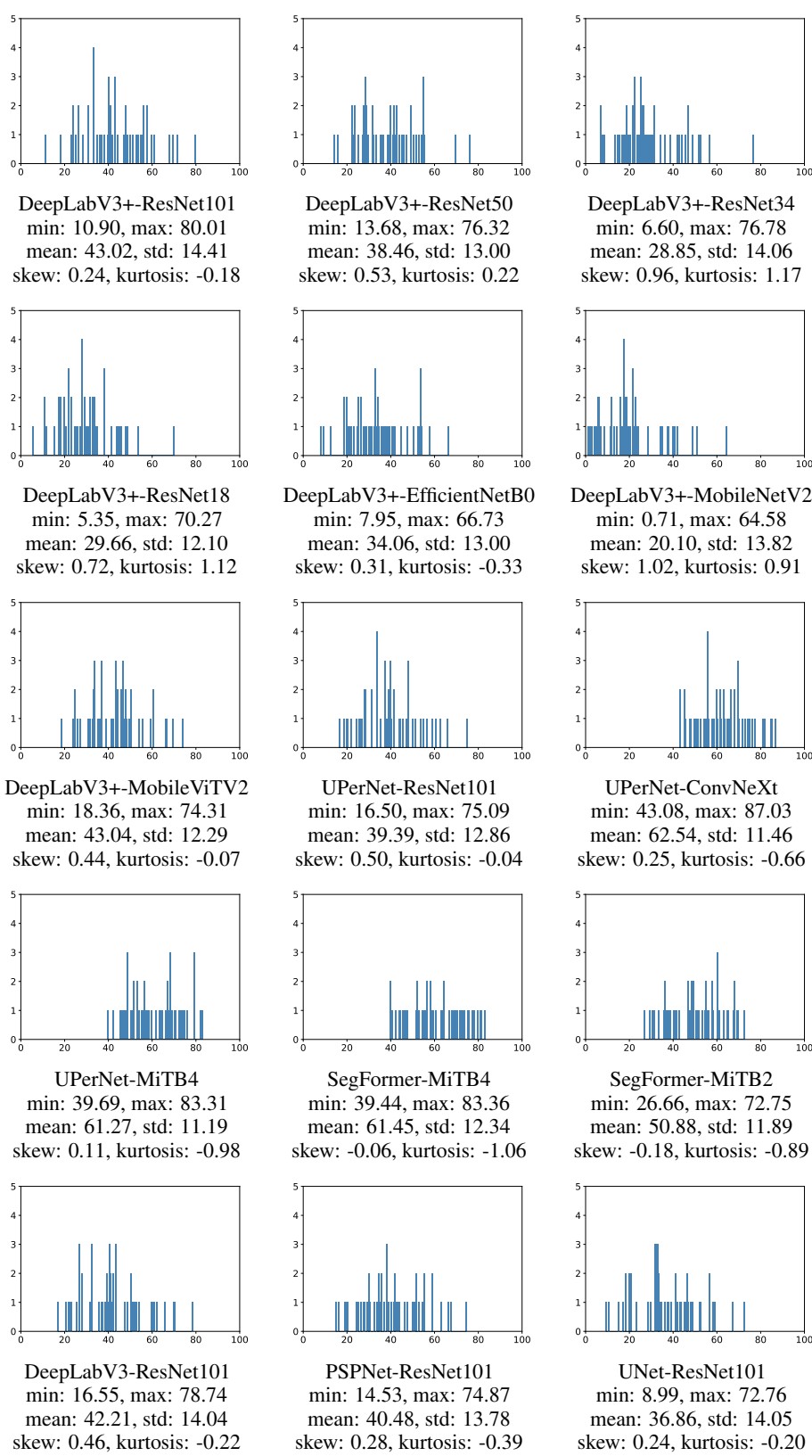

Figure 10: Histograms: Nighttime Driving.

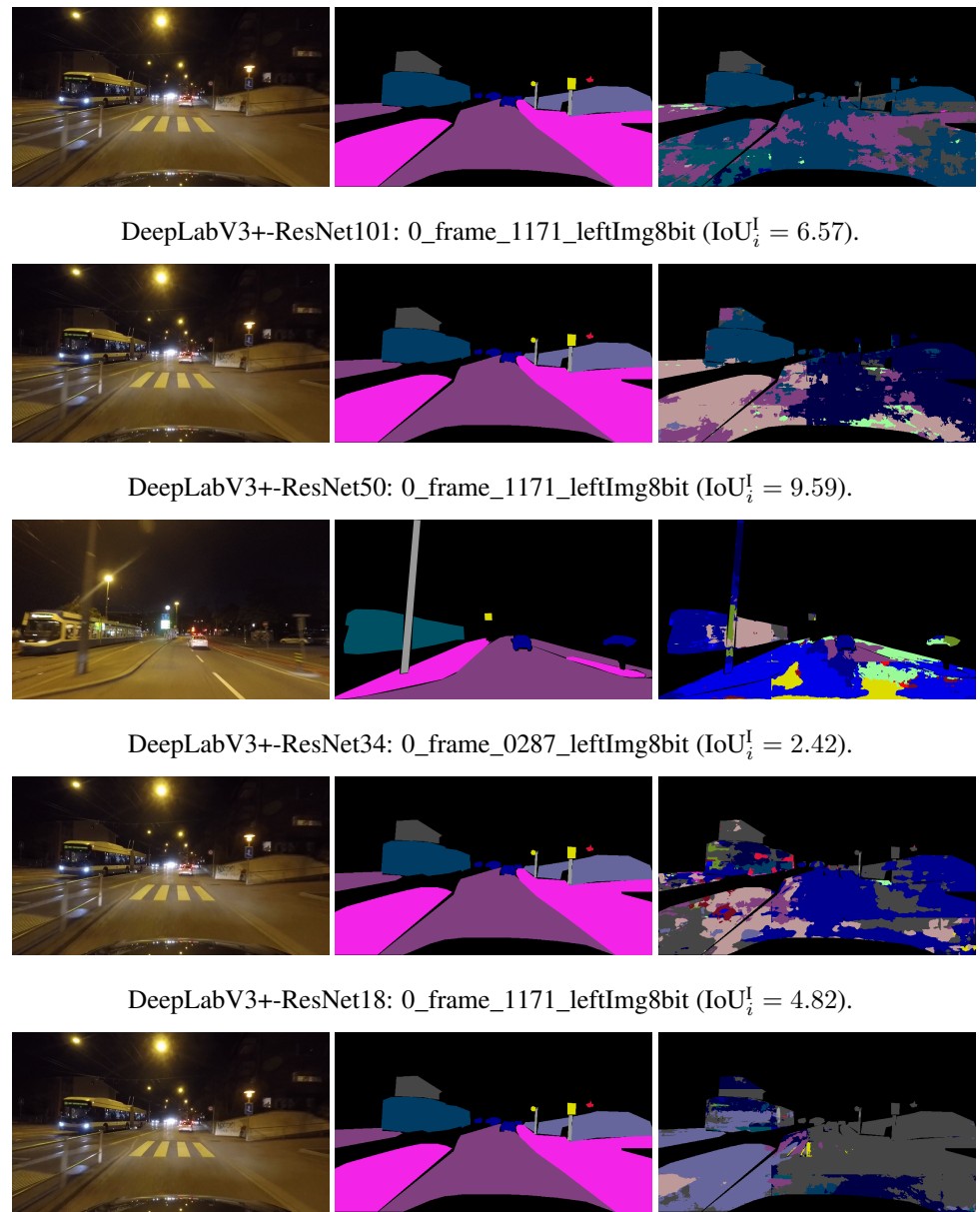

DeepLabV3+-ResNet101: 0_frame_1171_leftImg8bit (IoU$_i^I$ = 6.57).

DeepLabV3+-ResNet50: 0_frame_1171_leftImg8bit (IoU$_i^I$ = 9.59).

DeepLabV3+-ResNet34: 0_frame_0287_leftImg8bit (IoU$_i^I$ = 2.42).

DeepLabV3+-ResNet18: 0_frame_1171_leftImg8bit (IoU$_i^I$ = 4.82).

DeepLabV3+-EfficientNetB0: 0_frame_1171_leftImg8bit (IoU$_i^I$ = 1.53).

Figure 11: Worst-case images: Nighttime Driving 1 / 3. Left: image. Middle: label. Right: prediction.

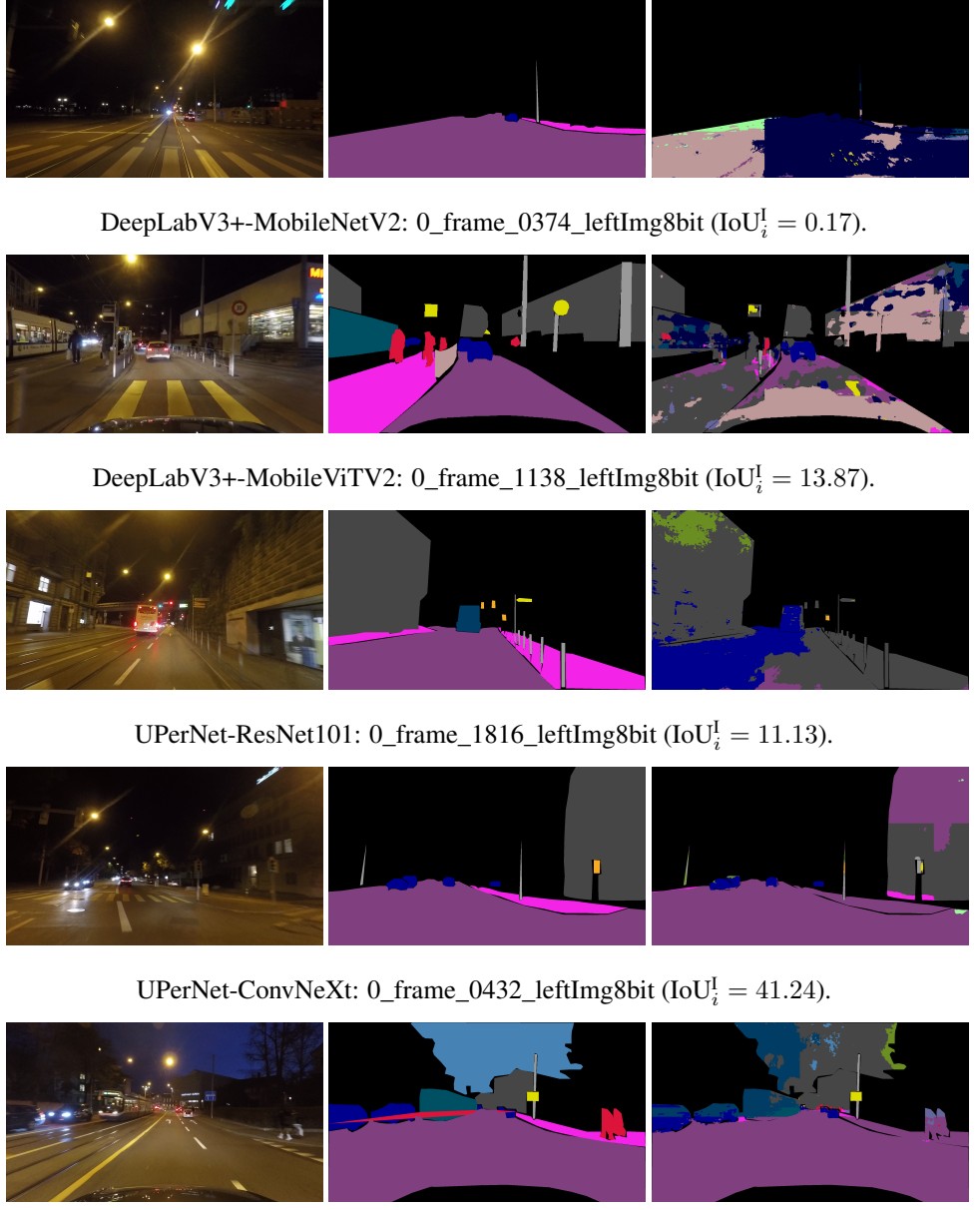

DeepLabV3+-MobileNetV2: 0_frame_0374_leftImg8bit ($IoU_i^I = 0.17$).

DeepLabV3+-MobileViTV2: 0_frame_1138_leftImg8bit ($IoU_i^I = 13.87$).

UPerNet-ResNet101: 0_frame_1816_leftImg8bit ($IoU_i^I = 11.13$).

UPerNet-ConvNeXt: 0_frame_0432_leftImg8bit ($IoU_i^I = 41.24$).

UPerNet-MiTB4: 0_frame_2548_leftImg8bit ($IoU_i^I = 37.89$).

Figure 12: Worst-case images: Nighttime Driving 2 / 3. Left: image. Middle: label. Right: prediction.

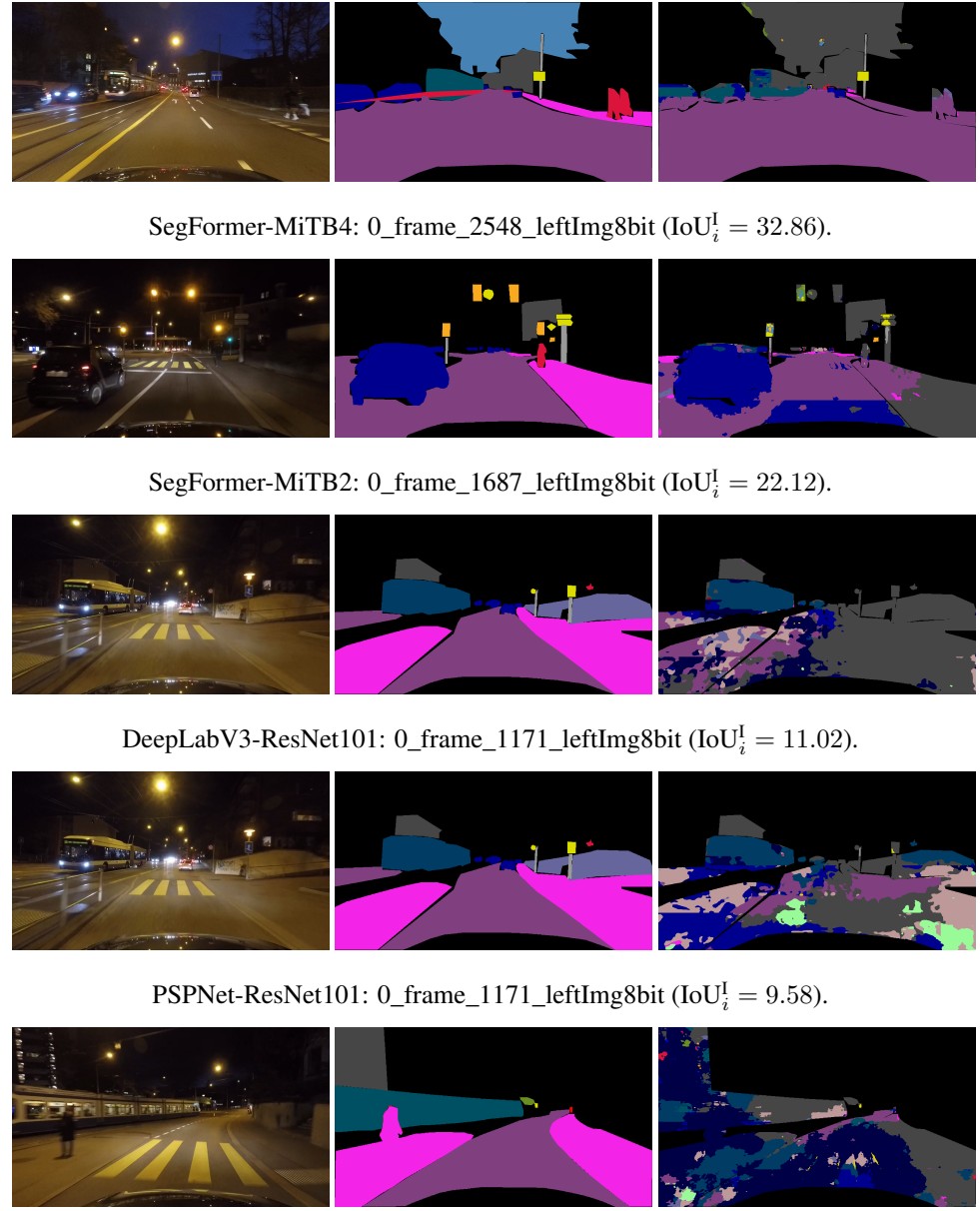

SegFormer-MiTB4: 0_frame_2548_leftImg8bit ($IoU_i^I = 32.86$).

SegFormer-MiTB2: 0_frame_1687_leftImg8bit ($IoU_i^I = 22.12$).

DeepLabV3-ResNet101: 0_frame_1171_leftImg8bit ($IoU_i^I = 11.02$).

PSPNet-ResNet101: 0_frame_1171_leftImg8bit ($IoU_i^I = 9.58$).

UNet-ResNet101: 0_frame_0461_leftImg8bit ($IoU_i^I = 3.58$).

Figure 13: Worst-case images: Nighttime Driving 3 / 3. Left: image. Middle: label. Right: prediction.

## G.3 Dark Zurich

Table 16: Results: Dark Zurich. Red: the best for each metric. Green: the worst for each metric.

| Method | Backbone | $mIoU^I$ / $mIoU^C$ / $mIoU^D$ | $mIoU^{I^{\bar{q}}}$ / $mIoU^{C^{\bar{q}}}$ / mAcc | $mIoU^{I^5}$ / $mIoU^{C^5}$ / Acc | $mIoU^{I^1}$ / $mIoU^{C^1}$ |
|---|---|---|---|---|---|
| DeepLabV3+ | ResNet101 | $21.36 \pm 0.56$ | $16.66 \pm 0.33$ | $8.88 \pm 0.41$ | $8.17 \pm 0.80$ |
| | | $18.41 \pm 0.69$ | $7.83 \pm 0.32$ | $2.07 \pm 0.09$ | $1.33 \pm 0.47$ |
| | | $16.64 \pm 1.25$ | $38.15 \pm 1.03$ | $49.65 \pm 1.22$ | |
| DeepLabV3+ | ResNet50 | $16.08 \pm 0.44$ | $11.66 \pm 0.50$ | $6.03 \pm 0.88$ | $5.77 \pm 0.84$ |
| | | $13.44 \pm 0.52$ | $5.00 \pm 0.21$ | $0.57 \pm 0.10$ | $0.00 \pm 0.00$ |
| | | $12.75 \pm 0.79$ | $33.18 \pm 1.04$ | $33.52 \pm 1.65$ | |
| DeepLabV3+ | ResNet34 | $12.32 \pm 0.13$ | $8.18 \pm 0.15$ | $3.33 \pm 0.57$ | $2.98 \pm 0.64$ |
| | | $9.87 \pm 0.26$ | $3.54 \pm 0.18$ | $0.64 \pm 0.41$ | $0.33 \pm 0.47$ |
| | | $8.42 \pm 0.43$ | $26.33 \pm 1.04$ | $35.17 \pm 2.29$ | |
| DeepLabV3+ | ResNet18 | $11.28 \pm 0.62$ | $7.86 \pm 0.39$ | $4.19 \pm 0.31$ | $3.90 \pm 0.57$ |
| | | $9.43 \pm 0.50$ | $3.54 \pm 0.16$ | $0.92 \pm 0.28$ | $0.67 \pm 0.47$ |
| | | $8.38 \pm 0.48$ | $26.04 \pm 0.50$ | $33.48 \pm 2.66$ | |
| DeepLabV3+ | EfficientNetB0 | $16.86 \pm 0.65$ | $12.48 \pm 0.42$ | $6.37 \pm 0.83$ | $5.57 \pm 1.31$ |
| | | $13.42 \pm 0.75$ | $5.49 \pm 0.23$ | $1.08 \pm 0.64$ | $0.33 \pm 0.47$ |
| | | $11.21 \pm 0.79$ | $29.31 \pm 1.74$ | $38.19 \pm 1.70$ | |
| DeepLabV3+ | MobileNetV2 | $9.49 \pm 0.83$ | $5.98 \pm 0.82$ | $1.52 \pm 0.65$ | $1.27 \pm 0.53$ |
| | | $6.95 \pm 0.71$ | $2.37 \pm 0.24$ | $0.12 \pm 0.10$ | $0.00 \pm 0.00$ |
| | | $6.54 \pm 0.93$ | $24.14 \pm 1.85$ | $25.17 \pm 4.32$ | |
| DeepLabV3+ | MobileViTV2 | $17.02 \pm 1.99$ | $12.96 \pm 2.19$ | $6.75 \pm 2.42$ | $5.85 \pm 2.18$ |
| | | $14.62 \pm 1.55$ | $6.12 \pm 0.92$ | $1.31 \pm 0.65$ | $0.67 \pm 0.47$ |
| | | $11.51 \pm 1.55$ | $29.13 \pm 2.03$ | $41.76 \pm 5.32$ | |
| UPerNet | ResNet101 | $17.29 \pm 1.14$ | $13.22 \pm 1.19$ | $6.87 \pm 1.83$ | $6.52 \pm 2.13$ |
| | | $15.38 \pm 0.82$ | $6.10 \pm 0.59$ | $1.17 \pm 0.32$ | $0.33 \pm 0.47$ |
| | | $13.39 \pm 0.24$ | $35.64 \pm 1.18$ | $41.75 \pm 2.27$ | |
| UPerNet | ConvNeXt | $26.03 \pm 1.26$ | $20.58 \pm 1.44$ | $12.90 \pm 1.78$ | $12.23 \pm 1.50$ |
| | | $23.19 \pm 0.69$ | $10.05 \pm 0.60$ | $1.73 \pm 0.09$ | $1.00 \pm 0.00$ |
| | | $21.98 \pm 1.34$ | $44.15 \pm 0.73$ | $51.71 \pm 1.93$ | |
| UPerNet | MiTB4 | $21.76 \pm 1.77$ | $16.64 \pm 1.66$ | $8.17 \pm 1.26$ | $7.27 \pm 1.24$ |
| | | $18.56 \pm 1.22$ | $8.01 \pm 0.65$ | $1.74 \pm 0.21$ | $1.00 \pm 0.00$ |
| | | $16.25 \pm 1.05$ | $39.52 \pm 0.93$ | $47.32 \pm 1.67$ | |
| SegFormer | MiTB4 | $24.33 \pm 1.17$ | $18.68 \pm 1.08$ | $8.59 \pm 0.62$ | $6.95 \pm 0.65$ |
| | | $21.01 \pm 0.88$ | $9.35 \pm 0.66$ | $1.92 \pm 0.21$ | $1.00 \pm 0.00$ |
| | | $18.98 \pm 1.25$ | $40.31 \pm 1.35$ | $50.83 \pm 1.04$ | |
| SegFormer | MiTB2 | $23.75 \pm 0.49$ | $18.56 \pm 0.49$ | $9.73 \pm 0.16$ | $8.30 \pm 0.80$ |
| | | $19.85 \pm 0.31$ | $8.44 \pm 0.48$ | $1.91 \pm 0.41$ | $1.33 \pm 0.47$ |
| | | $17.90 \pm 0.08$ | $36.58 \pm 0.59$ | $50.28 \pm 0.52$ | |
| DeepLabV3 | ResNet101 | $19.56 \pm 1.60$ | $14.27 \pm 1.72$ | $6.55 \pm 2.40$ | $5.83 \pm 2.21$ |
| | | $17.35 \pm 1.63$ | $6.68 \pm 0.93$ | $1.49 \pm 0.74$ | $1.00 \pm 0.82$ |
| | | $15.38 \pm 1.22$ | $35.65 \pm 2.53$ | $46.86 \pm 3.32$ | |
| PSPNet | ResNet101 | $19.80 \pm 1.59$ | $15.14 \pm 1.57$ | $7.38 \pm 2.11$ | $6.62 \pm 2.21$ |
| | | $17.09 \pm 1.32$ | $6.70 \pm 0.75$ | $1.21 \pm 0.39$ | $0.67 \pm 0.47$ |
| | | $15.61 \pm 0.78$ | $36.83 \pm 2.12$ | $45.22 \pm 3.65$ | |
| UNet | ResNet101 | $16.59 \pm 1.06$ | $12.34 \pm 0.86$ | $5.57 \pm 0.39$ | $5.23 \pm 0.28$ |
| | | $14.34 \pm 1.15$ | $6.02 \pm 0.40$ | $2.23 \pm 0.12$ | $2.00 \pm 0.00$ |
| | | $13.21 \pm 0.56$ | $34.30 \pm 1.80$ | $45.02 \pm 1.27$ | |

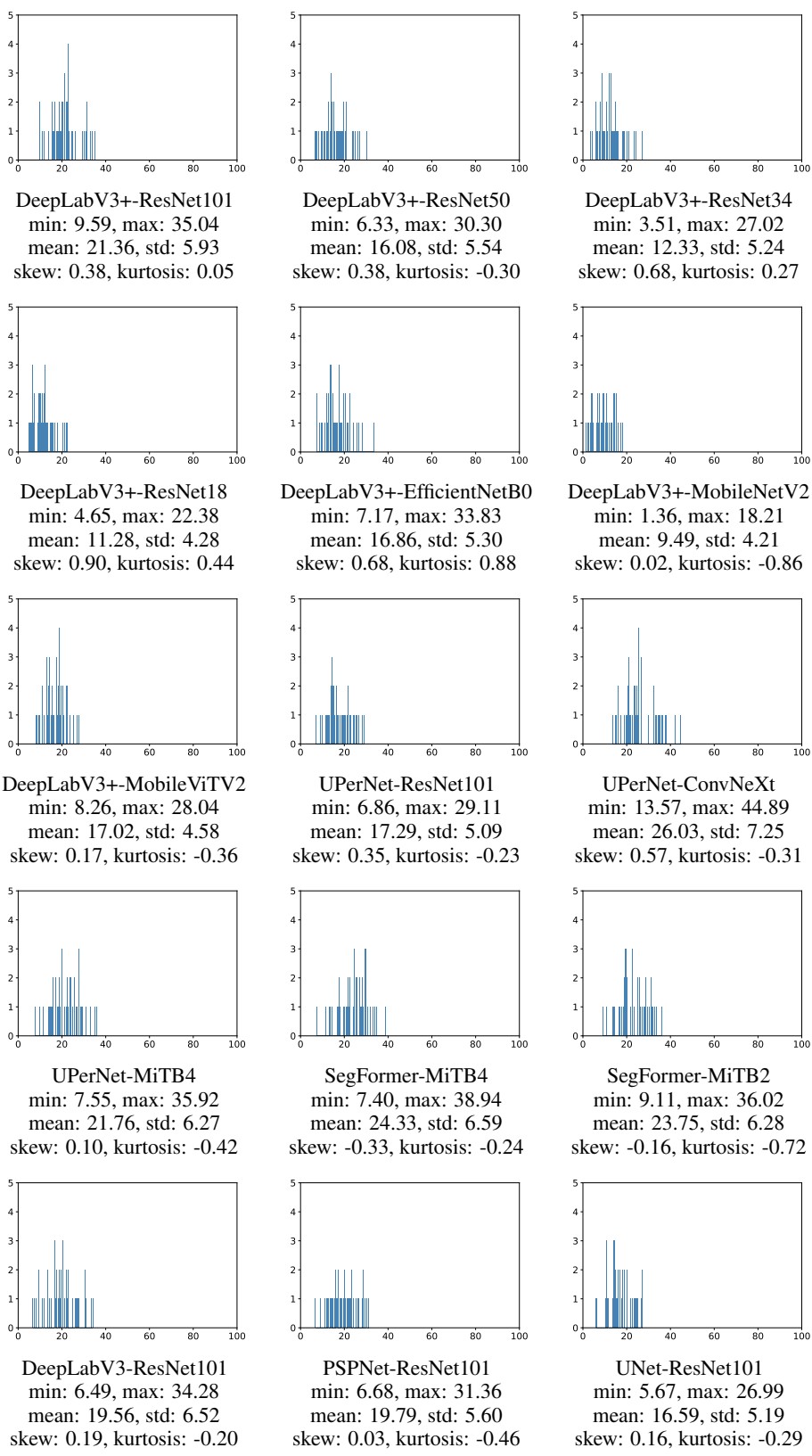

Figure 14: Histograms: Dark Zurich.

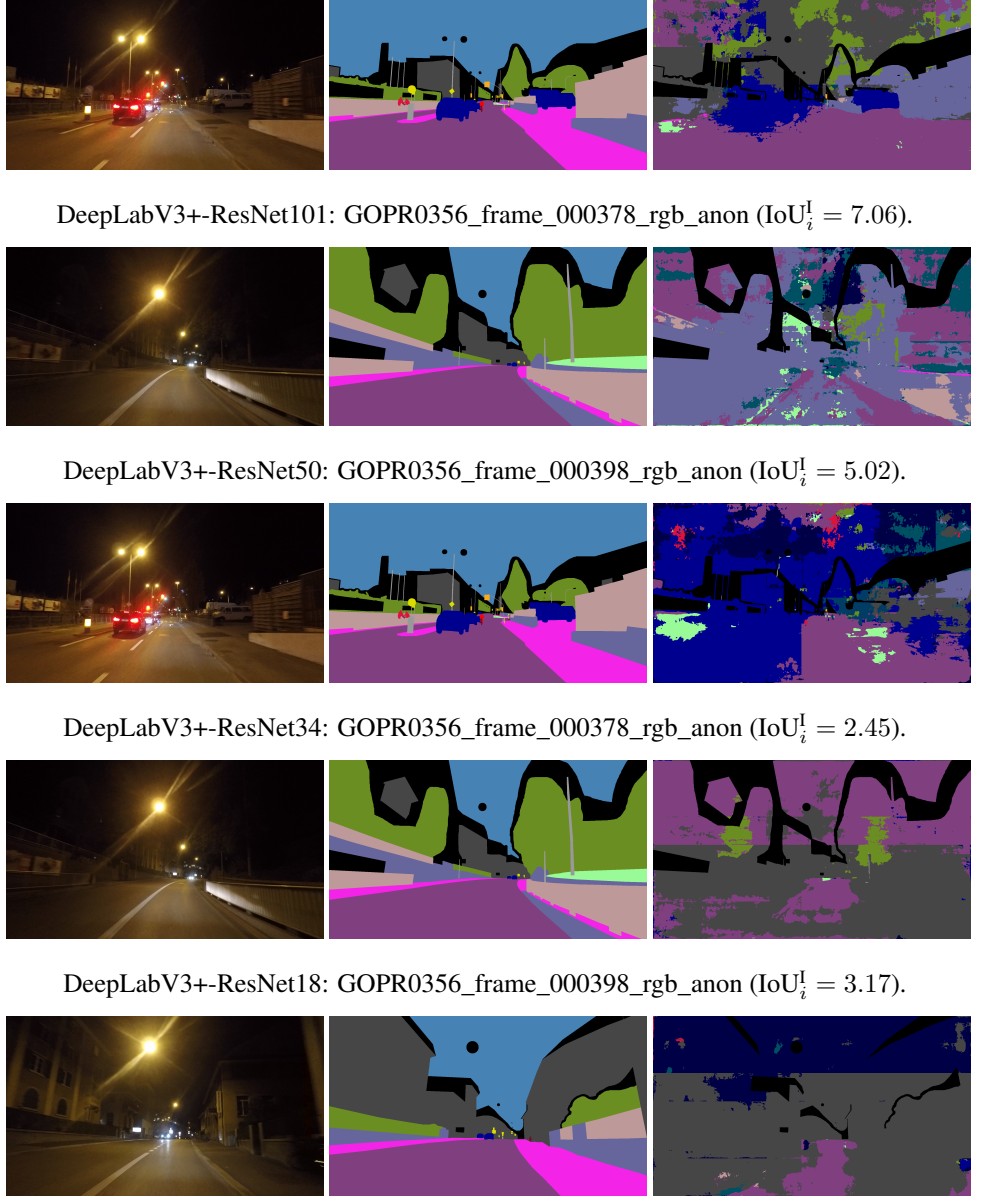

DeepLabV3+-ResNet101: GOPR0356_frame_000378_rgb_anon ($IoU_i^I = 7.06$).

DeepLabV3+-ResNet50: GOPR0356_frame_000398_rgb_anon ($IoU_i^I = 5.02$).

DeepLabV3+-ResNet34: GOPR0356_frame_000378_rgb_anon ($IoU_i^I = 2.45$).

DeepLabV3+-ResNet18: GOPR0356_frame_000398_rgb_anon ($IoU_i^I = 3.17$).

DeepLabV3+-EfficientNetB0: GOPR0356_frame_000401_rgb_anon ($IoU_i^I = 4.38$).

Figure 15: Worst-case images: Dark Zurich 1 / 3. Left: image. Middle: label. Right: prediction.

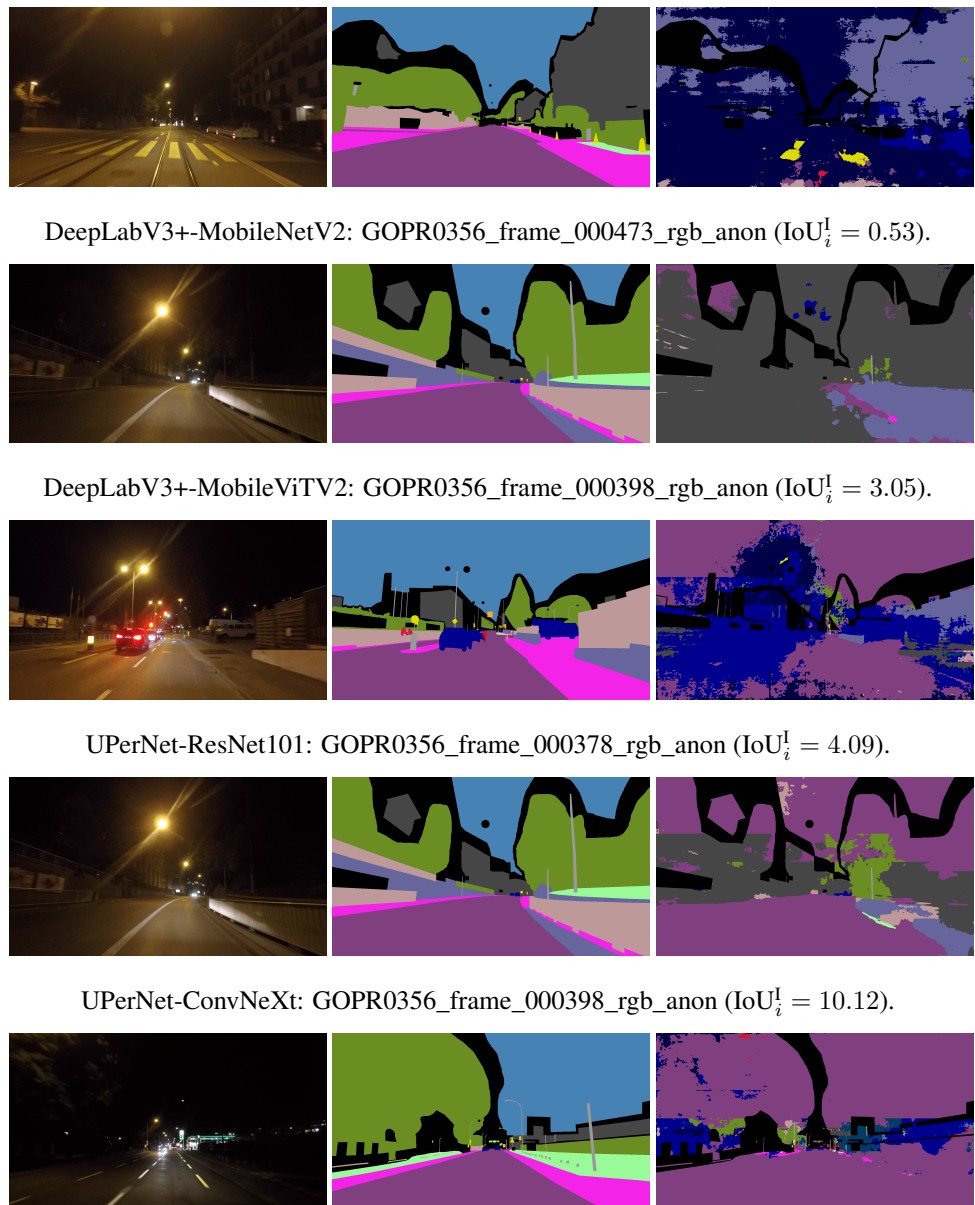

DeepLabV3+-MobileNetV2: GOPR0356_frame_000473_rgb_anon ($\text{IoU}_i^{\text{I}} = 0.53$).

DeepLabV3+-MobileViTV2: GOPR0356_frame_000398_rgb_anon ($\text{IoU}_i^{\text{I}} = 3.05$).

UPerNet-ResNet101: GOPR0356_frame_000378_rgb_anon ($\text{IoU}_i^{\text{I}} = 4.09$).

UPerNet-ConvNeXt: GOPR0356_frame_000398_rgb_anon ($\text{IoU}_i^{\text{I}} = 10.12$).

UPerNet-MiTB4: GOPR0356_frame_000333_rgb_anon ($\text{IoU}_i^{\text{I}} = 5.96$).

Figure 16: Worst-case images: Dark Zurich 2 / 3. Left: image. Middle: label. Right: prediction.

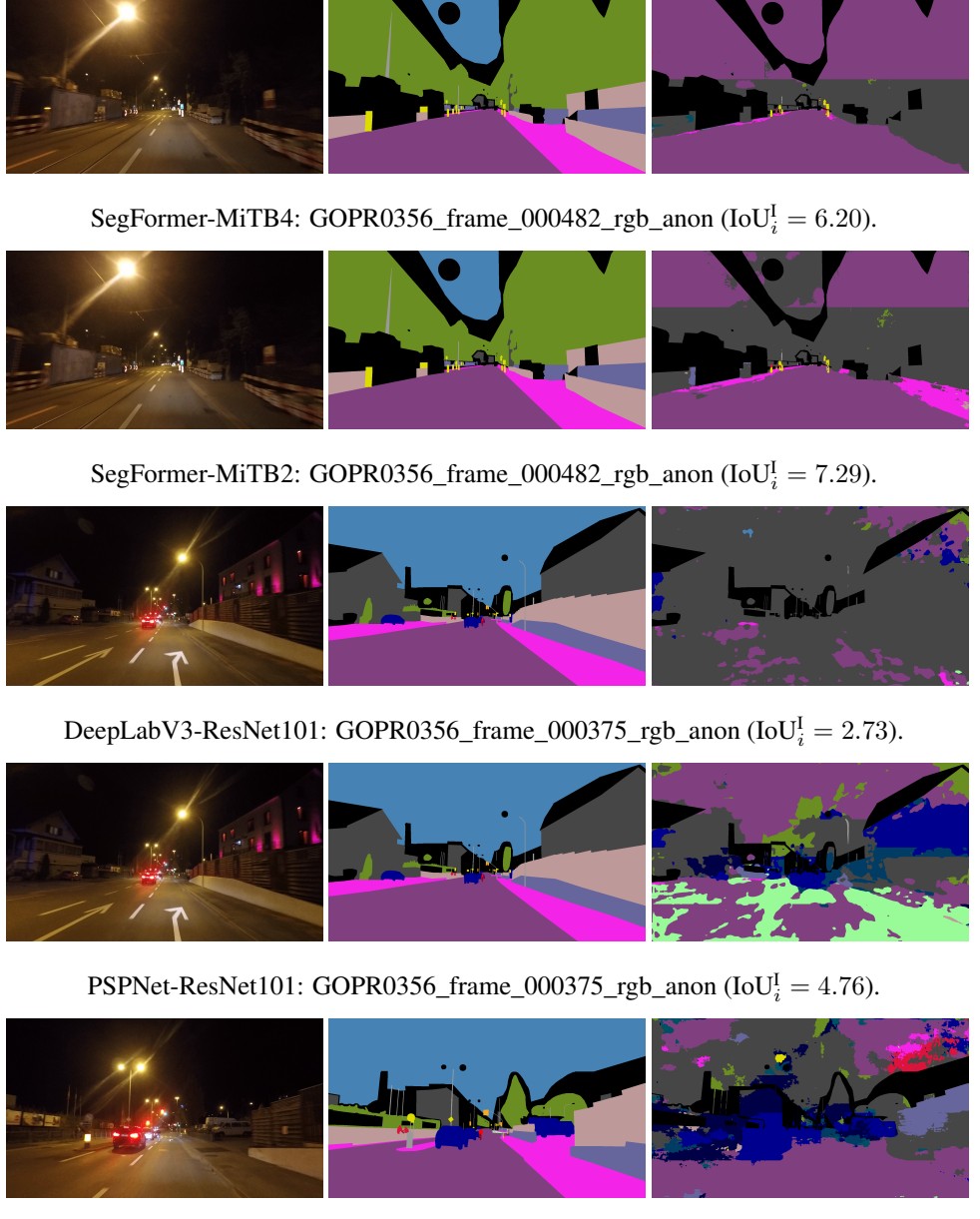

SegFormer-MiTB4: GOPR0356_frame_000482_rgb_anon ($IoU_i^I = 6.20$).

SegFormer-MiTB2: GOPR0356_frame_000482_rgb_anon ($IoU_i^I = 7.29$).

DeepLabV3-ResNet101: GOPR0356_frame_000375_rgb_anon ($IoU_i^I = 2.73$).

PSPNet-ResNet101: GOPR0356_frame_000375_rgb_anon ($IoU_i^I = 4.76$).

UNet-ResNet101: GOPR0356_frame_000378_rgb_anon ($IoU_i^I = 4.90$).

Figure 17: Worst-case images: Dark Zurich 3 / 3. Left: image. Middle: label. Right: prediction.

## G.4 Mapillary Vistas

Table 17: Results: Mapillary Vistas. Red: the best for each metric. Green: the worst for each metric.

| Method | Backbone | $\text{mIoU}^{\text{I}}$ / $\text{mIoU}^{\text{C}}$ / $\text{mIoU}^{\text{K}}$ / $\text{mIoU}^{\text{D}}$ | $\text{mIoU}^{\text{I}^{\bar{q}}}$ / $\text{mIoU}^{\text{C}^{\bar{q}}}$ / $\text{mIoU}^{\text{K}^{\bar{q}}}$ / mAcc | $\text{mIoU}^{\text{I}^{5}}$ / $\text{mIoU}^{\text{C}^{5}}$ / $\text{mIoU}^{\text{K}^{5}}$ / Acc | $\text{mIoU}^{\text{I}^{1}}$ / $\text{mIoU}^{\text{C}^{1}}$ / $\text{mIoU}^{\text{K}^{1}}$ |
|---|---|---|---|---|---|
| DeepLabV3+ | ResNet101 | $62.47 \pm 0.04$ | $56.01 \pm 0.04$ | $44.61 \pm 0.15$ | $38.47 \pm 0.21$ |
| | | $46.23 \pm 0.12$ | $27.17 \pm 0.07$ | $3.06 \pm 0.02$ | $0.00 \pm 0.00$ |
| | | $45.25 \pm 0.07$ | $25.12 \pm 0.04$ | $2.54 \pm 0.01$ | $0.00 \pm 0.00$ |
| | | $47.76 \pm 0.22$ | $60.20 \pm 0.27$ | $91.36 \pm 0.03$ | |
| DeepLabV3+ | ResNet50 | $60.61 \pm 0.03$ | $53.99 \pm 0.05$ | $42.58 \pm 0.12$ | $36.67 \pm 0.12$ |
| | | $43.31 \pm 0.03$ | $25.10 \pm 0.03$ | $2.95 \pm 0.00$ | $0.00 \pm 0.00$ |
| | | $42.52 \pm 0.02$ | $23.22 \pm 0.03$ | $2.48 \pm 0.01$ | $0.00 \pm 0.00$ |
| | | $44.30 \pm 0.13$ | $56.03 \pm 0.14$ | $90.76 \pm 0.03$ | |
| DeepLabV3+ | ResNet34 | $58.35 \pm 0.05$ | $51.47 \pm 0.05$ | $39.78 \pm 0.17$ | $33.99 \pm 0.54$ |
| | | $38.16 \pm 0.05$ | $22.13 \pm 0.07$ | $2.56 \pm 0.02$ | $0.00 \pm 0.00$ |
| | | $37.09 \pm 0.01$ | $20.25 \pm 0.07$ | $2.26 \pm 0.03$ | $0.00 \pm 0.00$ |
| | | $38.74 \pm 0.09$ | $49.59 \pm 0.14$ | $90.22 \pm 0.03$ | |
| DeepLabV3+ | ResNet18 | $56.55 \pm 0.08$ | $49.56 \pm 0.09$ | $37.75 \pm 0.23$ | $31.94 \pm 0.33$ |
| | | $36.00 \pm 0.10$ | $20.72 \pm 0.08$ | $2.41 \pm 0.04$ | $0.00 \pm 0.00$ |
| | | $35.03 \pm 0.08$ | $18.93 \pm 0.06$ | $2.13 \pm 0.03$ | $0.00 \pm 0.00$ |
| | | $36.58 \pm 0.26$ | $47.09 \pm 0.30$ | $89.63 \pm 0.08$ | |
| DeepLabV3+ | EfficientNetB0 | $58.28 \pm 0.06$ | $51.44 \pm 0.09$ | $39.96 \pm 0.12$ | $35.18 \pm 0.37$ |
| | | $38.17 \pm 0.15$ | $22.25 \pm 0.06$ | $2.72 \pm 0.03$ | $0.00 \pm 0.00$ |
| | | $37.11 \pm 0.13$ | $20.38 \pm 0.03$ | $2.37 \pm 0.04$ | $0.00 \pm 0.00$ |
| | | $39.89 \pm 0.39$ | $50.43 \pm 0.34$ | $90.64 \pm 0.06$ | |
| DeepLabV3+ | MobileNetV2 | $55.46 \pm 0.04$ | $48.50 \pm 0.05$ | $36.78 \pm 0.15$ | $31.03 \pm 0.33$ |
| | | $34.97 \pm 0.13$ | $19.87 \pm 0.10$ | $2.29 \pm 0.01$ | $0.00 \pm 0.00$ |
| | | $33.95 \pm 0.11$ | $18.14 \pm 0.09$ | $2.07 \pm 0.02$ | $0.00 \pm 0.00$ |
| | | $35.59 \pm 0.05$ | $45.60 \pm 0.09$ | $89.32 \pm 0.03$ | |
| DeepLabV3+ | MobileViTV2 | $58.28 \pm 0.02$ | $51.47 \pm 0.04$ | $40.05 \pm 0.08$ | $34.97 \pm 0.15$ |
| | | $38.22 \pm 0.03$ | $22.32 \pm 0.02$ | $2.75 \pm 0.02$ | $0.00 \pm 0.00$ |
| | | $37.21 \pm 0.01$ | $20.49 \pm 0.02$ | $2.36 \pm 0.01$ | $0.00 \pm 0.00$ |
| | | $40.40 \pm 0.12$ | $50.41 \pm 0.28$ | $90.57 \pm 0.01$ | |
| UPerNet | ResNet101 | $61.68 \pm 0.08$ | $55.19 \pm 0.06$ | $43.86 \pm 0.12$ | $37.29 \pm 0.22$ |
| | | $46.05 \pm 0.08$ | $26.60 \pm 0.04$ | $3.01 \pm 0.02$ | $0.00 \pm 0.00$ |
| | | $45.15 \pm 0.07$ | $24.66 \pm 0.01$ | $2.47 \pm 0.01$ | $0.00 \pm 0.00$ |
| | | $48.15 \pm 0.14$ | $60.10 \pm 0.34$ | $91.20 \pm 0.08$ | |
| UPerNet | ConvNeXt | $65.26 \pm 0.01$ | $59.12 \pm 0.03$ | $48.55 \pm 0.07$ | $42.90 \pm 0.26$ |
| | | $51.95 \pm 0.04$ | $30.91 \pm 0.06$ | $3.53 \pm 0.02$ | $0.00 \pm 0.00$ |
| | | $50.88 \pm 0.02$ | $28.68 \pm 0.02$ | $2.90 \pm 0.01$ | $0.00 \pm 0.00$ |
| | | $55.68 \pm 0.12$ | $67.63 \pm 0.14$ | $92.41 \pm 0.04$ | |
| UPerNet | MiTB4 | $64.48 \pm 0.01$ | $58.29 \pm 0.02$ | $47.66 \pm 0.15$ | $42.59 \pm 0.57$ |
| | | $50.08 \pm 0.10$ | $29.69 \pm 0.02$ | $3.48 \pm 0.03$ | $0.33 \pm 0.47$ |
| | | $49.11 \pm 0.10$ | $27.56 \pm 0.01$ | $2.83 \pm 0.04$ | $0.00 \pm 0.00$ |
| | | $54.22 \pm 0.09$ | $66.05 \pm 0.11$ | $92.41 \pm 0.01$ | |
| SegFormer | MiTB4 | $64.17 \pm 0.02$ | $57.97 \pm 0.02$ | $47.43 \pm 0.02$ | $42.50 \pm 0.27$ |
| | | $50.05 \pm 0.05$ | $29.59 \pm 0.02$ | $3.49 \pm 0.08$ | $0.33 \pm 0.47$ |
| | | $49.02 \pm 0.08$ | $27.45 \pm 0.06$ | $2.87 \pm 0.04$ | $0.33 \pm 0.47$ |
| | | $54.10 \pm 0.04$ | $65.96 \pm 0.13$ | $92.38 \pm 0.03$ | |
| SegFormer | MiTB2 | $62.15 \pm 0.03$ | $55.83 \pm 0.01$ | $45.15 \pm 0.12$ | $39.94 \pm 0.14$ |
| | | $46.70 \pm 0.07$ | $27.18 \pm 0.06$ | $3.15 \pm 0.04$ | $0.00 \pm 0.00$ |
| | | $45.75 \pm 0.04$ | $25.18 \pm 0.05$ | $2.63 \pm 0.02$ | $0.00 \pm 0.00$ |
| | | $51.02 \pm 0.13$ | $62.55 \pm 0.24$ | $91.91 \pm 0.03$ | |
| DeepLabV3 | ResNet101 | $62.33 \pm 0.03$ | $55.94 \pm 0.03$ | $44.82 \pm 0.05$ | $38.66 \pm 0.29$ |
| | | $47.13 \pm 0.02$ | $27.44 \pm 0.01$ | $3.06 \pm 0.02$ | $0.00 \pm 0.00$ |
| | | $46.00 \pm 0.08$ | $25.26 \pm 0.04$ | $2.55 \pm 0.01$ | $0.00 \pm 0.00$ |
| | | $49.32 \pm 0.33$ | $61.35 \pm 0.27$ | $91.44 \pm 0.03$ | |
| PSPNet | ResNet101 | $61.85 \pm 0.07$ | $55.46 \pm 0.08$ | $44.41 \pm 0.08$ | $38.12 \pm 0.37$ |
| | | $46.85 \pm 0.18$ | $27.19 \pm 0.14$ | $3.05 \pm 0.03$ | $0.00 \pm 0.00$ |
| | | $45.85 \pm 0.12$ | $25.11 \pm 0.09$ | $2.54 \pm 0.03$ | $0.00 \pm 0.00$ |
| | | $48.74 \pm 0.30$ | $60.43 \pm 0.26$ | $91.27 \pm 0.03$ | |
| UNet | ResNet101 | $57.29 \pm 0.20$ | $50.30 \pm 0.26$ | $38.45 \pm 0.36$ | $31.93 \pm 0.38$ |
| | | $36.03 \pm 0.33$ | $20.81 \pm 0.28$ | $2.36 \pm 0.08$ | $0.00 \pm 0.00$ |
| | | $35.34 \pm 0.32$ | $19.23 \pm 0.27$ | $2.12 \pm 0.05$ | $0.00 \pm 0.00$ |
| | | $33.30 \pm 0.36$ | $42.07 \pm 0.46$ | $87.63 \pm 0.22$ | |

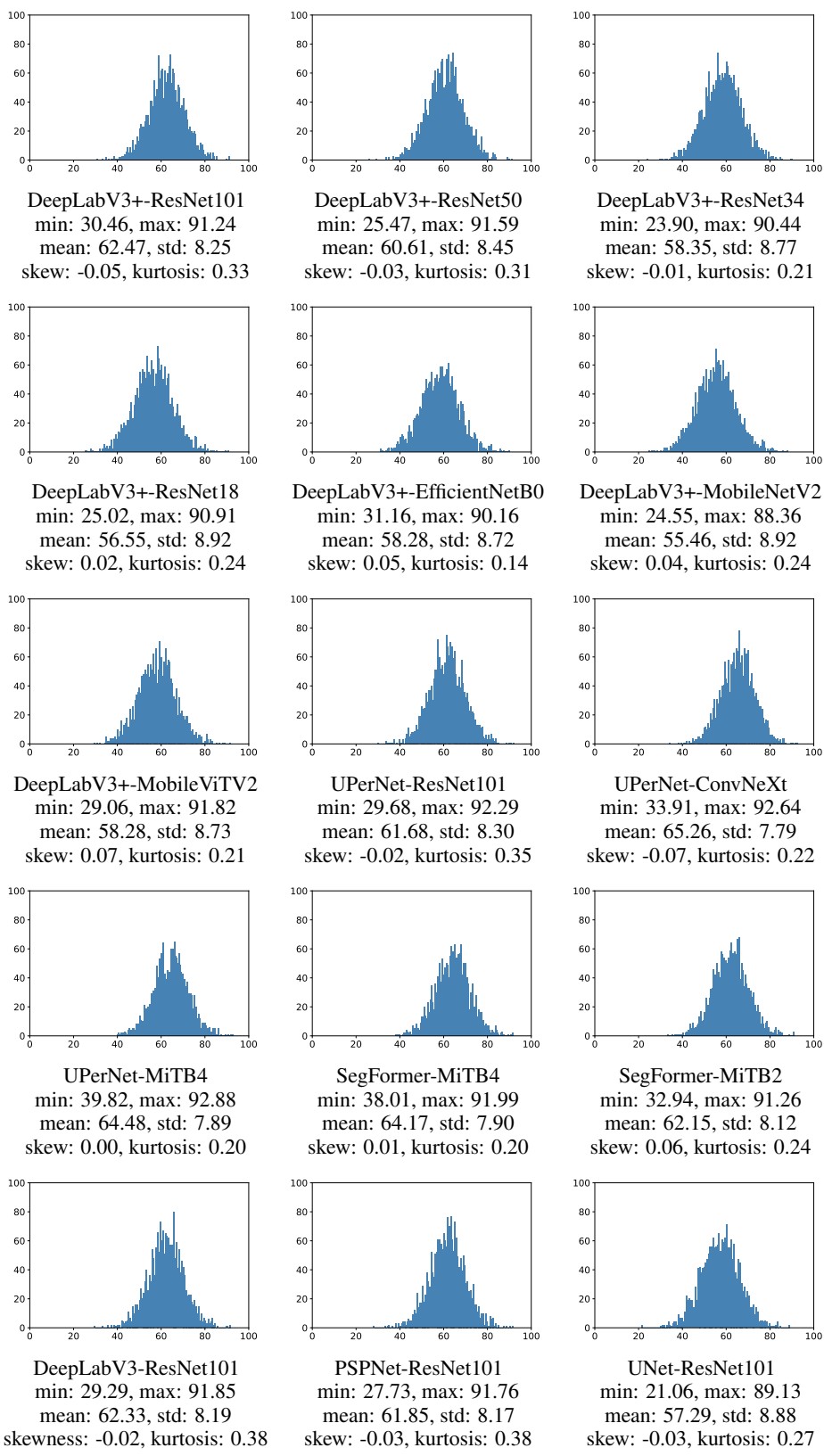

Figure 18: Histograms: Mapillary Vistas.

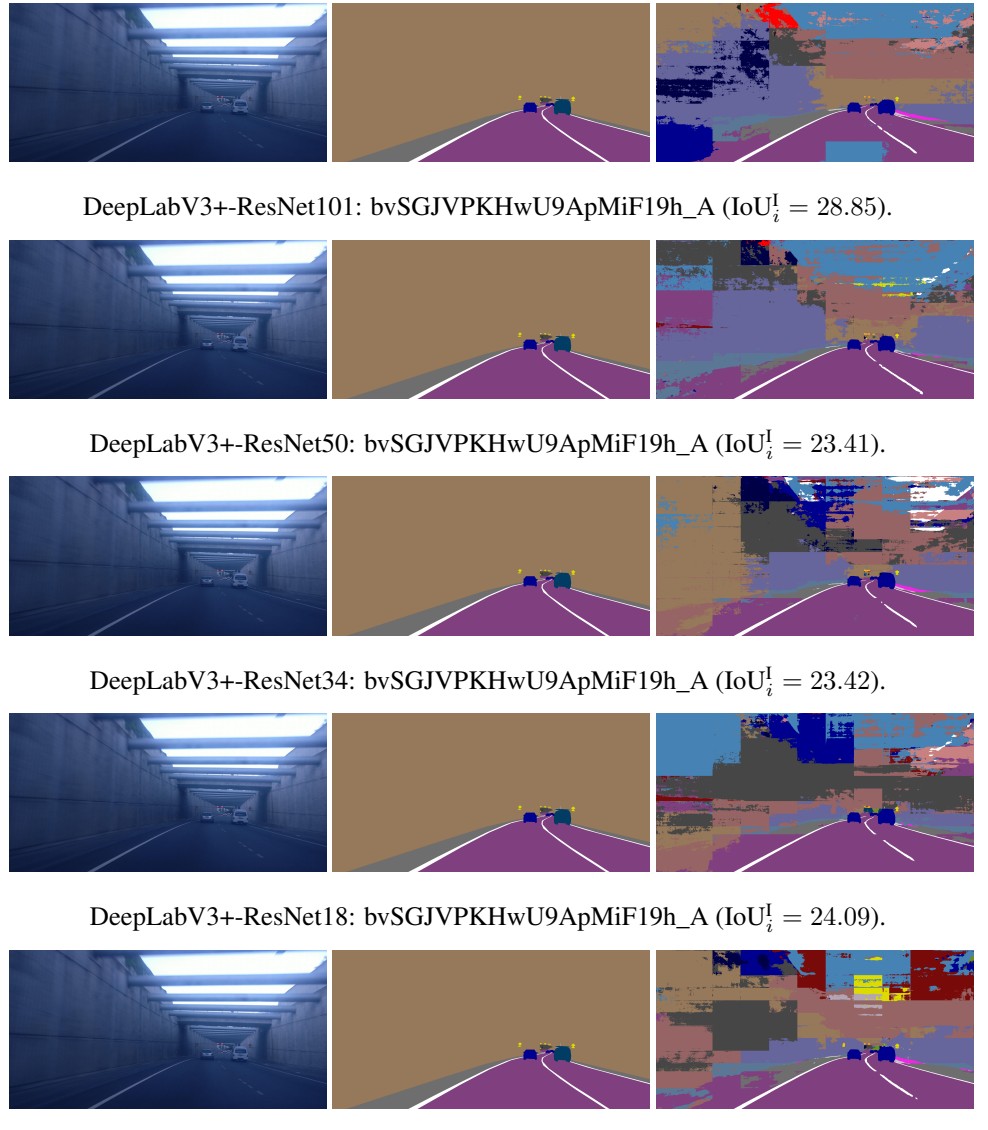

DeepLabV3+-ResNet101: bvSGJVPKHwU9ApMiF19h_A ($IoU_i^I = 28.85$).

DeepLabV3+-ResNet50: bvSGJVPKHwU9ApMiF19h_A ($IoU_i^I = 23.41$).

DeepLabV3+-ResNet34: bvSGJVPKHwU9ApMiF19h_A ($IoU_i^I = 23.42$).

DeepLabV3+-ResNet18: bvSGJVPKHwU9ApMiF19h_A ($IoU_i^I = 24.09$).

DeepLabV3+-EfficientNetB0: bvSGJVPKHwU9ApMiF19h_A ($IoU_i^I = 26.10$).

Figure 19: Worst-case images: Mapillary Vistas 1 / 3. Left: image. Middle: label. Right: prediction.

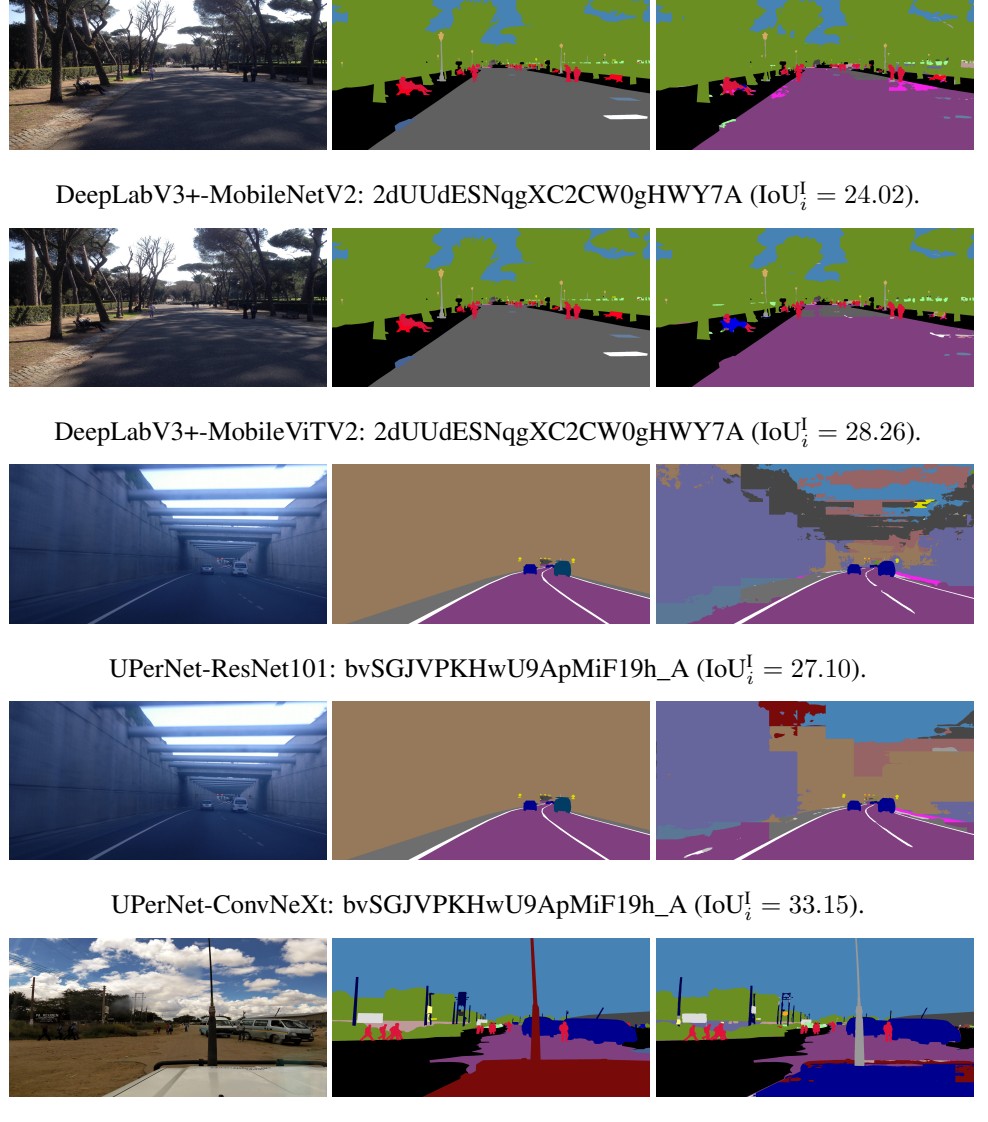

DeepLabV3+-MobileNetV2: 2dUUdESNqgXC2CW0gHWY7A ($\text{IoU}_i^I = 24.02$).

DeepLabV3+-MobileViTV2: 2dUUdESNqgXC2CW0gHWY7A ($\text{IoU}_i^I = 28.26$).

UPerNet-ResNet101: bvSGJVPKHwU9ApMiF19h_A ($\text{IoU}_i^I = 27.10$).

UPerNet-ConvNeXt: bvSGJVPKHwU9ApMiF19h_A ($\text{IoU}_i^I = 33.15$).

UPerNet-MiTB4: F1rWWZI_pxNjQ7FNya_QKg ($\text{IoU}_i^I = 37.43$).

Figure 20: Worst-case images: Mapillary Vistas 2 / 3. Left: image. Middle: label. Right: prediction.

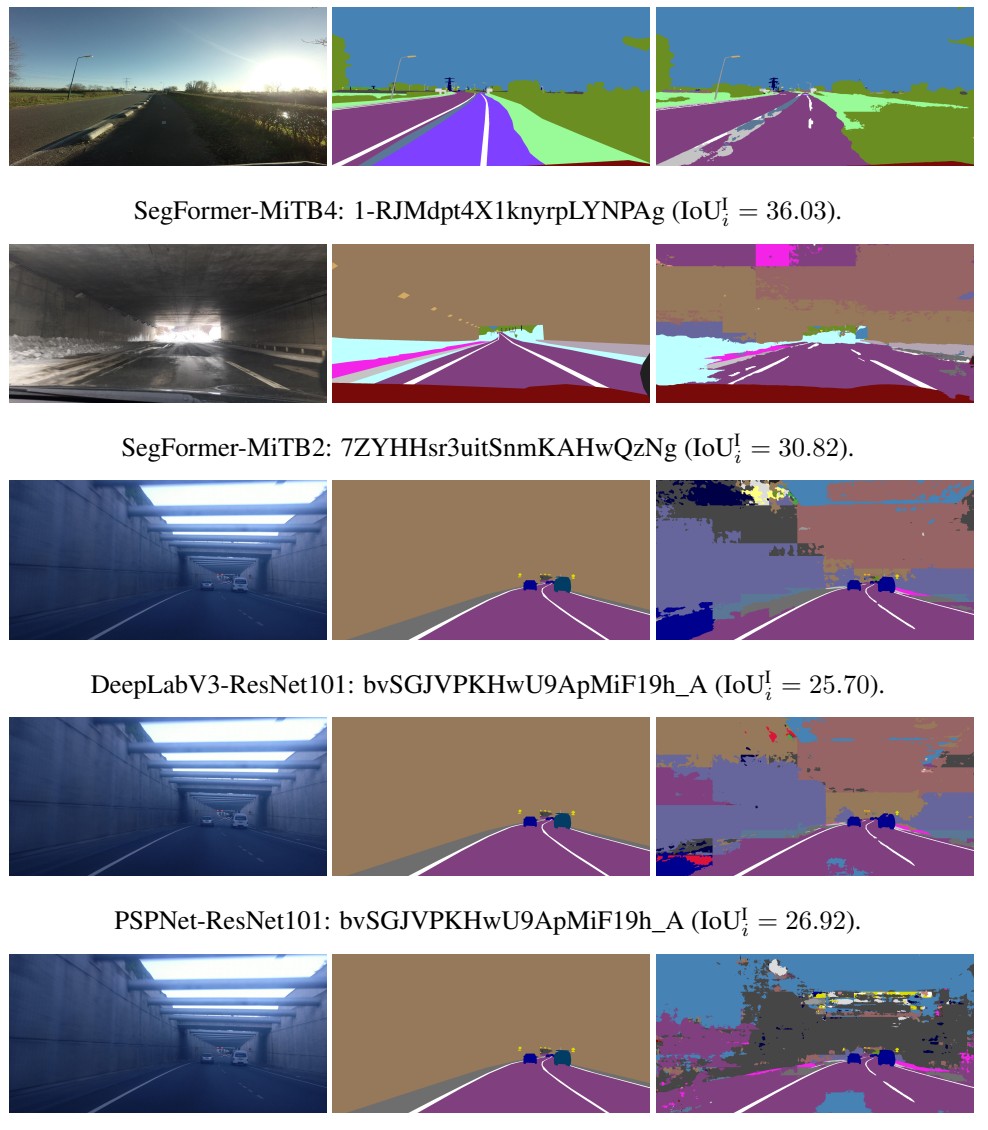

SegFormer-MiTB4: 1-RJMdpt4X1knyrpLYNPAg (IoU$_i^I$ = 36.03).

SegFormer-MiTB2: 7ZYHHsr3uitSnmKAHwQzNg (IoU$_i^I$ = 30.82).

DeepLabV3-ResNet101: bvSGJVPKHwU9ApMiF19h_A (IoU$_i^I$ = 25.70).

PSPNet-ResNet101: bvSGJVPKHwU9ApMiF19h_A (IoU$_i^I$ = 26.92).

UNet-ResNet101: bvSGJVPKHwU9ApMiF19h_A (IoU$_i^I$ = 17.85).

Figure 21: Worst-case images: Mapillary Vistas 3 / 3. Left: image. Middle: label. Right: prediction.

## G.5 CamVid

Table 18: Results: CamVid. Red: the best for each metric. Green: the worst for each metric.

| Method | Backbone | $\text{mIoU}^{I}$ / $\text{mIoU}^{C}$ / $\text{mIoU}^{D}$ | $\text{mIoU}^{I^{\bar{q}}}$ / $\text{mIoU}^{C^{\bar{q}}}$ / mAcc | $\text{mIoU}^{I^{5}}$ / $\text{mIoU}^{C^{5}}$ / Acc | $\text{mIoU}^{I^{1}}$ / $\text{mIoU}^{C^{1}}$ |
|---|---|---|---|---|---|
| DeepLabV3+ | ResNet101 | $79.97 \pm 0.12$ | $74.46 \pm 0.20$ | $65.19 \pm 0.18$ | $62.46 \pm 0.66$ |
|  |  | $79.48 \pm 0.12$ | $67.72 \pm 0.28$ | $37.01 \pm 0.94$ | $25.00 \pm 0.82$ |
|  |  | $83.30 \pm 0.12$ | $89.84 \pm 0.10$ | $95.71 \pm 0.05$ |  |
| DeepLabV3+ | ResNet50 | $79.22 \pm 0.11$ | $73.58 \pm 0.22$ | $63.77 \pm 0.61$ | $60.59 \pm 0.59$ |
|  |  | $78.69 \pm 0.16$ | $66.50 \pm 0.27$ | $35.00 \pm 0.68$ | $23.33 \pm 0.94$ |
|  |  | $82.56 \pm 0.10$ | $89.06 \pm 0.01$ | $95.50 \pm 0.03$ |  |
| DeepLabV3+ | ResNet34 | $79.16 \pm 0.24$ | $72.92 \pm 0.44$ | $62.06 \pm 1.11$ | $58.63 \pm 1.84$ |
|  |  | $78.71 \pm 0.32$ | $66.18 \pm 0.60$ | $33.83 \pm 1.15$ | $22.33 \pm 1.70$ |
|  |  | $81.98 \pm 0.17$ | $88.80 \pm 0.03$ | $95.42 \pm 0.07$ |  |
| DeepLabV3+ | ResNet18 | $78.12 \pm 0.03$ | $71.56 \pm 0.05$ | $60.96 \pm 0.15$ | $57.11 \pm 1.03$ |
|  |  | $77.55 \pm 0.03$ | $64.44 \pm 0.04$ | $32.25 \pm 0.27$ | $18.67 \pm 1.25$ |
|  |  | $80.79 \pm 0.13$ | $87.89 \pm 0.11$ | $95.19 \pm 0.01$ |  |
| DeepLabV3+ | EfficientNetB0 | $78.61 \pm 0.27$ | $72.30 \pm 0.41$ | $62.11 \pm 0.91$ | $57.70 \pm 1.40$ |
|  |  | $78.15 \pm 0.32$ | $65.59 \pm 0.40$ | $34.51 \pm 0.57$ | $21.00 \pm 0.82$ |
|  |  | $80.97 \pm 0.38$ | $88.33 \pm 0.30$ | $95.17 \pm 0.08$ |  |
| DeepLabV3+ | MobileNetV2 | $77.89 \pm 0.08$ | $71.35 \pm 0.04$ | $61.18 \pm 0.45$ | $56.75 \pm 0.65$ |
|  |  | $77.39 \pm 0.12$ | $64.42 \pm 0.21$ | $32.39 \pm 0.35$ | $20.33 \pm 0.94$ |
|  |  | $80.30 \pm 0.16$ | $87.64 \pm 0.03$ | $95.08 \pm 0.09$ |  |
| DeepLabV3+ | MobileViTV2 | $78.58 \pm 0.16$ | $72.03 \pm 0.23$ | $61.70 \pm 0.34$ | $58.49 \pm 0.43$ |
|  |  | $78.07 \pm 0.05$ | $65.36 \pm 0.04$ | $33.81 \pm 0.45$ | $19.00 \pm 0.82$ |
|  |  | $80.62 \pm 0.18$ | $88.06 \pm 0.26$ | $94.91 \pm 0.26$ |  |
| UPerNet | ResNet101 | $79.76 \pm 0.01$ | $74.22 \pm 0.06$ | $65.47 \pm 0.07$ | $63.18 \pm 0.34$ |
|  |  | $79.24 \pm 0.03$ | $67.47 \pm 0.10$ | $37.60 \pm 0.63$ | $25.33 \pm 0.47$ |
|  |  | $83.00 \pm 0.13$ | $89.17 \pm 0.06$ | $95.74 \pm 0.02$ |  |
| UPerNet | ConvNeXt | $80.52 \pm 0.13$ | $75.40 \pm 0.18$ | $66.30 \pm 0.59$ | $63.35 \pm 0.52$ |
|  |  | $80.09 \pm 0.14$ | $68.92 \pm 0.21$ | $38.66 \pm 0.35$ | $24.67 \pm 0.47$ |
|  |  | $84.01 \pm 0.10$ | $90.51 \pm 0.12$ | $95.96 \pm 0.04$ |  |
| UPerNet | MiTB4 | $80.53 \pm 0.09$ | $75.19 \pm 0.20$ | $65.56 \pm 0.76$ | $62.82 \pm 1.32$ |
|  |  | $80.09 \pm 0.13$ | $68.48 \pm 0.28$ | $37.77 \pm 0.65$ | $24.33 \pm 1.25$ |
|  |  | $83.32 \pm 0.09$ | $89.69 \pm 0.04$ | $95.82 \pm 0.03$ |  |
| SegFormer | MiTB4 | $80.04 \pm 0.13$ | $74.61 \pm 0.19$ | $64.98 \pm 0.44$ | $61.53 \pm 0.58$ |
|  |  | $79.52 \pm 0.12$ | $67.88 \pm 0.19$ | $36.94 \pm 0.55$ | $23.00 \pm 0.82$ |
|  |  | $82.95 \pm 0.23$ | $89.68 \pm 0.16$ | $95.71 \pm 0.06$ |  |
| SegFormer | MiTB2 | $79.31 \pm 0.05$ | $73.45 \pm 0.10$ | $64.22 \pm 0.27$ | $62.01 \pm 1.04$ |
|  |  | $78.84 \pm 0.08$ | $66.73 \pm 0.13$ | $35.77 \pm 0.40$ | $23.00 \pm 0.82$ |
|  |  | $82.27 \pm 0.19$ | $89.14 \pm 0.15$ | $95.53 \pm 0.02$ |  |
| DeepLabV3 | ResNet101 | $78.81 \pm 0.04$ | $73.33 \pm 0.10$ | $64.12 \pm 0.37$ | $61.73 \pm 0.55$ |
|  |  | $78.36 \pm 0.02$ | $66.63 \pm 0.05$ | $35.13 \pm 0.39$ | $22.67 \pm 2.05$ |
|  |  | $82.43 \pm 0.08$ | $89.15 \pm 0.05$ | $95.50 \pm 0.04$ |  |
| PSPNet | ResNet101 | $78.62 \pm 0.15$ | $73.13 \pm 0.27$ | $64.18 \pm 0.65$ | $61.57 \pm 0.62$ |
|  |  | $78.12 \pm 0.14$ | $66.32 \pm 0.26$ | $35.74 \pm 0.47$ | $23.67 \pm 0.47$ |
|  |  | $82.13 \pm 0.12$ | $89.14 \pm 0.01$ | $95.46 \pm 0.05$ |  |
| UNet | ResNet101 | $79.69 \pm 0.29$ | $73.83 \pm 0.43$ | $64.32 \pm 0.79$ | $61.48 \pm 0.98$ |
|  |  | $79.19 \pm 0.35$ | $66.99 \pm 0.59$ | $35.29 \pm 0.64$ | $23.00 \pm 1.41$ |
|  |  | $82.84 \pm 0.31$ | $89.26 \pm 0.26$ | $95.63 \pm 0.03$ |  |

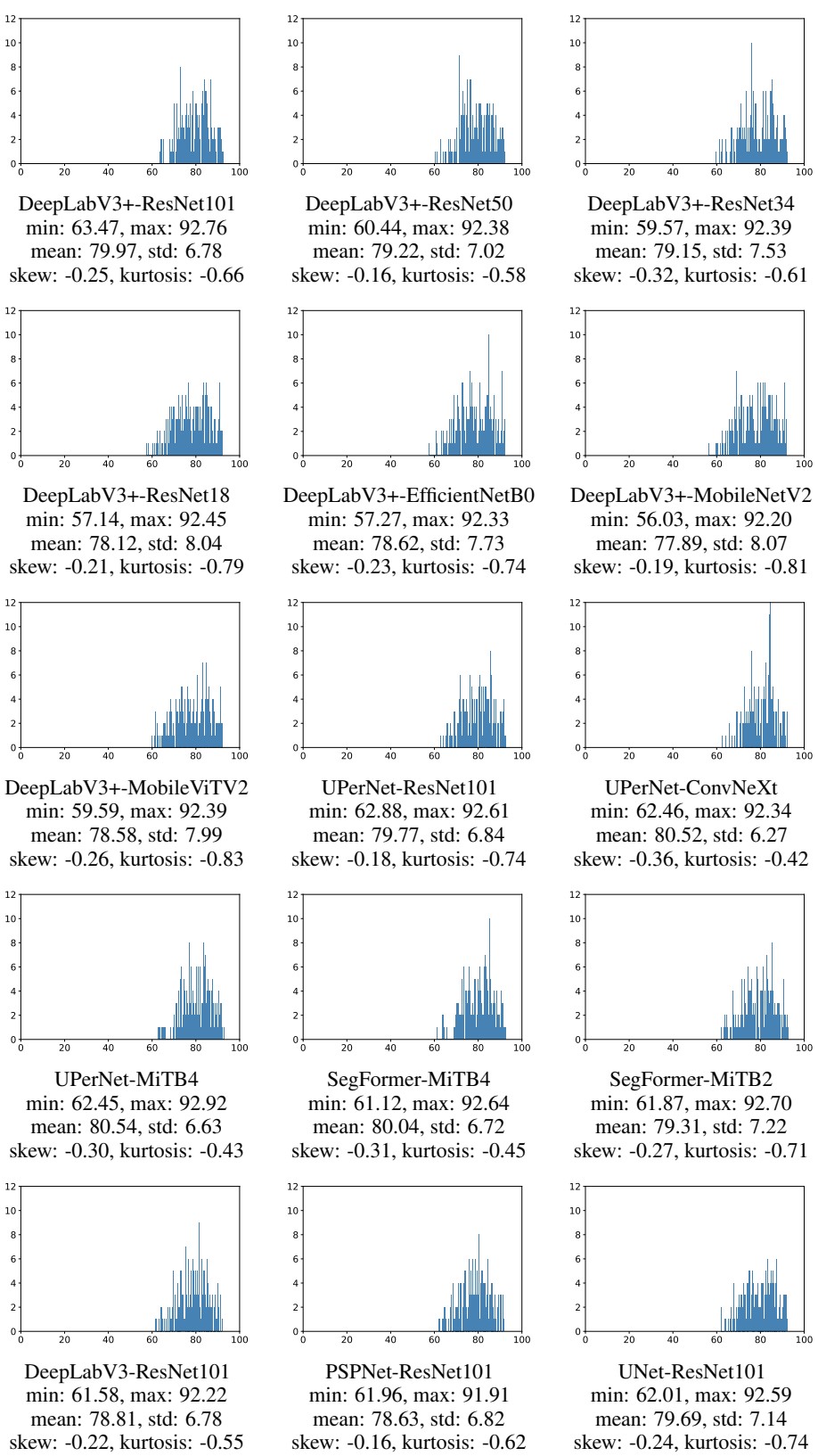

Figure 22: Histograms: CamVid.

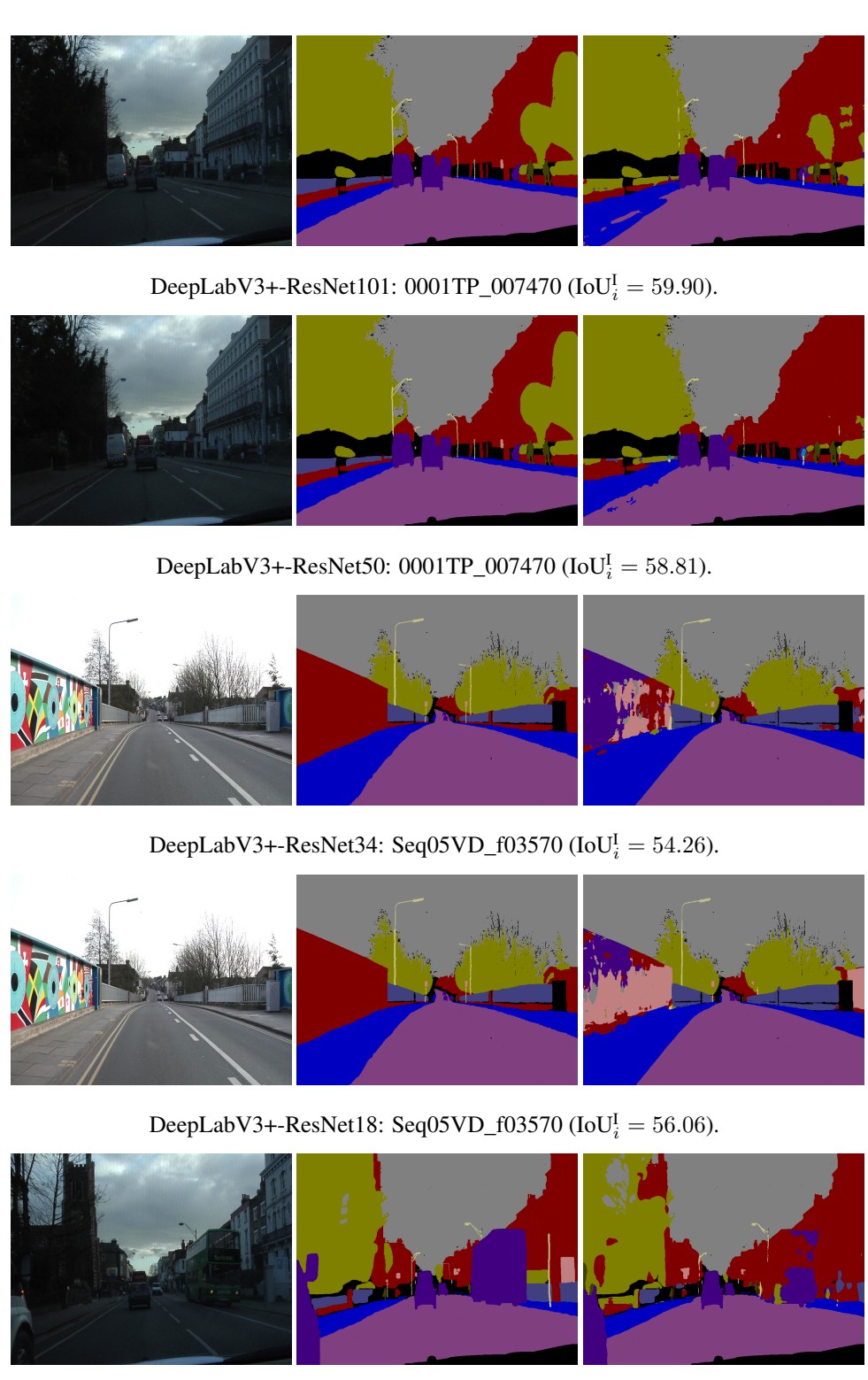

DeepLabV3+-ResNet101: 0001TP_007470 (IoU$_i^I$ = 59.90).

DeepLabV3+-ResNet50: 0001TP_007470 (IoU$_i^I$ = 58.81).

DeepLabV3+-ResNet34: Seq05VD_f03570 (IoU$_i^I$ = 54.26).

DeepLabV3+-ResNet18: Seq05VD_f03570 (IoU$_i^I$ = 56.06).

DeepLabV3+-EfficientNetB0: 0001TP_007560 (IoU$_i^I$ = 53.86).

Figure 23: Worst-case images: CamVid 1 / 3. Left: image. Middle: label. Right: prediction.

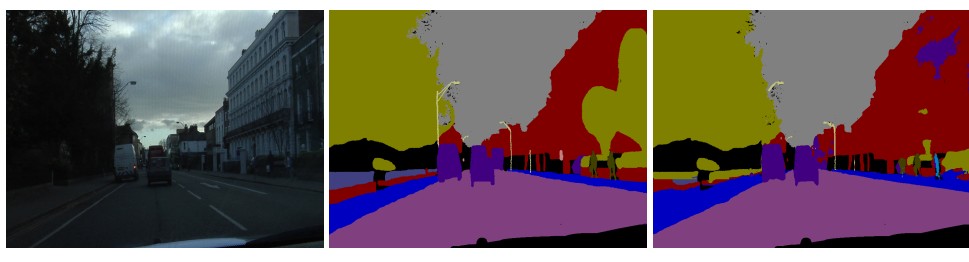

DeepLabV3+-MobileNetV2: 0001TP_007470 (IoU$_i^I$ = 52.72).

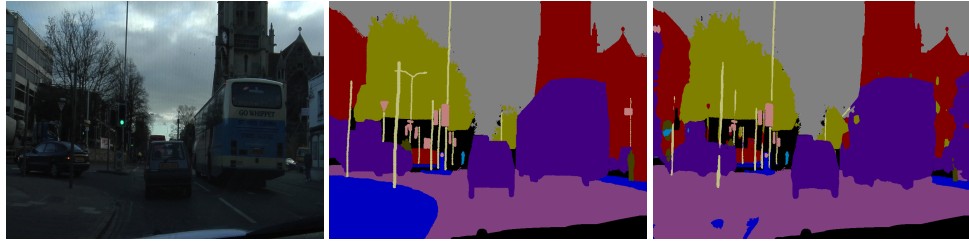

DeepLabV3+-MobileViTV2: 0001TP_006930 (IoU$_i^I$ = 57.51).

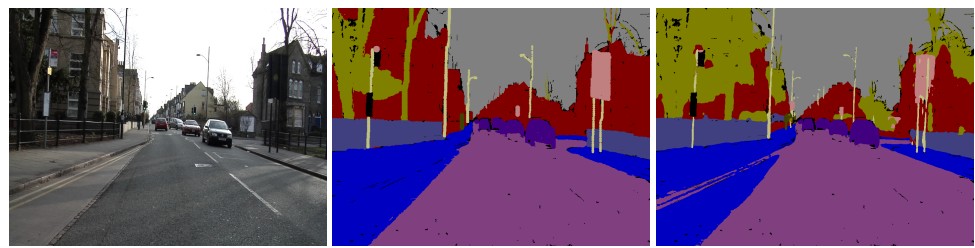

UPerNet-ResNet101: Seq05VD_f00240 (IoU$_i^I$ = 62.33).

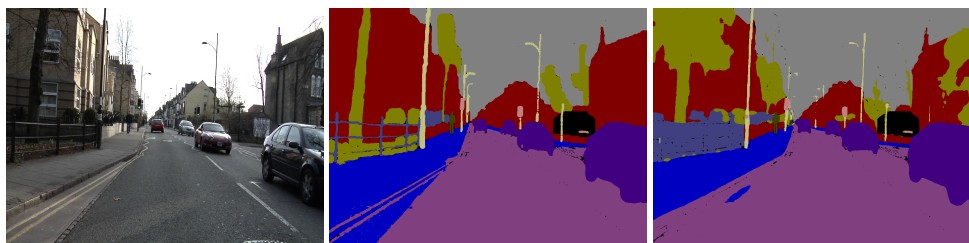

UPerNet-ConvNeXt: Seq05VD_f00270 (IoU$_i^I$ = 61.81).

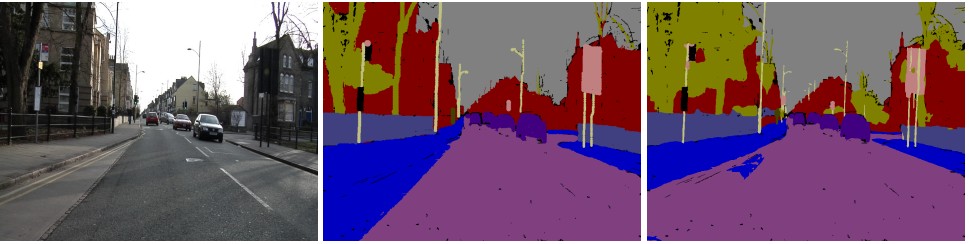

UPerNet-MiTB4: Seq05VD_f00240 (IoU$_i^I$ = 59.61).

Figure 24: Worst-case images: CamVid 2 / 3. Left: image. Middle: label. Right: prediction.

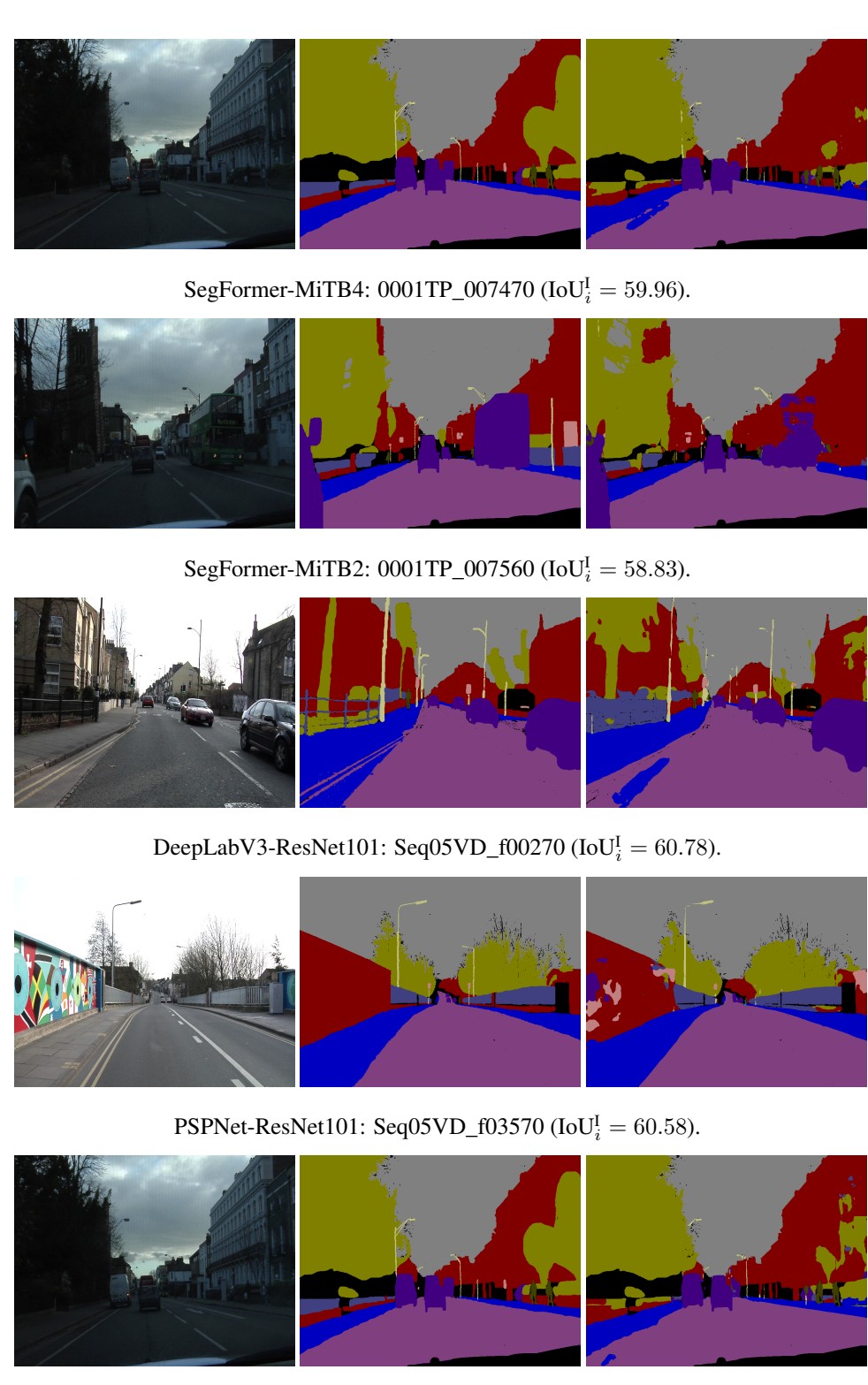

SegFormer-MiTB4: 0001TP_007470 (IoU$_i^I$ = 59.96).

SegFormer-MiTB2: 0001TP_007560 (IoU$_i^I$ = 58.83).

DeepLabV3-ResNet101: Seq05VD_f00270 (IoU$_i^I$ = 60.78).

PSPNet-ResNet101: Seq05VD_f03570 (IoU$_i^I$ = 60.58).

UNet-ResNet101: 0001TP_007470 (IoU$_i^I$ = 60.12).

Figure 25: Worst-case images: CamVid 3 / 3. Left: image. Middle: label. Right: prediction.

Table 19: Results: ADE20K. Red: the best for each metric. Green: the worst for each metric.

| Method | Backbone | $mIoU^I$ / $mIoU^C$ / $mIoU^K$ / $mIoU^D$ | $mIoU^{I\bar{q}}$ / $mIoU^{C\bar{q}}$ / $mIoU^{K\bar{q}}$ / mAcc | $mIoU^{I5}$ / $mIoU^{C5}$ / $mIoU^{K5}$ / Acc | $mIoU^{I1}$ / $mIoU^{C1}$ / $mIoU^{K1}$ |
|---|---|---|---|---|---|
| DeepLabV3+ | ResNet101 | 58.79 ± 0.04 | 45.99 ± 0.05 | 25.18 ± 0.22 | 16.74 ± 0.87 |
|  |  | 46.52 ± 0.30 | 23.86 ± 0.33 | 1.87 ± 0.28 | 1.33 ± 0.47 |
|  |  | 45.84 ± 0.30 | 22.72 ± 0.35 | 1.76 ± 0.30 | 1.00 ± 0.00 |
|  |  | 46.32 ± 0.33 | 60.96 ± 0.51 | 81.25 ± 0.03 |  |
| DeepLabV3+ | ResNet50 | 57.15 ± 0.16 | 44.41 ± 0.14 | 23.71 ± 0.23 | 15.46 ± 0.42 |
|  |  | 44.25 ± 0.36 | 22.06 ± 0.11 | 1.48 ± 0.20 | 0.67 ± 0.47 |
|  |  | 43.73 ± 0.39 | 21.16 ± 0.11 | 1.40 ± 0.16 | 0.67 ± 0.47 |
|  |  | 44.14 ± 0.31 | 59.41 ± 0.51 | 80.16 ± 0.10 |  |
| DeepLabV3+ | ResNet34 | 55.58 ± 0.07 | 42.52 ± 0.09 | 21.73 ± 0.16 | 13.51 ± 0.83 |
|  |  | 40.65 ± 0.09 | 19.98 ± 0.02 | 1.64 ± 0.10 | 1.00 ± 0.00 |
|  |  | 40.02 ± 0.14 | 18.97 ± 0.09 | 1.63 ± 0.10 | 1.00 ± 0.00 |
|  |  | 40.50 ± 0.17 | 54.94 ± 0.23 | 79.49 ± 0.10 |  |
| DeepLabV3+ | ResNet18 | 53.78 ± 0.12 | 40.89 ± 0.11 | 20.54 ± 0.08 | 11.39 ± 0.80 |
|  |  | 37.98 ± 0.17 | 18.31 ± 0.09 | 1.41 ± 0.08 | 1.00 ± 0.00 |
|  |  | 37.52 ± 0.17 | 17.43 ± 0.11 | 1.40 ± 0.09 | 1.00 ± 0.00 |
|  |  | 38.40 ± 0.25 | 52.09 ± 0.26 | 78.48 ± 0.17 |  |
| DeepLabV3+ | EfficientNetB0 | 54.17 ± 0.16 | 40.90 ± 0.17 | 19.40 ± 0.32 | 9.63 ± 0.65 |
|  |  | 38.63 ± 0.13 | 18.49 ± 0.11 | 1.32 ± 0.07 | 1.00 ± 0.00 |
|  |  | 38.23 ± 0.07 | 17.68 ± 0.12 | 1.30 ± 0.08 | 1.00 ± 0.00 |
|  |  | 39.06 ± 0.44 | 53.40 ± 0.25 | 78.88 ± 0.39 |  |
| DeepLabV3+ | MobileNetV2 | 52.85 ± 0.13 | 39.92 ± 0.12 | 19.94 ± 0.18 | 11.25 ± 0.29 |
|  |  | 36.42 ± 0.04 | 17.49 ± 0.09 | 1.37 ± 0.13 | 1.00 ± 0.00 |
|  |  | 35.93 ± 0.11 | 16.55 ± 0.13 | 1.35 ± 0.12 | 1.00 ± 0.00 |
|  |  | 36.85 ± 0.34 | 50.50 ± 0.27 | 77.99 ± 0.22 |  |
| DeepLabV3+ | MobileViTV2 | 55.05 ± 0.11 | 42.01 ± 0.14 | 21.73 ± 0.30 | 14.52 ± 0.44 |
|  |  | 39.98 ± 0.11 | 19.12 ± 0.19 | 1.18 ± 0.22 | 0.67 ± 0.47 |
|  |  | 39.62 ± 0.15 | 18.30 ± 0.19 | 1.11 ± 0.20 | 0.33 ± 0.47 |
|  |  | 41.36 ± 0.43 | 55.07 ± 0.52 | 79.34 ± 0.05 |  |
| UPerNet | ResNet101 | 57.85 ± 0.01 | 45.14 ± 0.07 | 24.22 ± 0.30 | 14.91 ± 0.85 |
|  |  | 45.33 ± 0.25 | 22.90 ± 0.23 | 1.91 ± 0.19 | 1.00 ± 0.00 |
|  |  | 44.85 ± 0.27 | 21.99 ± 0.25 | 1.77 ± 0.16 | 1.00 ± 0.00 |
|  |  | 45.89 ± 0.11 | 59.65 ± 0.35 | 80.97 ± 0.06 |  |
| UPerNet | ConvNeXt | 60.62 ± 0.07 | 48.13 ± 0.06 | 28.39 ± 0.11 | 19.61 ± 0.34 |
|  |  | 48.45 ± 0.15 | 25.28 ± 0.03 | 2.14 ± 0.03 | 1.00 ± 0.00 |
|  |  | 47.58 ± 0.17 | 24.08 ± 0.08 | 1.98 ± 0.03 | 1.00 ± 0.00 |
|  |  | 51.08 ± 0.42 | 63.34 ± 0.50 | 83.55 ± 0.07 |  |
| UPerNet | MiTB4 | 60.64 ± 0.11 | 47.81 ± 0.15 | 26.84 ± 0.04 | 17.48 ± 0.22 |
|  |  | 48.58 ± 0.10 | 25.47 ± 0.08 | 2.36 ± 0.09 | 1.67 ± 0.47 |
|  |  | 47.97 ± 0.13 | 24.49 ± 0.15 | 2.16 ± 0.12 | 1.00 ± 0.00 |
|  |  | 50.25 ± 0.15 | 64.13 ± 0.22 | 83.23 ± 0.12 |  |
| SegFormer | MiTB4 | 60.29 ± 0.07 | 47.64 ± 0.07 | 26.96 ± 0.08 | 17.39 ± 0.31 |
|  |  | 48.10 ± 0.09 | 25.29 ± 0.12 | 2.32 ± 0.16 | 1.33 ± 0.47 |
|  |  | 47.47 ± 0.12 | 24.25 ± 0.16 | 2.09 ± 0.15 | 1.00 ± 0.00 |
|  |  | 50.23 ± 0.26 | 64.19 ± 0.26 | 83.09 ± 0.03 |  |
| SegFormer | MiTB2 | 58.28 ± 0.03 | 45.72 ± 0.06 | 25.45 ± 0.02 | 17.02 ± 0.21 |
|  |  | 45.79 ± 0.09 | 23.59 ± 0.18 | 2.52 ± 0.18 | 1.67 ± 0.47 |
|  |  | 45.18 ± 0.07 | 22.71 ± 0.18 | 2.37 ± 0.18 | 1.67 ± 0.47 |
|  |  | 47.18 ± 0.07 | 61.34 ± 0.20 | 81.72 ± 0.05 |  |
| DeepLabV3 | ResNet101 | 57.27 ± 0.06 | 44.60 ± 0.09 | 24.09 ± 0.23 | 15.67 ± 0.43 |
|  |  | 44.75 ± 0.11 | 22.72 ± 0.19 | 1.89 ± 0.17 | 1.00 ± 0.00 |
|  |  | 43.96 ± 0.15 | 21.59 ± 0.21 | 1.78 ± 0.19 | 1.00 ± 0.00 |
|  |  | 46.08 ± 0.08 | 60.20 ± 0.25 | 81.07 ± 0.08 |  |
| PSPNet | ResNet101 | 57.37 ± 0.06 | 44.90 ± 0.03 | 24.79 ± 0.18 | 16.81 ± 0.35 |
|  |  | 44.86 ± 0.22 | 22.84 ± 0.21 | 1.79 ± 0.14 | 1.00 ± 0.00 |
|  |  | 44.29 ± 0.25 | 21.86 ± 0.20 | 1.72 ± 0.14 | 1.00 ± 0.00 |
|  |  | 45.69 ± 0.33 | 59.42 ± 0.57 | 81.00 ± 0.14 |  |
| UNet | ResNet101 | 56.86 ± 0.05 | 44.15 ± 0.02 | 24.05 ± 0.31 | 14.84 ± 0.48 |
|  |  | 42.17 ± 0.05 | 21.04 ± 0.09 | 1.49 ± 0.04 | 1.00 ± 0.00 |
|  |  | 41.43 ± 0.02 | 19.83 ± 0.05 | 1.42 ± 0.02 | 1.00 ± 0.00 |
|  |  | 40.78 ± 0.07 | 54.58 ± 0.15 | 79.24 ± 0.04 |  |

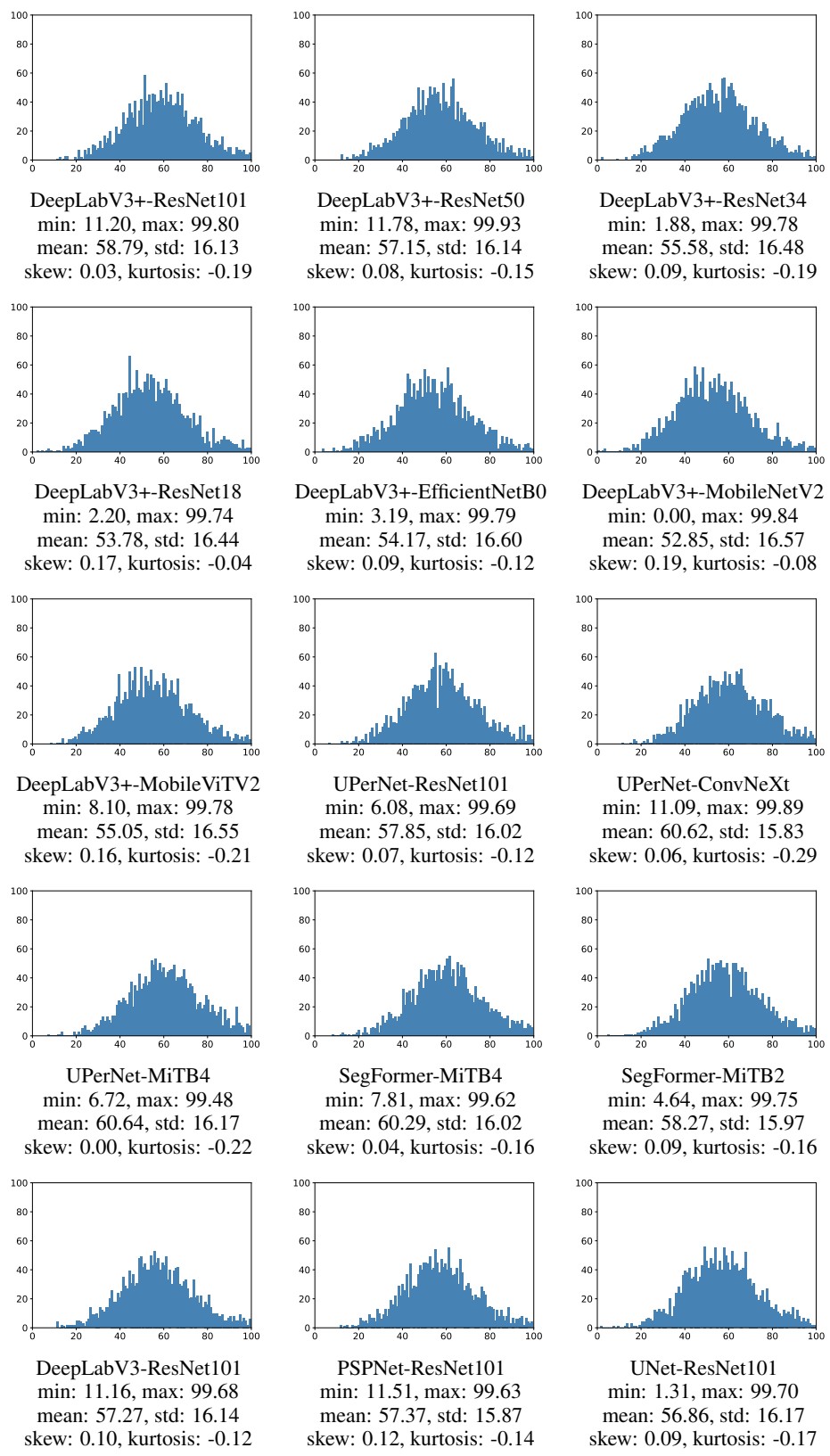

Figure 26: Histograms: ADE20K.

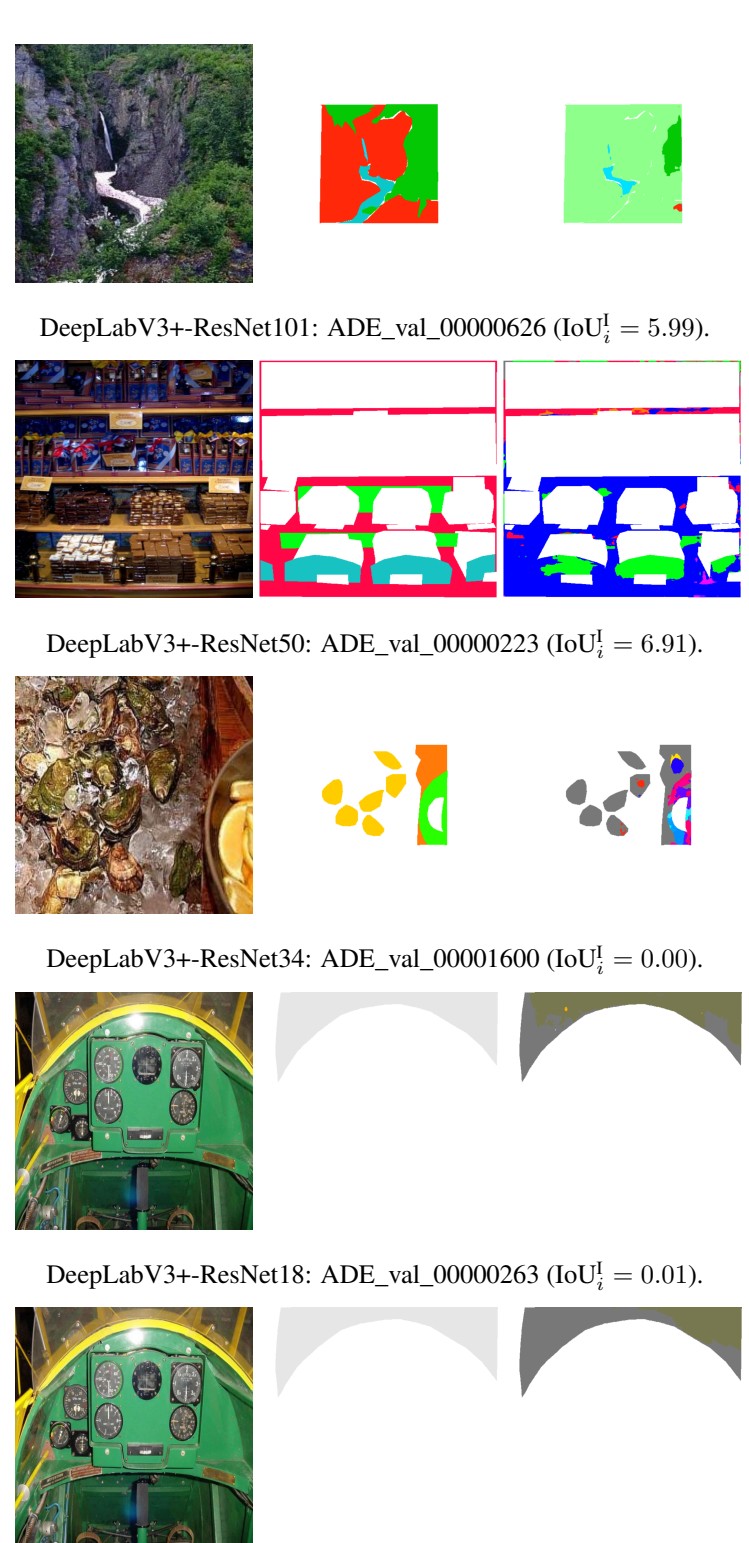

DeepLabV3+-ResNet101: ADE_val_00000626 ($IoU_i^I = 5.99$).

DeepLabV3+-ResNet50: ADE_val_00000223 ($IoU_i^I = 6.91$).

DeepLabV3+-ResNet34: ADE_val_00001600 ($IoU_i^I = 0.00$).

DeepLabV3+-ResNet18: ADE_val_00000263 ($IoU_i^I = 0.01$).

DeepLabV3+-EfficientNetB0: ADE_val_00000263 ($IoU_i^I = 0.00$).

Figure 27: Worst-case images: ADE20K 1 / 3. Left: image. Middle: label. Right: prediction.

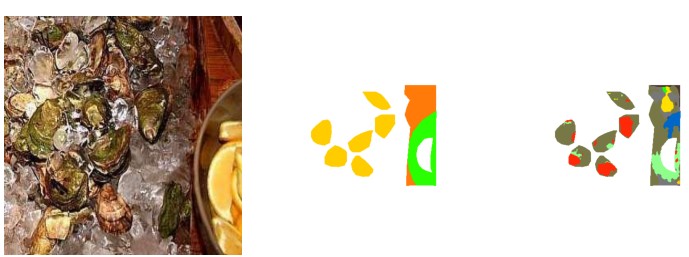

DeepLabV3+-MobileNetV2: ADE_val_00001600 ($\text{IoU}_i^\text{I} = 0.00$).

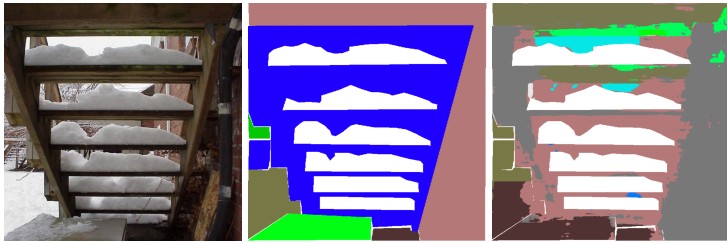

DeepLabV3+-MobileViTV2: ADE_val_00000054 ($\text{IoU}_i^\text{I} = 4.27$).

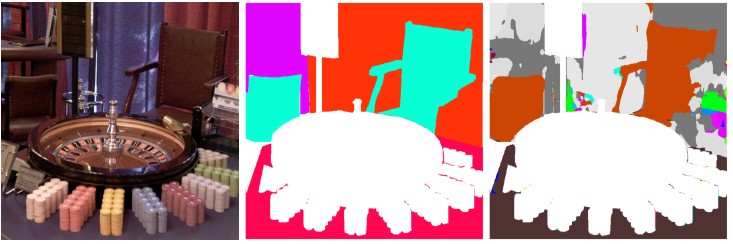

UPerNet-ResNet101: ADE_val_00001229 ($\text{IoU}_i^\text{I} = 0.06$).

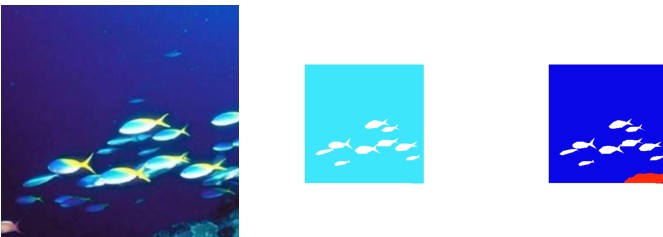

UPerNet-ConvNeXt: ADE_val_00001897 ($\text{IoU}_i^\text{I} = 0.00$).

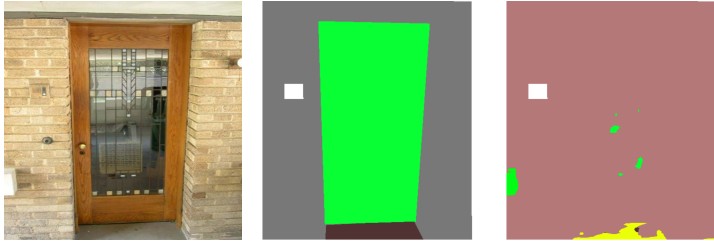

UPerNet-MiTB4: ADE_val_00001951 ($\text{IoU}_i^\text{I} = 0.58$).

Figure 28: Worst-case images: ADE20K 2 / 3. Left: image. Middle: label. Right: prediction.

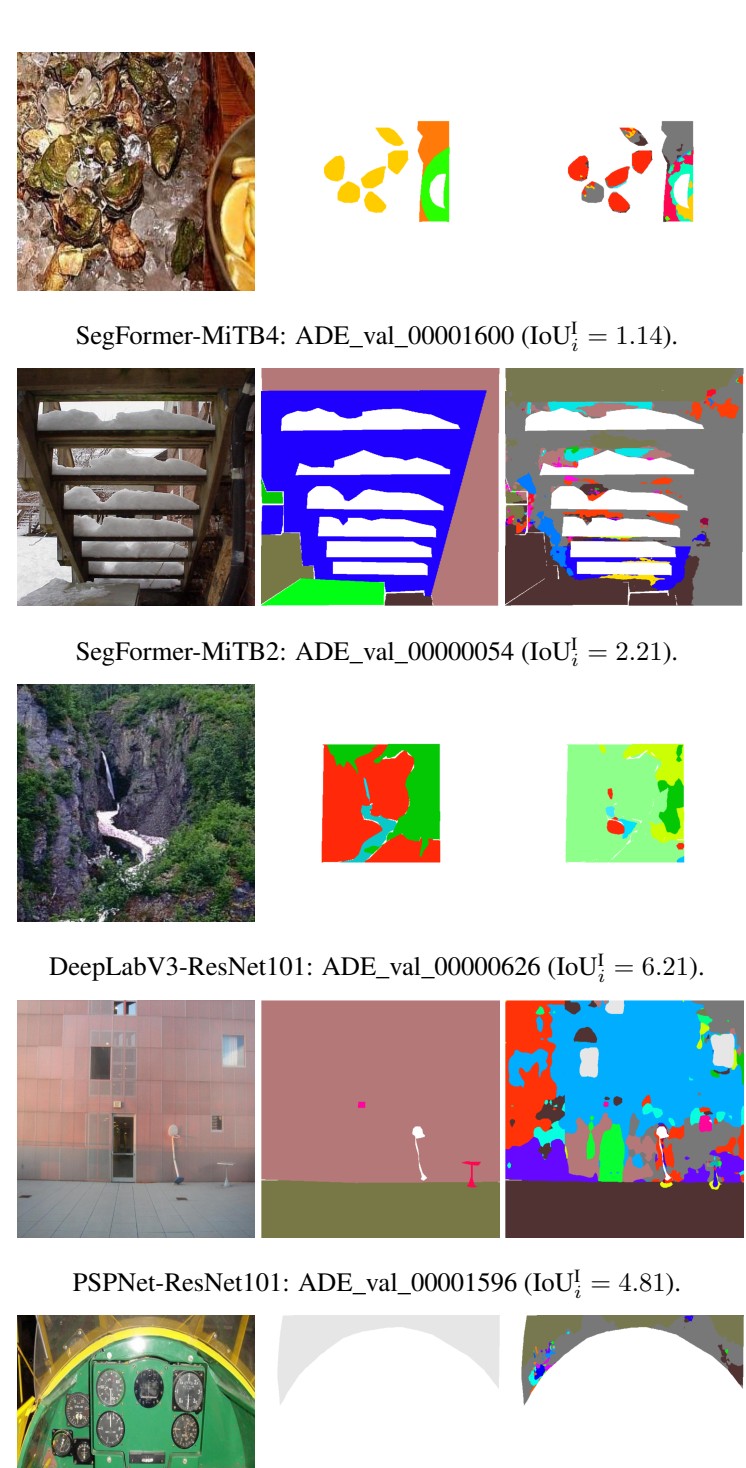

SegFormer-MiTB4: ADE_val_00001600 ($IoU_i^I = 1.14$).

SegFormer-MiTB2: ADE_val_00000054 ($IoU_i^I = 2.21$).

DeepLabV3-ResNet101: ADE_val_00000626 ($IoU_i^I = 6.21$).

PSPNet-ResNet101: ADE_val_00001596 ($IoU_i^I = 4.81$).

UNet-ResNet101: ADE_val_00000263 ($IoU_i^I = 0.00$).

Figure 29: Worst-case images: ADE20K 3 / 3. Left: image. Middle: label. Right: prediction.

## G.7 COCO-Stuff

Table 20: Results: COCO-Stuff. Red: the best for each metric. Green: the worst for each metric.

| Method | Backbone | $\text{mIoU}^{\text{I}}$ / $\text{mIoU}^{\text{C}}$ / $\text{mIoU}^{\text{D}}$ | $\text{mIoU}^{\text{I}^{\bar{q}}}$ / $\text{mIoU}^{\text{C}^{\bar{q}}}$ / mAcc | $\text{mIoU}^{\text{I}^5}$ / $\text{mIoU}^{\text{C}^5}$ / Acc | $\text{mIoU}^{\text{I}^1}$ / $\text{mIoU}^{\text{C}^1}$ |
|---|---|---|---|---|---|
| DeepLabV3+ | ResNet101 | 48.71 ± 0.06 | 35.26 ± 0.09 | 17.07 ± 0.22 | 8.48 ± 0.39 |
| | | 41.98 ± 0.04 | 22.46 ± 0.04 | 1.33 ± 0.07 | 0.00 ± 0.00 |
| | | 44.09 ± 0.17 | 59.35 ± 0.14 | 69.56 ± 0.12 | |
| DeepLabV3+ | ResNet50 | 46.91 ± 0.04 | 33.39 ± 0.06 | 15.40 ± 0.23 | 7.34 ± 0.53 |
| | | 39.66 ± 0.04 | 20.75 ± 0.08 | 1.19 ± 0.04 | 0.00 ± 0.00 |
| | | 42.15 ± 0.03 | 56.93 ± 0.08 | 68.63 ± 0.01 | |
| DeepLabV3+ | ResNet34 | 43.32 ± 0.04 | 29.54 ± 0.08 | 11.76 ± 0.20 | 3.76 ± 0.23 |
| | | 34.39 ± 0.06 | 16.77 ± 0.03 | 0.56 ± 0.04 | 0.00 ± 0.00 |
| | | 36.30 ± 0.18 | 50.50 ± 0.16 | 66.12 ± 0.07 | |
| DeepLabV3+ | ResNet18 | 41.60 ± 0.05 | 27.86 ± 0.07 | 10.48 ± 0.15 | 2.79 ± 0.27 |
| | | 32.33 ± 0.04 | 15.43 ± 0.06 | 0.38 ± 0.03 | 0.00 ± 0.00 |
| | | 34.18 ± 0.18 | 47.90 ± 0.17 | 64.78 ± 0.05 | |
| DeepLabV3+ | EfficientNetB0 | 42.61 ± 0.16 | 28.84 ± 0.25 | 11.39 ± 0.54 | 3.92 ± 0.44 |
| | | 33.64 ± 0.24 | 16.22 ± 0.19 | 0.35 ± 0.08 | 0.00 ± 0.00 |
| | | 35.65 ± 0.22 | 50.23 ± 0.32 | 65.59 ± 0.16 | |
| DeepLabV3+ | MobileNetV2 | 40.36 ± 0.09 | 26.64 ± 0.08 | 9.54 ± 0.03 | 2.57 ± 0.12 |
| | | 30.95 ± 0.05 | 14.38 ± 0.04 | 0.36 ± 0.03 | 0.00 ± 0.00 |
| | | 32.65 ± 0.17 | 46.15 ± 0.09 | 63.87 ± 0.08 | |
| DeepLabV3+ | MobileViTV2 | 44.54 ± 0.02 | 30.90 ± 0.02 | 13.68 ± 0.11 | 6.35 ± 0.29 |
| | | 36.23 ± 0.03 | 18.06 ± 0.06 | 0.75 ± 0.10 | 0.00 ± 0.00 |
| | | 38.99 ± 0.03 | 53.15 ± 0.12 | 67.01 ± 0.03 | |
| UPerNet | ResNet101 | 48.16 ± 0.10 | 34.73 ± 0.13 | 16.69 ± 0.24 | 8.20 ± 0.76 |
| | | 41.16 ± 0.16 | 21.75 ± 0.14 | 1.31 ± 0.05 | 0.00 ± 0.00 |
| | | 43.87 ± 0.07 | 58.57 ± 0.08 | 69.47 ± 0.03 | |
| UPerNet | ConvNeXt | 51.34 ± 0.04 | 38.06 ± 0.02 | 19.77 ± 0.05 | 10.99 ± 0.07 |
| | | 44.91 ± 0.04 | 24.74 ± 0.03 | 1.83 ± 0.04 | 0.00 ± 0.00 |
| | | 48.51 ± 0.03 | 62.61 ± 0.06 | 71.78 ± 0.04 | |
| UPerNet | MiTB4 | 50.74 ± 0.03 | 37.23 ± 0.05 | 19.11 ± 0.20 | 11.50 ± 0.45 |
| | | 43.69 ± 0.07 | 23.62 ± 0.05 | 1.52 ± 0.10 | 0.00 ± 0.00 |
| | | 47.02 ± 0.14 | 61.41 ± 0.15 | 71.52 ± 0.04 | |
| SegFormer | MiTB4 | 50.46 ± 0.17 | 37.02 ± 0.17 | 19.02 ± 0.03 | 11.52 ± 0.13 |
| | | 43.33 ± 0.24 | 23.37 ± 0.20 | 1.47 ± 0.08 | 0.00 ± 0.00 |
| | | 46.73 ± 0.37 | 61.32 ± 0.19 | 71.34 ± 0.11 | |
| SegFormer | MiTB2 | 48.74 ± 0.02 | 35.28 ± 0.01 | 17.65 ± 0.06 | 10.12 ± 0.37 |
| | | 41.36 ± 0.00 | 21.89 ± 0.04 | 1.46 ± 0.09 | 0.00 ± 0.00 |
| | | 44.99 ± 0.09 | 59.44 ± 0.04 | 70.35 ± 0.02 | |
| DeepLabV3 | ResNet101 | 48.21 ± 0.01 | 34.91 ± 0.02 | 17.12 ± 0.09 | 9.12 ± 0.08 |
| | | 41.37 ± 0.11 | 21.96 ± 0.07 | 1.34 ± 0.01 | 0.00 ± 0.00 |
| | | 44.29 ± 0.03 | 59.29 ± 0.14 | 69.63 ± 0.09 | |
| PSPNet | ResNet101 | 48.05 ± 0.08 | 34.78 ± 0.08 | 16.90 ± 0.08 | 8.49 ± 0.53 |
| | | 41.30 ± 0.04 | 21.94 ± 0.07 | 1.30 ± 0.07 | 0.00 ± 0.00 |
| | | 44.19 ± 0.05 | 59.04 ± 0.01 | 69.51 ± 0.14 | |
| UNet | ResNet101 | 45.80 ± 0.10 | 32.54 ± 0.06 | 14.73 ± 0.33 | 7.27 ± 0.91 |
| | | 37.76 ± 0.19 | 19.43 ± 0.26 | 0.97 ± 0.20 | 0.00 ± 0.00 |
| | | 37.14 ± 0.34 | 51.43 ± 0.25 | 64.68 ± 0.14 | |

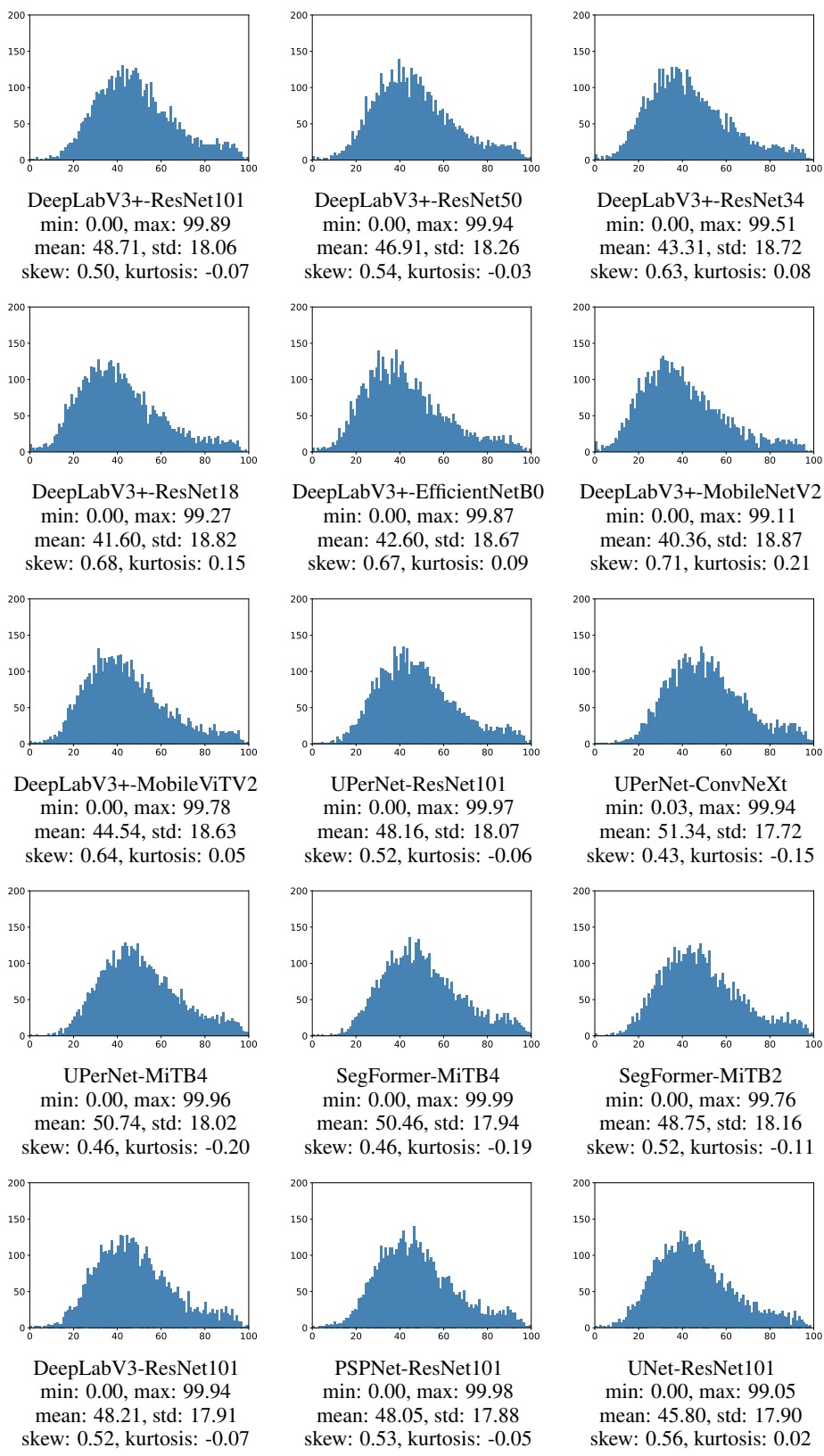

Figure 30: Histograms: COCO-Stuff.

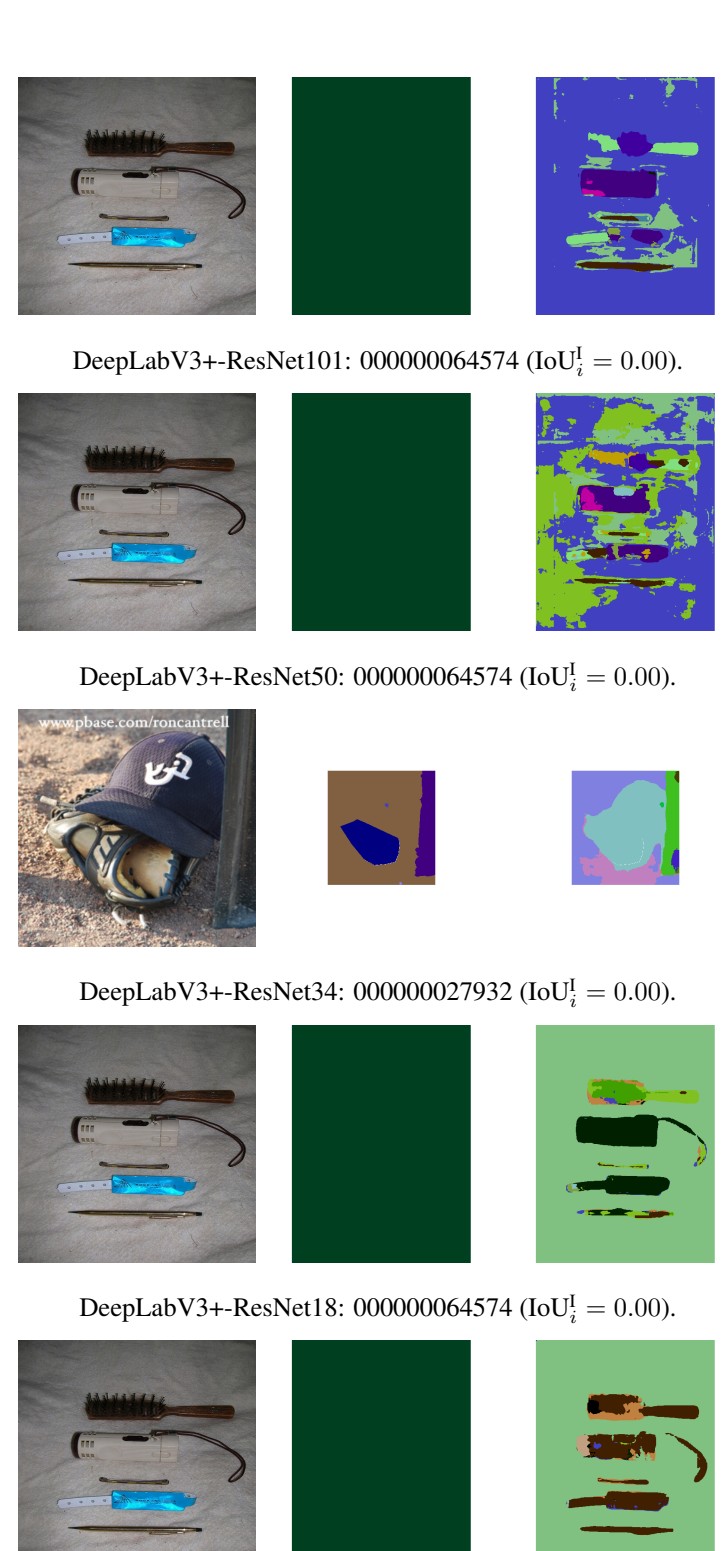

DeepLabV3+-ResNet101: 000000064574 ($\text{IoU}_i^I = 0.00$).

DeepLabV3+-ResNet50: 000000064574 ($\text{IoU}_i^I = 0.00$).

DeepLabV3+-ResNet34: 000000027932 ($\text{IoU}_i^I = 0.00$).

DeepLabV3+-ResNet18: 000000064574 ($\text{IoU}_i^I = 0.00$).

DeepLabV3+-EfficientNetB0: 000000064574 ($\text{IoU}_i^I = 0.00$).

Figure 31: Worst-case images: COCO-Stuff 1 / 3. Left: image. Middle: label. Right: prediction.

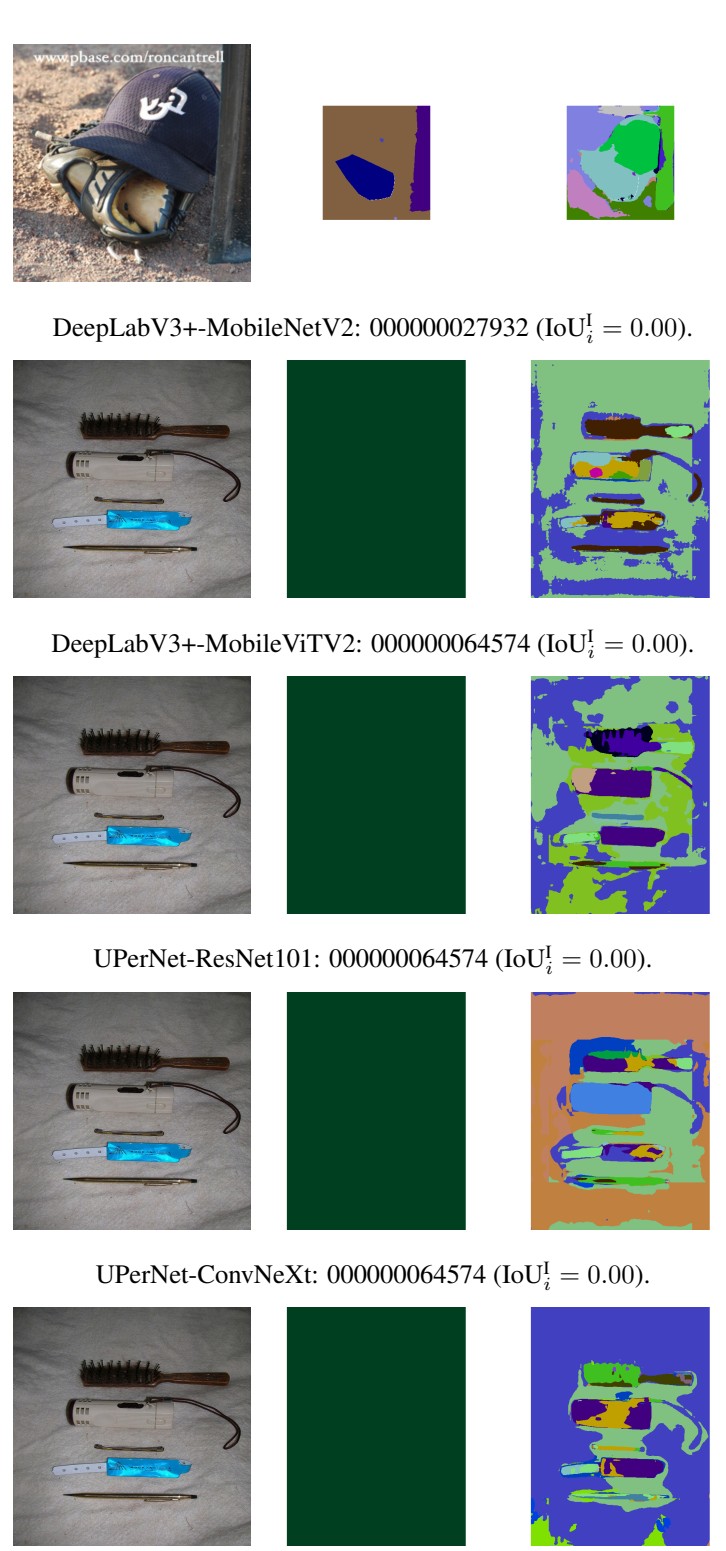

DeepLabV3+-MobileNetV2: 000000027932 ($\text{IoU}_i^\text{I} = 0.00$).

DeepLabV3+-MobileViTV2: 000000064574 ($\text{IoU}_i^\text{I} = 0.00$).

UPerNet-ResNet101: 000000064574 ($\text{IoU}_i^\text{I} = 0.00$).

UPerNet-ConvNeXt: 000000064574 ($\text{IoU}_i^\text{I} = 0.00$).

UPerNet-MiTB4: 000000064574 ($\text{IoU}_i^\text{I} = 0.00$).

Figure 32: Worst-case images: COCO-Stuff 2 / 3. Left: image. Middle: label. Right: prediction.

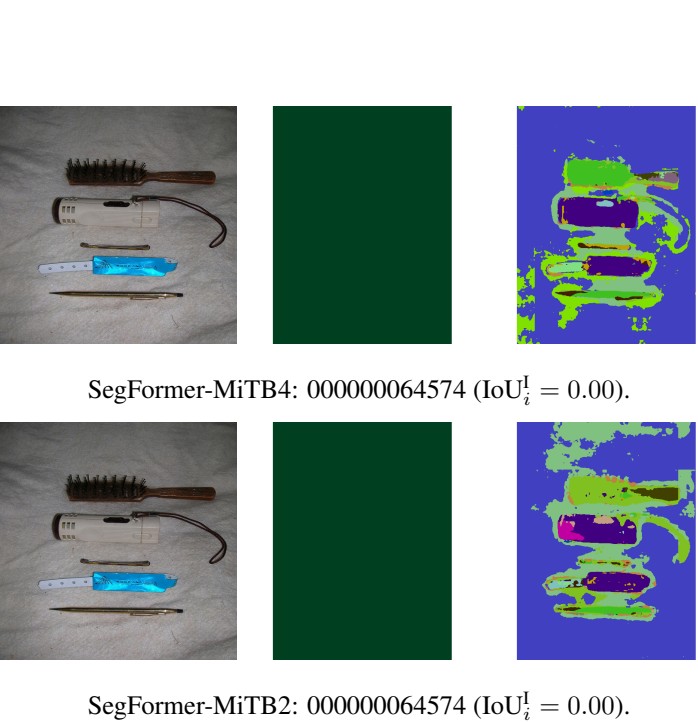

SegFormer-MiTB4: 000000064574 (IoU$_i^{\text{I}}$ = 0.00).

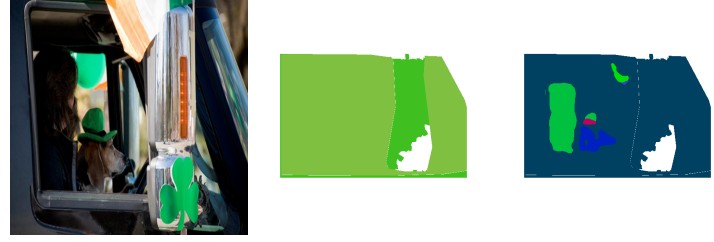

SegFormer-MiTB2: 000000064574 (IoU$_i^{\text{I}}$ = 0.00).

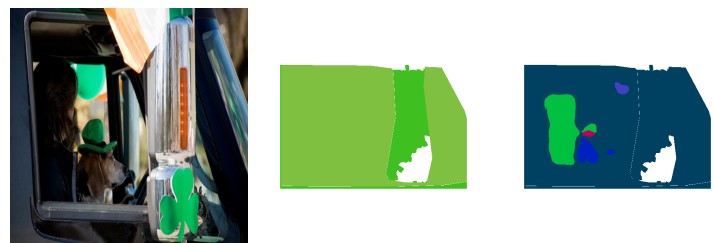

DeepLabV3-ResNet101: 000000053529 (IoU$_i^{\text{I}}$ = 0.00).

PSPNet-ResNet101: 000000053529 (IoU$_i^{\text{I}}$ = 0.00).

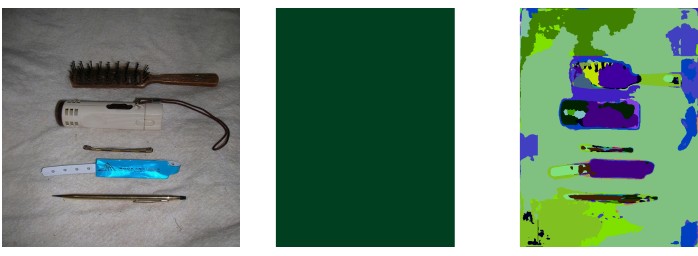

UNet-ResNet101: 000000064574 (IoU$_i^{\text{I}}$ = 0.00).

Figure 33: Worst-case images: COCO-Stuff 3 / 3. Left: image. Middle: label. Right: prediction.

## G.8  PASCAL VOC

Table 21: Results: PASCAL VOC. Red: the best for each metric. Green: the worst for each metric.

| Method | Backbone | $mIoU^I$ / $mIoU^C$ / $mIoU^D$ | $mIoU^{I^{\bar{q}}}$ / $mIoU^{C^{\bar{q}}}$ / mAcc | $mIoU^{I^5}$ / $mIoU^{C^5}$ / Acc | $mIoU^{I^1}$ / $mIoU^{C^1}$ |
|---|---|---|---|---|---|
| DeepLabV3+ | ResNet101 | $86.95 \pm 0.06$ / $79.07 \pm 0.18$ / $81.07 \pm 0.28$ | $75.76 \pm 0.09$ / $62.71 \pm 0.39$ / $88.68 \pm 0.34$ | $45.67 \pm 0.27$ / $17.32 \pm 1.27$ / $95.27 \pm 0.02$ | $31.61 \pm 0.41$ / $1.67 \pm 0.47$ |
| DeepLabV3+ | ResNet50 | $85.80 \pm 0.12$ / $77.56 \pm 0.03$ / $79.71 \pm 0.14$ | $74.12 \pm 0.17$ / $60.20 \pm 0.21$ / $87.66 \pm 0.22$ | $45.54 \pm 0.41$ / $13.03 \pm 1.65$ / $94.86 \pm 0.11$ | $30.95 \pm 0.42$ / $2.67 \pm 1.25$ |
| DeepLabV3+ | ResNet34 | $82.96 \pm 0.09$ / $73.16 \pm 0.19$ / $74.37 \pm 0.68$ | $69.47 \pm 0.24$ / $52.11 \pm 0.38$ / $84.97 \pm 0.53$ | $39.79 \pm 0.54$ / $5.12 \pm 0.32$ / $93.54 \pm 0.11$ | $24.23 \pm 0.82$ / $0.33 \pm 0.47$ |
| DeepLabV3+ | ResNet18 | $81.95 \pm 0.07$ / $71.63 \pm 0.17$ / $73.30 \pm 0.22$ | $68.27 \pm 0.18$ / $50.42 \pm 0.22$ / $83.86 \pm 0.23$ | $38.89 \pm 0.37$ / $4.04 \pm 0.21$ / $93.16 \pm 0.04$ | $24.60 \pm 1.02$ / $0.00 \pm 0.00$ |
| DeepLabV3+ | EfficientNetB0 | $82.22 \pm 0.06$ / $72.06 \pm 0.13$ / $73.58 \pm 0.78$ | $68.57 \pm 0.10$ / $51.56 \pm 0.14$ / $83.85 \pm 0.47$ | $38.06 \pm 0.14$ / $5.73 \pm 0.51$ / $93.32 \pm 0.16$ | $21.28 \pm 0.63$ / $0.00 \pm 0.00$ |
| DeepLabV3+ | MobileNetV2 | $81.00 \pm 0.09$ / $70.06 \pm 0.10$ / $72.05 \pm 0.24$ | $66.93 \pm 0.14$ / $48.38 \pm 0.25$ / $82.75 \pm 0.41$ | $37.47 \pm 0.18$ / $3.37 \pm 0.44$ / $92.89 \pm 0.05$ | $22.27 \pm 0.36$ / $0.67 \pm 0.47$ |
| DeepLabV3+ | MobileViTV2 | $83.41 \pm 0.12$ / $73.32 \pm 0.17$ / $75.75 \pm 0.20$ | $70.12 \pm 0.20$ / $53.02 \pm 0.43$ / $85.34 \pm 0.18$ | $39.78 \pm 0.68$ / $5.91 \pm 0.76$ / $93.95 \pm 0.07$ | $26.77 \pm 1.44$ / $0.00 \pm 0.00$ |
| UPerNet | ResNet101 | $86.52 \pm 0.03$ / $78.23 \pm 0.08$ / $80.50 \pm 0.15$ | $74.98 \pm 0.08$ / $61.48 \pm 0.26$ / $87.70 \pm 0.07$ | $44.76 \pm 0.25$ / $14.80 \pm 0.90$ / $95.19 \pm 0.04$ | $29.14 \pm 0.53$ / $1.67 \pm 1.25$ |
| UPerNet | ConvNeXt | $88.30 \pm 0.11$ / $81.08 \pm 0.25$ / $83.48 \pm 0.08$ | $78.14 \pm 0.27$ / $66.35 \pm 0.39$ / $89.01 \pm 0.17$ | $49.92 \pm 0.64$ / $24.58 \pm 0.52$ / $95.95 \pm 0.03$ | $33.31 \pm 0.64$ / $8.33 \pm 1.25$ |
| UPerNet | MiTB4 | $87.76 \pm 0.12$ / $80.02 \pm 0.29$ / $82.71 \pm 0.22$ | $76.98 \pm 0.27$ / $63.68 \pm 0.54$ / $89.58 \pm 0.30$ | $47.25 \pm 0.72$ / $16.30 \pm 2.20$ / $95.73 \pm 0.05$ | $32.96 \pm 0.19$ / $3.00 \pm 1.41$ |
| SegFormer | MiTB4 | $87.54 \pm 0.15$ / $79.80 \pm 0.30$ / $82.33 \pm 0.35$ | $76.74 \pm 0.28$ / $63.89 \pm 0.51$ / $89.16 \pm 0.29$ | $46.88 \pm 0.32$ / $20.05 \pm 1.57$ / $95.65 \pm 0.08$ | $30.82 \pm 0.82$ / $4.67 \pm 1.70$ |
| SegFormer | MiTB2 | $86.26 \pm 0.10$ / $77.57 \pm 0.23$ / $80.45 \pm 0.29$ | $74.51 \pm 0.18$ / $60.23 \pm 0.15$ / $88.02 \pm 0.30$ | $43.45 \pm 0.31$ / $11.76 \pm 0.50$ / $95.17 \pm 0.08$ | $28.08 \pm 0.68$ / $1.33 \pm 0.47$ |
| DeepLabV3 | ResNet101 | $86.32 \pm 0.03$ / $77.93 \pm 0.09$ / $81.24 \pm 0.09$ | $75.01 \pm 0.08$ / $61.43 \pm 0.21$ / $88.27 \pm 0.23$ | $45.09 \pm 0.24$ / $17.67 \pm 0.67$ / $95.30 \pm 0.01$ | $31.52 \pm 1.30$ / $5.33 \pm 0.94$ |
| PSPNet | ResNet101 | $86.23 \pm 0.05$ / $77.84 \pm 0.11$ / $81.10 \pm 0.16$ | $74.85 \pm 0.14$ / $61.11 \pm 0.28$ / $88.13 \pm 0.23$ | $45.39 \pm 0.74$ / $15.95 \pm 1.32$ / $95.25 \pm 0.05$ | $31.36 \pm 1.48$ / $3.33 \pm 1.70$ |
| UNet | ResNet101 | $85.58 \pm 0.09$ / $76.85 \pm 0.10$ / $74.94 \pm 0.33$ | $73.74 \pm 0.16$ / $59.49 \pm 0.24$ / $83.62 \pm 0.20$ | $43.90 \pm 0.82$ / $16.00 \pm 1.34$ / $94.06 \pm 0.09$ | $29.84 \pm 1.25$ / $3.33 \pm 1.25$ |

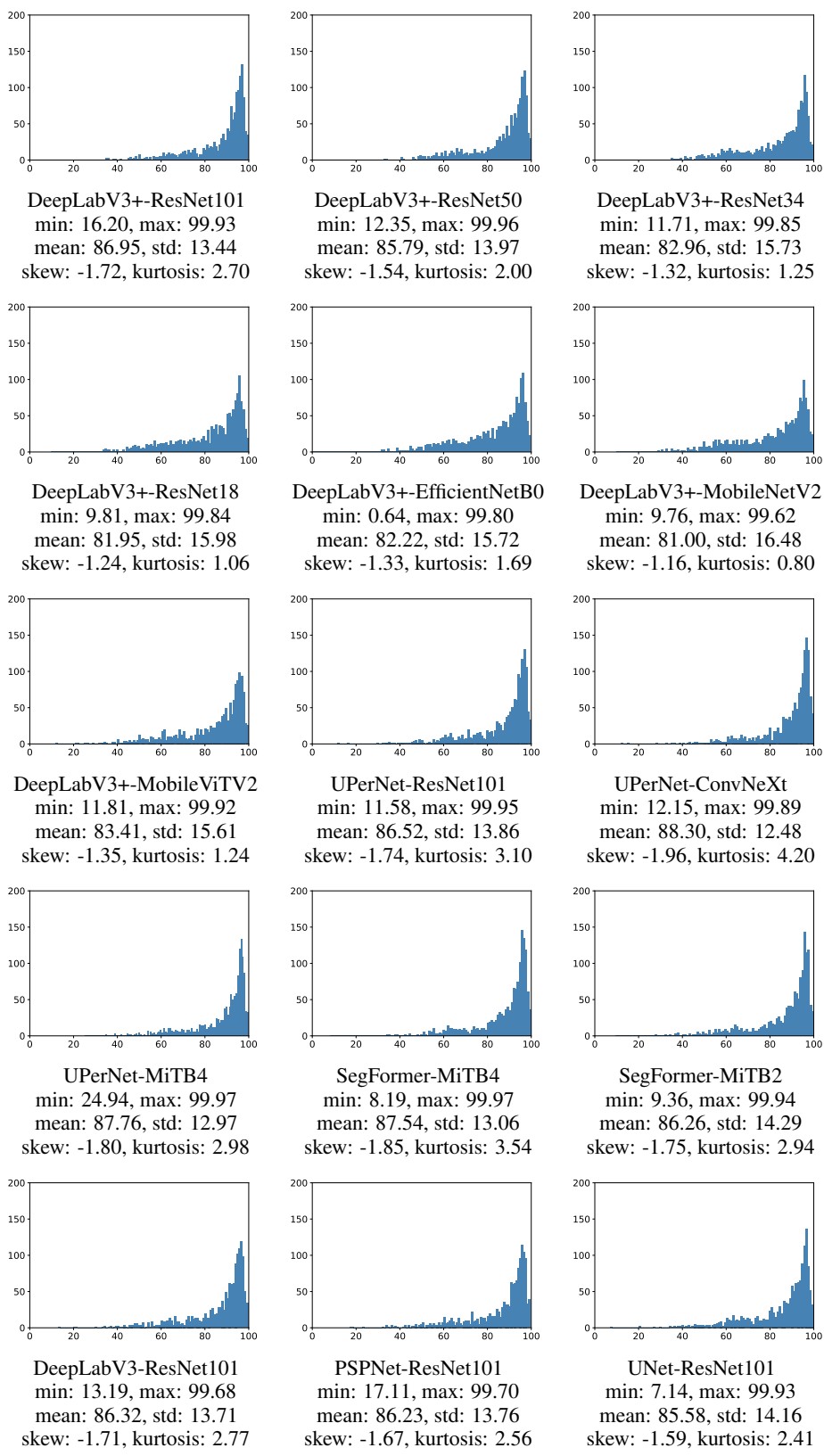

Figure 34: Histograms: PASCAL VOC.

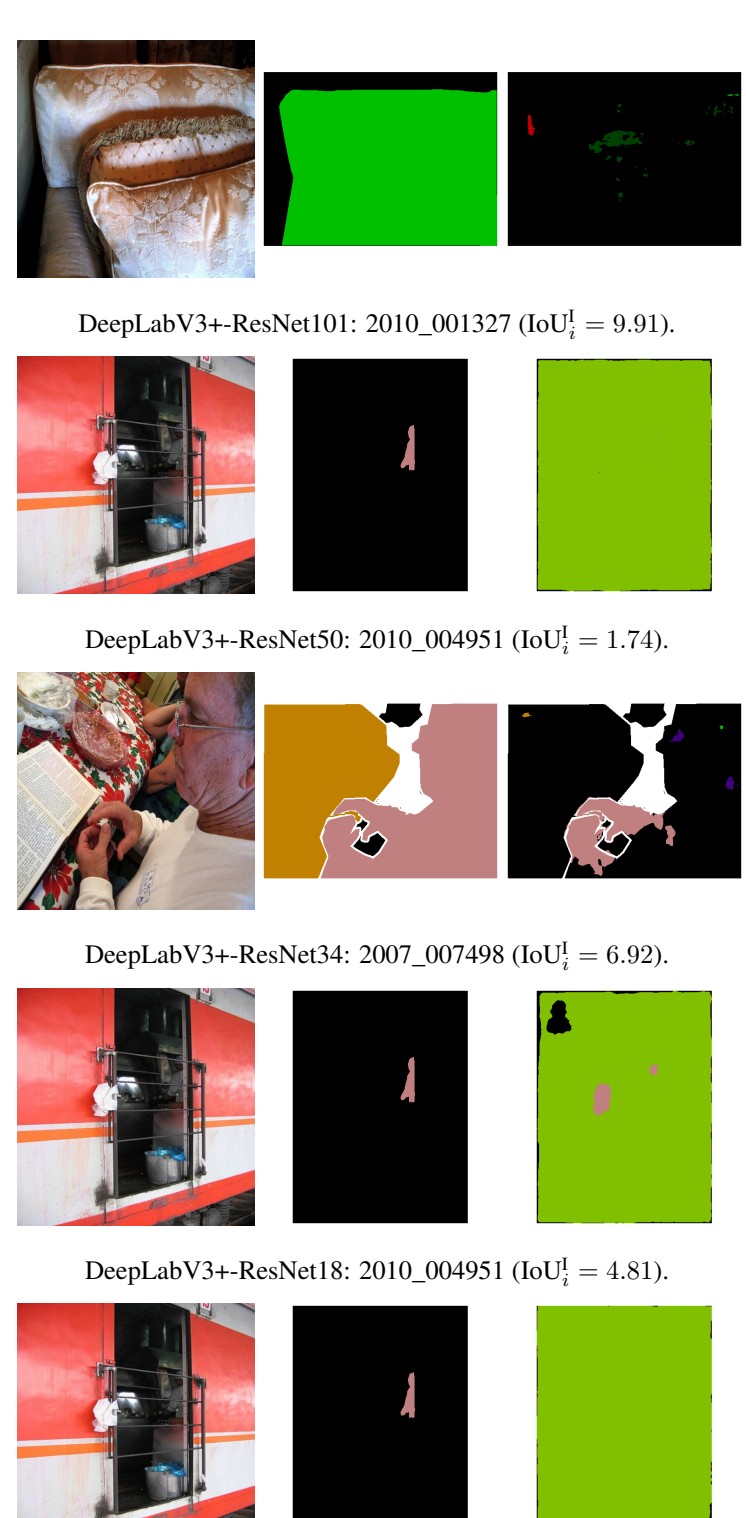

DeepLabV3+-ResNet101: 2010_001327 (IoU$_i^I$ = 9.91).

DeepLabV3+-ResNet50: 2010_004951 (IoU$_i^I$ = 1.74).

DeepLabV3+-ResNet34: 2007_007498 (IoU$_i^I$ = 6.92).

DeepLabV3+-ResNet18: 2010_004951 (IoU$_i^I$ = 4.81).

DeepLabV3+-EfficientNetB0: 2010_004951 (IoU$_i^I$ = 0.35).

Figure 35: Worst-case images: PASCAL VOC 1 / 3. Left: image. Middle: label. Right: prediction.

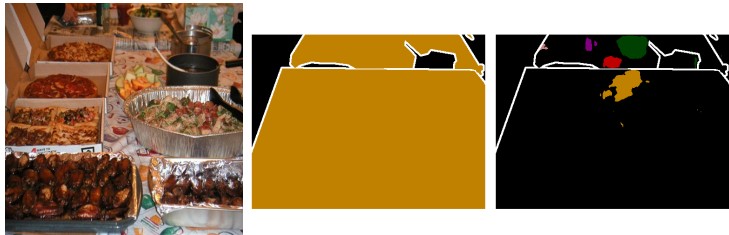

DeepLabV3+-MobileNetV2: 2008_008051 ($\text{IoU}_i^\text{I} = 5.63$).

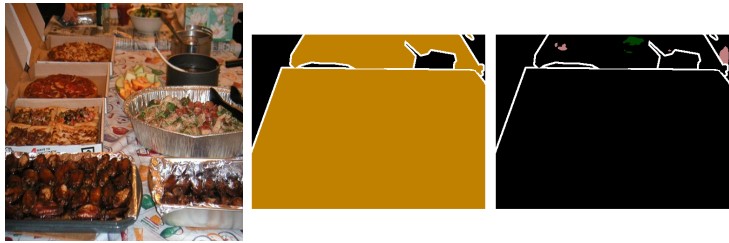

DeepLabV3+-MobileViTV2: 2008_008051 ($\text{IoU}_i^\text{I} = 4.42$).

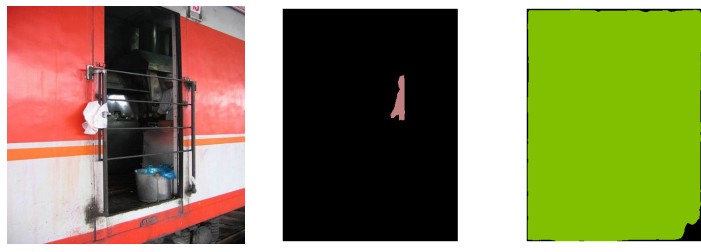

UPerNet-ResNet101: 2010_004951 ($\text{IoU}_i^\text{I} = 1.57$).

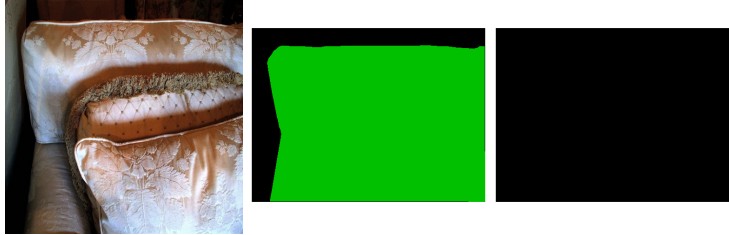

UPerNet-ConvNeXt: 2010_001327 ($\text{IoU}_i^\text{I} = 9.69$).

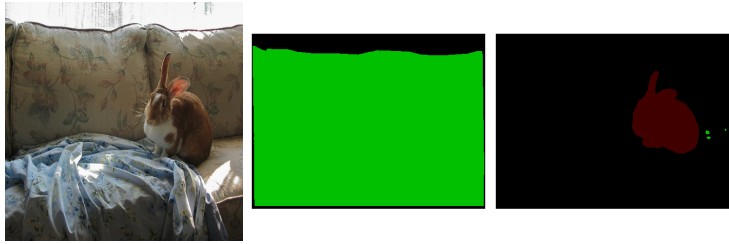

UPerNet-MiTB4: 2011_003182 ($\text{IoU}_i^\text{I} = 7.39$).

Figure 36: Worst-case images: PASCAL VOC 2 / 3. Left: image. Middle: label. Right: prediction.

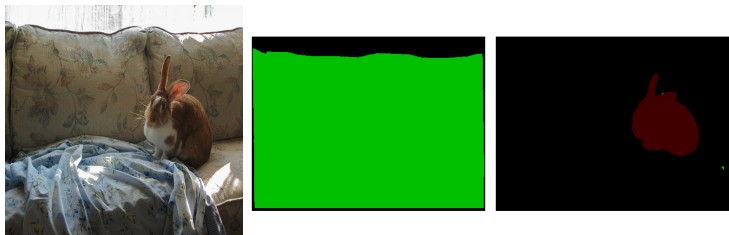

SegFormer-MiTB4: 2011_003182 (IoU$_i^I$ = 7.36).

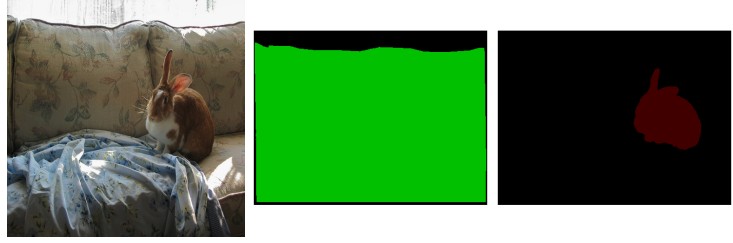

SegFormer-MiTB2: 2011_003182 (IoU$_i^I$ = 7.34).

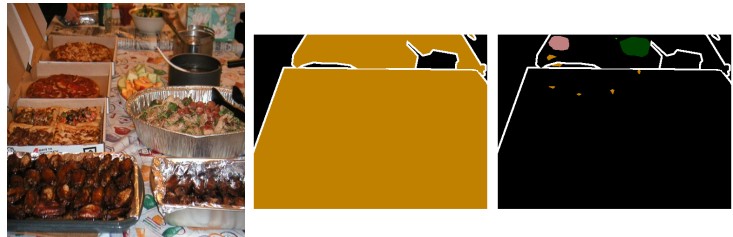

DeepLabV3-ResNet101: 2008_008051 (IoU$_i^I$ = 4.72).

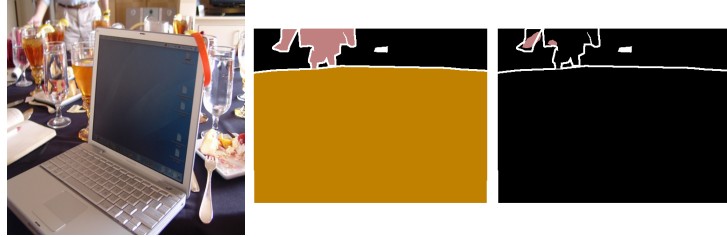

PSPNet-ResNet101: 2008_000763 (IoU$_i^I$ = 9.23).

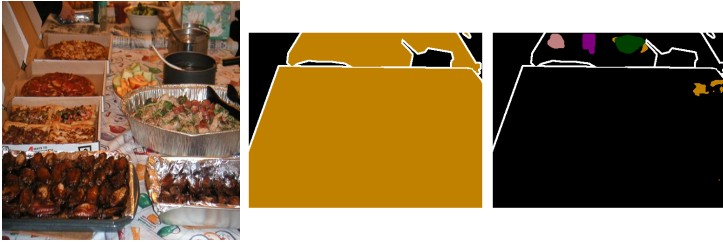

UNet-ResNet101: 2008_008051 (IoU$_i^I$ = 5.04).

Figure 37: Worst-case images: PASCAL VOC 3 / 3. Left: image. Middle: label. Right: prediction.

## G.9 PASCAL Context

Table 22: Results: PASCAL Context. Red: the best for each metric. Green: the worst for each metric.

| Method | Backbone | $\text{mIoU}^I$ / $\text{mIoU}^C$ / $\text{mIoU}^D$ | $\text{mIoU}^{I^{\bar{q}}}$ / $\text{mIoU}^{C^{\bar{q}}}$ / mAcc | $\text{mIoU}^{I^5}$ / $\text{mIoU}^{C^5}$ / Acc | $\text{mIoU}^{I^1}$ / $\text{mIoU}^{C^1}$ |
|---|---|---|---|---|---|
| DeepLabV3+ | ResNet101 | $67.16 \pm 0.04$ | $51.85 \pm 0.03$ | $27.15 \pm 0.19$ | $14.81 \pm 1.23$ |
| | | $55.89 \pm 0.07$ | $34.93 \pm 0.06$ | $3.24 \pm 0.29$ | $0.00 \pm 0.00$ |
| | | $56.49 \pm 0.10$ | $67.66 \pm 0.27$ | $81.76 \pm 0.09$ | |
| DeepLabV3+ | ResNet50 | $65.42 \pm 0.07$ | $49.91 \pm 0.08$ | $24.97 \pm 0.14$ | $12.35 \pm 0.75$ |
| | | $53.95 \pm 0.18$ | $33.12 \pm 0.11$ | $2.57 \pm 0.05$ | $0.00 \pm 0.00$ |
| | | $54.50 \pm 0.08$ | $65.74 \pm 0.23$ | $80.57 \pm 0.07$ | |
| DeepLabV3+ | ResNet34 | $61.92 \pm 0.13$ | $45.52 \pm 0.19$ | $18.58 \pm 0.63$ | $3.60 \pm 0.97$ |
| | | $49.49 \pm 0.20$ | $27.84 \pm 0.10$ | $0.40 \pm 0.04$ | $0.00 \pm 0.00$ |
| | | $48.82 \pm 0.07$ | $60.90 \pm 0.15$ | $77.45 \pm 0.08$ | |
| DeepLabV3+ | ResNet18 | $60.39 \pm 0.11$ | $43.87 \pm 0.12$ | $16.93 \pm 0.40$ | $2.76 \pm 0.69$ |
| | | $47.37 \pm 0.26$ | $26.09 \pm 0.26$ | $0.45 \pm 0.03$ | $0.00 \pm 0.00$ |
| | | $46.59 \pm 0.33$ | $58.58 \pm 0.39$ | $76.40 \pm 0.12$ | |
| DeepLabV3+ | EfficientNetB0 | $60.95 \pm 0.13$ | $44.33 \pm 0.24$ | $17.24 \pm 0.65$ | $2.85 \pm 0.92$ |
| | | $48.13 \pm 0.05$ | $26.56 \pm 0.11$ | $0.42 \pm 0.08$ | $0.00 \pm 0.00$ |
| | | $47.90 \pm 0.18$ | $60.02 \pm 0.04$ | $77.31 \pm 0.16$ | |
| DeepLabV3+ | MobileNetV2 | $59.30 \pm 0.10$ | $42.89 \pm 0.14$ | $16.84 \pm 0.33$ | $3.19 \pm 0.29$ |
| | | $46.00 \pm 0.18$ | $24.87 \pm 0.23$ | $0.22 \pm 0.04$ | $0.00 \pm 0.00$ |
| | | $45.64 \pm 0.11$ | $57.77 \pm 0.21$ | $75.72 \pm 0.08$ | |
| DeepLabV3+ | MobileViTV2 | $62.48 \pm 0.26$ | $45.77 \pm 0.34$ | $18.91 \pm 0.62$ | $5.23 \pm 0.78$ |
| | | $49.96 \pm 0.21$ | $28.15 \pm 0.18$ | $0.39 \pm 0.12$ | $0.00 \pm 0.00$ |
| | | $50.38 \pm 0.36$ | $61.95 \pm 0.40$ | $78.53 \pm 0.21$ | |
| UPerNet | ResNet101 | $66.42 \pm 0.10$ | $50.85 \pm 0.09$ | $25.78 \pm 0.36$ | $13.32 \pm 0.92$ |
| | | $55.04 \pm 0.10$ | $34.48 \pm 0.10$ | $3.05 \pm 0.07$ | $0.00 \pm 0.00$ |
| | | $55.65 \pm 0.10$ | $66.19 \pm 0.17$ | $81.24 \pm 0.06$ | |
| UPerNet | ConvNeXt | $70.36 \pm 0.09$ | $55.78 \pm 0.07$ | $31.27 \pm 0.29$ | $17.88 \pm 0.80$ |
| | | $60.17 \pm 0.02$ | $39.11 \pm 0.04$ | $6.05 \pm 0.11$ | $0.00 \pm 0.00$ |
| | | $61.33 \pm 0.06$ | $71.84 \pm 0.06$ | $84.09 \pm 0.06$ | |
| UPerNet | MiTB4 | $69.83 \pm 0.03$ | $54.57 \pm 0.02$ | $29.50 \pm 0.20$ | $16.10 \pm 0.35$ |
| | | $58.55 \pm 0.12$ | $37.15 \pm 0.14$ | $3.83 \pm 0.34$ | $0.00 \pm 0.00$ |
| | | $60.50 \pm 0.13$ | $70.90 \pm 0.11$ | $84.07 \pm 0.10$ | |
| SegFormer | MiTB4 | $69.54 \pm 0.05$ | $54.40 \pm 0.07$ | $29.72 \pm 0.23$ | $16.11 \pm 0.90$ |
| | | $58.29 \pm 0.14$ | $36.97 \pm 0.22$ | $3.63 \pm 0.28$ | $0.00 \pm 0.00$ |
| | | $60.21 \pm 0.14$ | $70.75 \pm 0.20$ | $83.90 \pm 0.04$ | |
| SegFormer | MiTB2 | $67.24 \pm 0.03$ | $51.69 \pm 0.03$ | $26.39 \pm 0.07$ | $12.76 \pm 0.30$ |
| | | $55.26 \pm 0.04$ | $34.02 \pm 0.04$ | $2.06 \pm 0.01$ | $0.00 \pm 0.00$ |
| | | $56.73 \pm 0.06$ | $67.33 \pm 0.10$ | $82.20 \pm 0.02$ | |
| DeepLabV3 | ResNet101 | $65.84 \pm 0.04$ | $50.57 \pm 0.11$ | $26.41 \pm 0.46$ | $14.25 \pm 0.71$ |
| | | $54.33 \pm 0.17$ | $33.94 \pm 0.13$ | $2.96 \pm 0.12$ | $0.00 \pm 0.00$ |
| | | $55.80 \pm 0.08$ | $66.18 \pm 0.22$ | $81.41 \pm 0.07$ | |
| PSPNet | ResNet101 | $65.86 \pm 0.11$ | $50.54 \pm 0.14$ | $26.09 \pm 0.10$ | $13.81 \pm 0.43$ |
| | | $54.35 \pm 0.03$ | $33.89 \pm 0.08$ | $3.03 \pm 0.10$ | $0.00 \pm 0.00$ |
| | | $55.52 \pm 0.11$ | $66.03 \pm 0.07$ | $81.23 \pm 0.06$ | |
| UNet | ResNet101 | $65.55 \pm 0.07$ | $50.60 \pm 0.03$ | $26.27 \pm 0.40$ | $14.08 \pm 0.90$ |
| | | $53.88 \pm 0.15$ | $33.07 \pm 0.09$ | $2.80 \pm 0.07$ | $0.00 \pm 0.00$ |
| | | $51.11 \pm 0.33$ | $63.92 \pm 0.41$ | $78.74 \pm 0.08$ | |

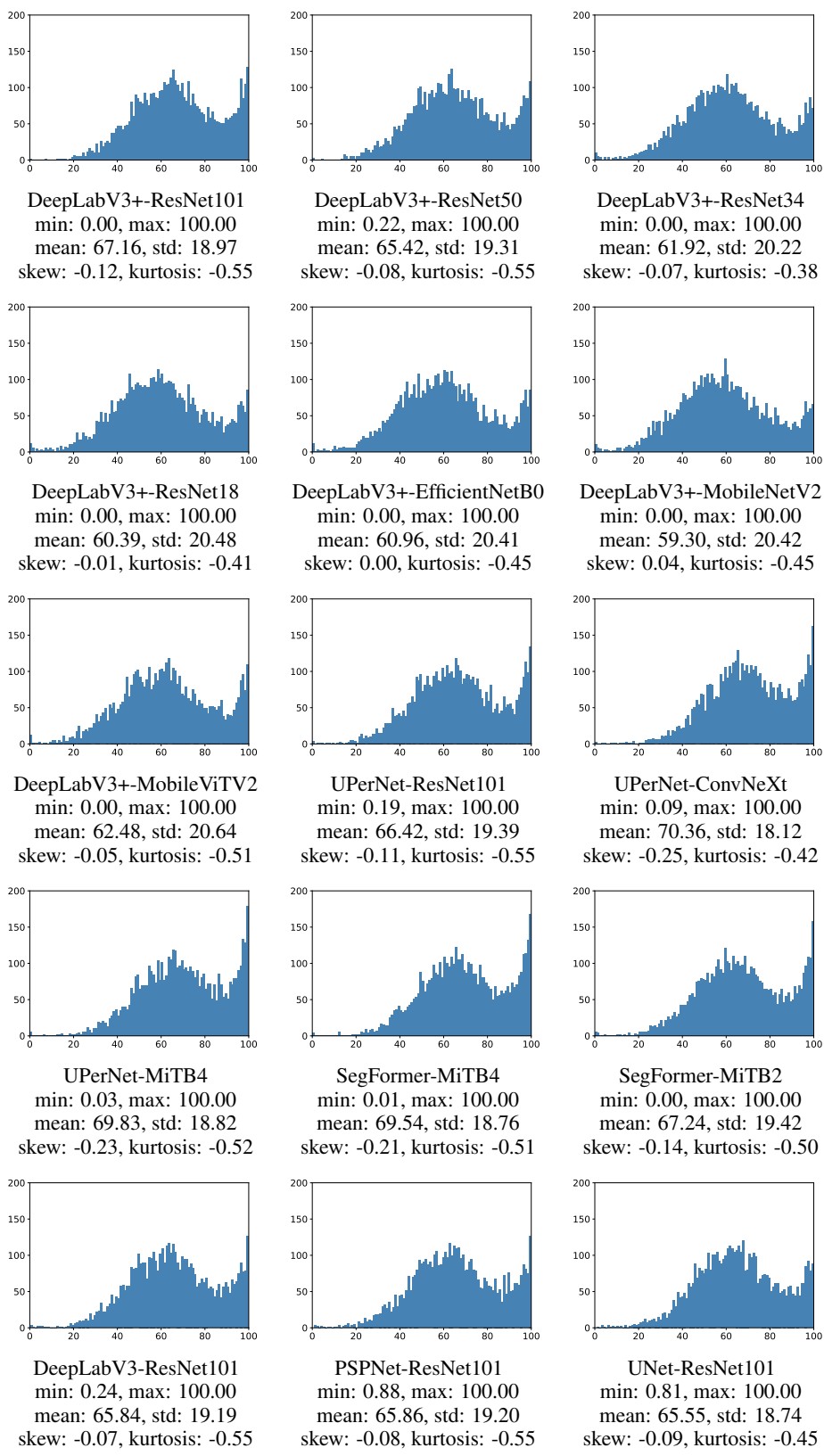

Figure 38: Histograms: PASCAL Context.

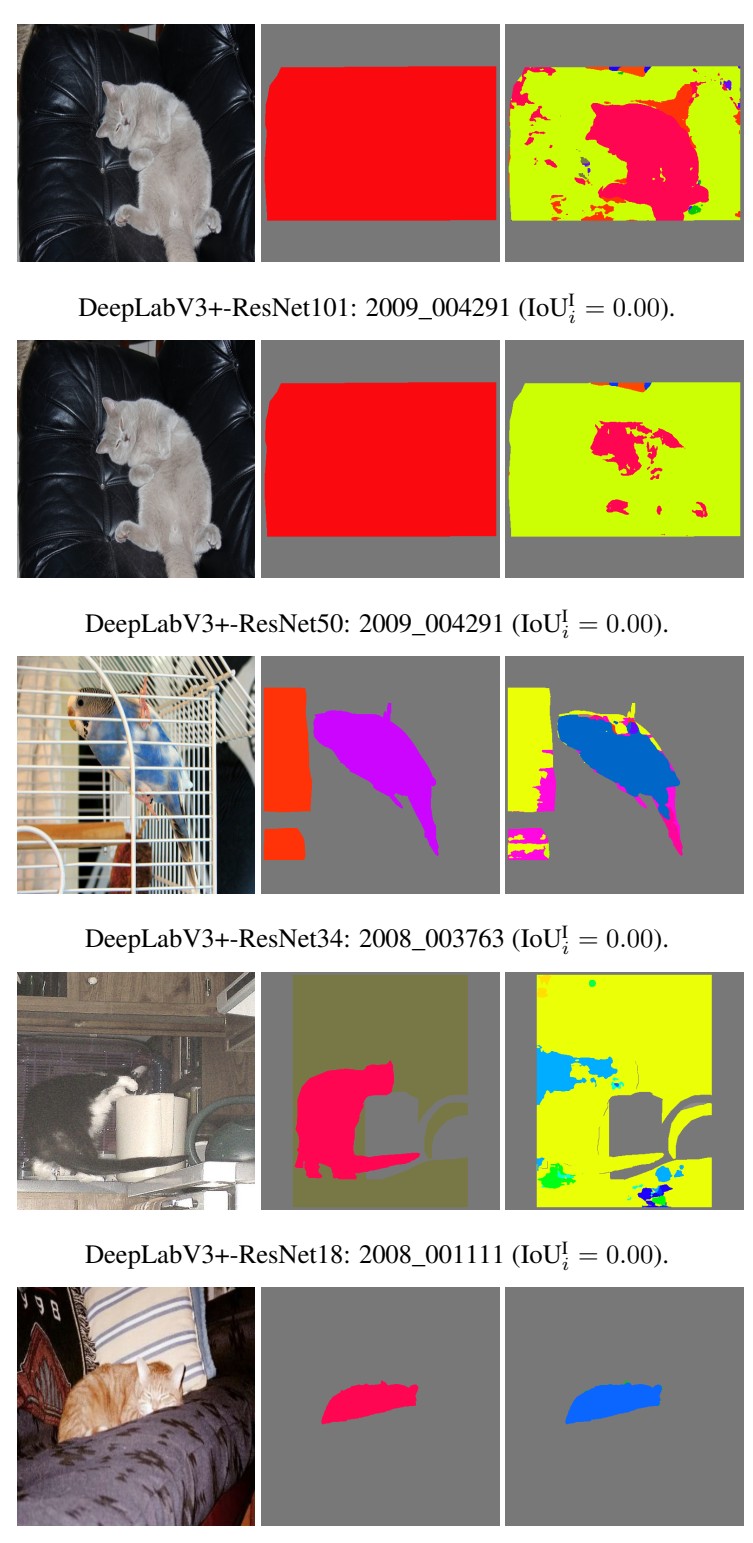

DeepLabV3+-ResNet101: 2009_004291 (IoU$_i^I$ = 0.00).

DeepLabV3+-ResNet50: 2009_004291 (IoU$_i^I$ = 0.00).

DeepLabV3+-ResNet34: 2008_003763 (IoU$_i^I$ = 0.00).

DeepLabV3+-ResNet18: 2008_001111 (IoU$_i^I$ = 0.00).

DeepLabV3+-EfficientNetB0: 2008_000670 (IoU$_i^I$ = 0.00).

Figure 39: Worst-case images: PASCAL Context 1 / 3. Left: image. Middle: label. Right: prediction.

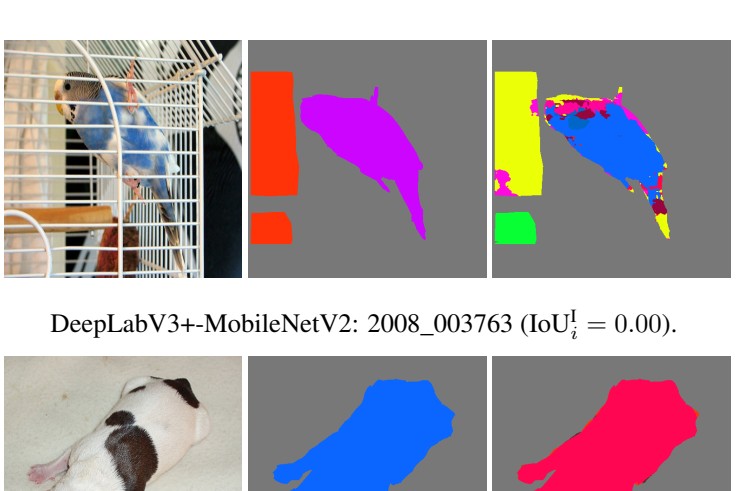

DeepLabV3+-MobileNetV2: 2008_003763 ($\text{IoU}_i^\text{I} = 0.00$).

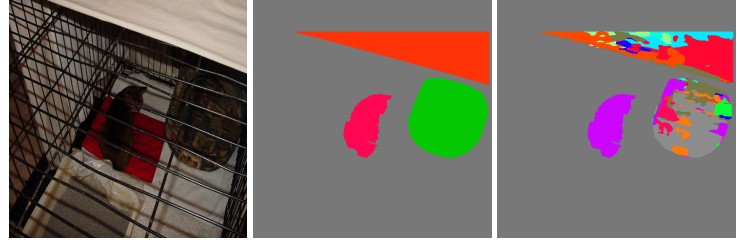

DeepLabV3+-MobileViTV2: 2008_005748 ($\text{IoU}_i^\text{I} = 0.00$).

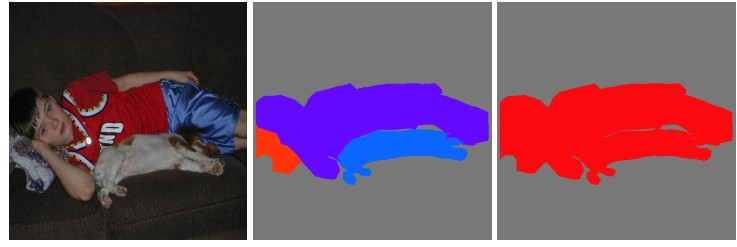

UPerNet-ResNet101: 2008_004736 ($\text{IoU}_i^\text{I} = 0.00$).

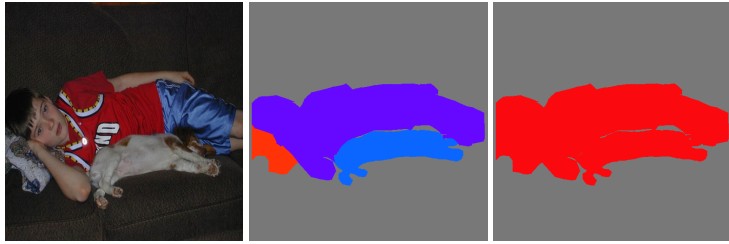

UPerNet-ConvNeXt: 2008_005046 ($\text{IoU}_i^\text{I} = 0.00$).

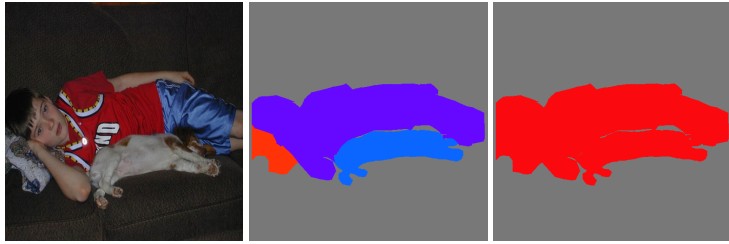

UPerNet-MiTB4: 2008_005046 ($\text{IoU}_i^\text{I} = 0.00$).

Figure 40: Worst-case images: PASCAL Context 2 / 3. Left: image. Middle: label. Right: prediction.

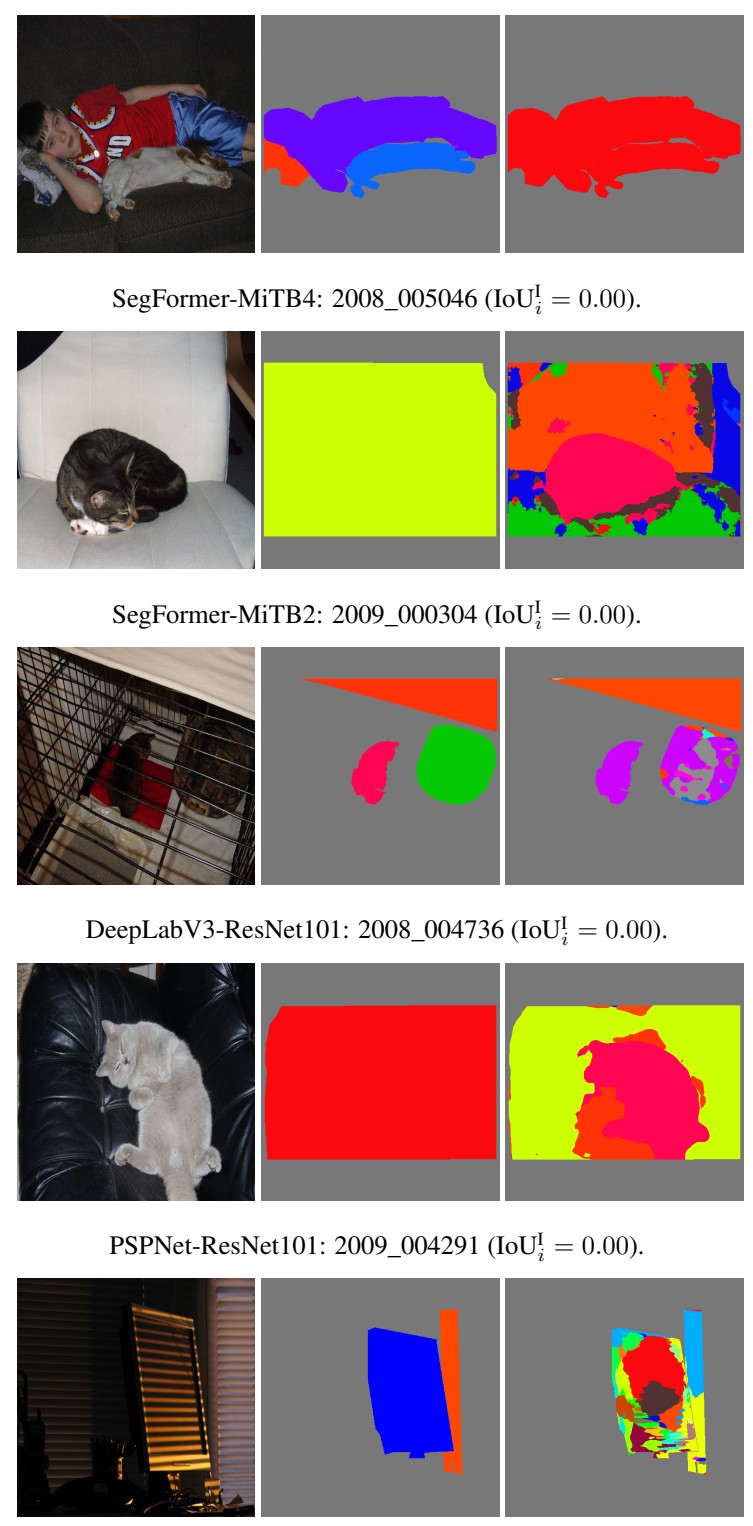

SegFormer-MiTB4: 2008_005046 (IoU$_i^I$ = 0.00).

SegFormer-MiTB2: 2009_000304 (IoU$_i^I$ = 0.00).

DeepLabV3-ResNet101: 2008_004736 (IoU$_i^I$ = 0.00).

PSPNet-ResNet101: 2009_004291 (IoU$_i^I$ = 0.00).

UNet-ResNet101: 2010_000379 (IoU$_i^I$ = 0.08).

Figure 41: Worst-case images: PASCAL Context 3 / 3. Left: image. Middle: label. Right: prediction.

## G.10 DeepGlobe Land

Table 23: Results: DeepGlobe Land. Red: the best for each metric. Green: the worst for each metric.

| Method | Backbone | $mIoU^I$ / $mIoU^C$ / $mIoU^D$ | $mIoU^{I^{\bar{q}}}$ / $mIoU^{C^{\bar{q}}}$ / mAcc | $mIoU^{I^5}$ / $mIoU^{C^5}$ / Acc | $mIoU^{I^1}$ / $mIoU^{C^1}$ |
|---|---|---|---|---|---|
| DeepLabV3+ | ResNet101 | $57.26 \pm 0.46$ | $45.96 \pm 0.49$ | $28.81 \pm 0.72$ | $21.66 \pm 1.18$ |
| | | $53.76 \pm 0.40$ | $31.62 \pm 0.28$ | $2.80 \pm 0.18$ | $0.00 \pm 0.00$ |
| | | $71.27 \pm 0.21$ | $82.80 \pm 0.16$ | $87.21 \pm 0.08$ | |
| DeepLabV3+ | ResNet50 | $57.13 \pm 0.12$ | $46.02 \pm 0.29$ | $30.83 \pm 1.13$ | $22.38 \pm 2.81$ |
| | | $53.63 \pm 0.17$ | $31.48 \pm 0.18$ | $2.53 \pm 0.02$ | $0.00 \pm 0.00$ |
| | | $71.44 \pm 0.07$ | $82.94 \pm 0.15$ | $87.23 \pm 0.09$ | |
| DeepLabV3+ | ResNet34 | $56.84 \pm 0.32$ | $45.83 \pm 0.48$ | $28.28 \pm 1.87$ | $18.53 \pm 1.97$ |
| | | $53.50 \pm 0.36$ | $31.42 \pm 0.44$ | $2.39 \pm 0.17$ | $0.00 \pm 0.00$ |
| | | $70.76 \pm 0.24$ | $82.93 \pm 0.24$ | $86.85 \pm 0.15$ | |
| DeepLabV3+ | ResNet18 | $56.37 \pm 0.05$ | $45.34 \pm 0.13$ | $28.60 \pm 0.48$ | $19.41 \pm 0.30$ |
| | | $53.01 \pm 0.09$ | $30.87 \pm 0.14$ | $2.64 \pm 0.16$ | $0.00 \pm 0.00$ |
| | | $70.76 \pm 0.12$ | $82.66 \pm 0.24$ | $86.81 \pm 0.05$ | |
| DeepLabV3+ | EfficientNetB0 | $58.15 \pm 0.10$ | $47.16 \pm 0.13$ | $30.29 \pm 0.20$ | $18.44 \pm 0.09$ |
| | | $55.03 \pm 0.07$ | $33.14 \pm 0.22$ | $2.84 \pm 0.06$ | $0.00 \pm 0.00$ |
| | | $71.67 \pm 0.21$ | $83.40 \pm 0.22$ | $87.52 \pm 0.13$ | |
| DeepLabV3+ | MobileNetV2 | $56.61 \pm 0.60$ | $45.45 \pm 0.77$ | $28.86 \pm 1.23$ | $19.43 \pm 0.69$ |
| | | $53.25 \pm 0.72$ | $31.28 \pm 0.92$ | $2.69 \pm 0.20$ | $0.00 \pm 0.00$ |
| | | $70.34 \pm 0.39$ | $81.87 \pm 0.64$ | $86.79 \pm 0.19$ | |
| DeepLabV3+ | MobileViTV2 | $58.64 \pm 0.29$ | $47.48 \pm 0.37$ | $31.49 \pm 1.15$ | $23.23 \pm 3.07$ |
| | | $55.44 \pm 0.42$ | $33.25 \pm 0.36$ | $2.74 \pm 0.15$ | $0.00 \pm 0.00$ |
| | | $72.86 \pm 0.51$ | $83.69 \pm 0.54$ | $87.97 \pm 0.25$ | |
| UPerNet | ResNet101 | $56.47 \pm 0.10$ | $44.93 \pm 0.25$ | $26.45 \pm 0.90$ | $17.53 \pm 4.38$ |
| | | $52.72 \pm 0.20$ | $30.42 \pm 0.21$ | $2.31 \pm 0.12$ | $0.00 \pm 0.00$ |
| | | $70.73 \pm 0.17$ | $81.81 \pm 0.15$ | $86.99 \pm 0.03$ | |
| UPerNet | ConvNeXt | $58.85 \pm 0.21$ | $47.69 \pm 0.31$ | $30.95 \pm 0.52$ | $19.93 \pm 0.80$ |
| | | $55.83 \pm 0.29$ | $33.68 \pm 0.32$ | $2.75 \pm 0.24$ | $0.00 \pm 0.00$ |
| | | $72.91 \pm 0.18$ | $83.18 \pm 0.29$ | $88.01 \pm 0.10$ | |
| UPerNet | MiTB4 | $59.31 \pm 0.20$ | $47.99 \pm 0.25$ | $31.13 \pm 1.04$ | $21.49 \pm 2.50$ |
| | | $56.42 \pm 0.28$ | $34.35 \pm 0.40$ | $3.28 \pm 0.06$ | $0.00 \pm 0.00$ |
| | | $73.21 \pm 0.11$ | $83.75 \pm 0.14$ | $88.15 \pm 0.10$ | |
| SegFormer | MiTB4 | $59.38 \pm 0.24$ | $48.20 \pm 0.28$ | $31.93 \pm 0.35$ | $21.28 \pm 2.94$ |
| | | $56.44 \pm 0.30$ | $34.26 \pm 0.36$ | $3.23 \pm 0.03$ | $0.00 \pm 0.00$ |
| | | $73.36 \pm 0.25$ | $84.03 \pm 0.47$ | $88.17 \pm 0.14$ | |
| SegFormer | MiTB2 | $59.41 \pm 0.35$ | $48.08 \pm 0.38$ | $30.98 \pm 0.60$ | $20.27 \pm 0.98$ |
| | | $56.21 \pm 0.28$ | $34.14 \pm 0.27$ | $3.24 \pm 0.20$ | $0.00 \pm 0.00$ |
| | | $73.22 \pm 0.22$ | $83.69 \pm 0.13$ | $88.18 \pm 0.11$ | |
| DeepLabV3 | ResNet101 | $57.31 \pm 0.13$ | $46.07 \pm 0.24$ | $28.46 \pm 1.26$ | $20.32 \pm 1.43$ |
| | | $53.80 \pm 0.09$ | $31.70 \pm 0.10$ | $2.53 \pm 0.15$ | $0.00 \pm 0.00$ |
| | | $71.39 \pm 0.22$ | $82.53 \pm 0.25$ | $87.30 \pm 0.20$ | |
| PSPNet | ResNet101 | $57.03 \pm 0.63$ | $45.84 \pm 0.67$ | $29.44 \pm 0.94$ | $21.82 \pm 2.49$ |
| | | $53.45 \pm 0.71$ | $31.38 \pm 0.71$ | $2.48 \pm 0.15$ | $0.00 \pm 0.00$ |
| | | $71.02 \pm 0.60$ | $82.39 \pm 0.75$ | $87.07 \pm 0.30$ | |
| UNet | ResNet101 | $56.65 \pm 0.08$ | $45.31 \pm 0.13$ | $28.04 \pm 0.28$ | $19.69 \pm 0.48$ |
| | | $53.13 \pm 0.07$ | $30.98 \pm 0.09$ | $2.48 \pm 0.17$ | $0.00 \pm 0.00$ |
| | | $70.71 \pm 0.30$ | $82.44 \pm 0.05$ | $86.97 \pm 0.14$ | |

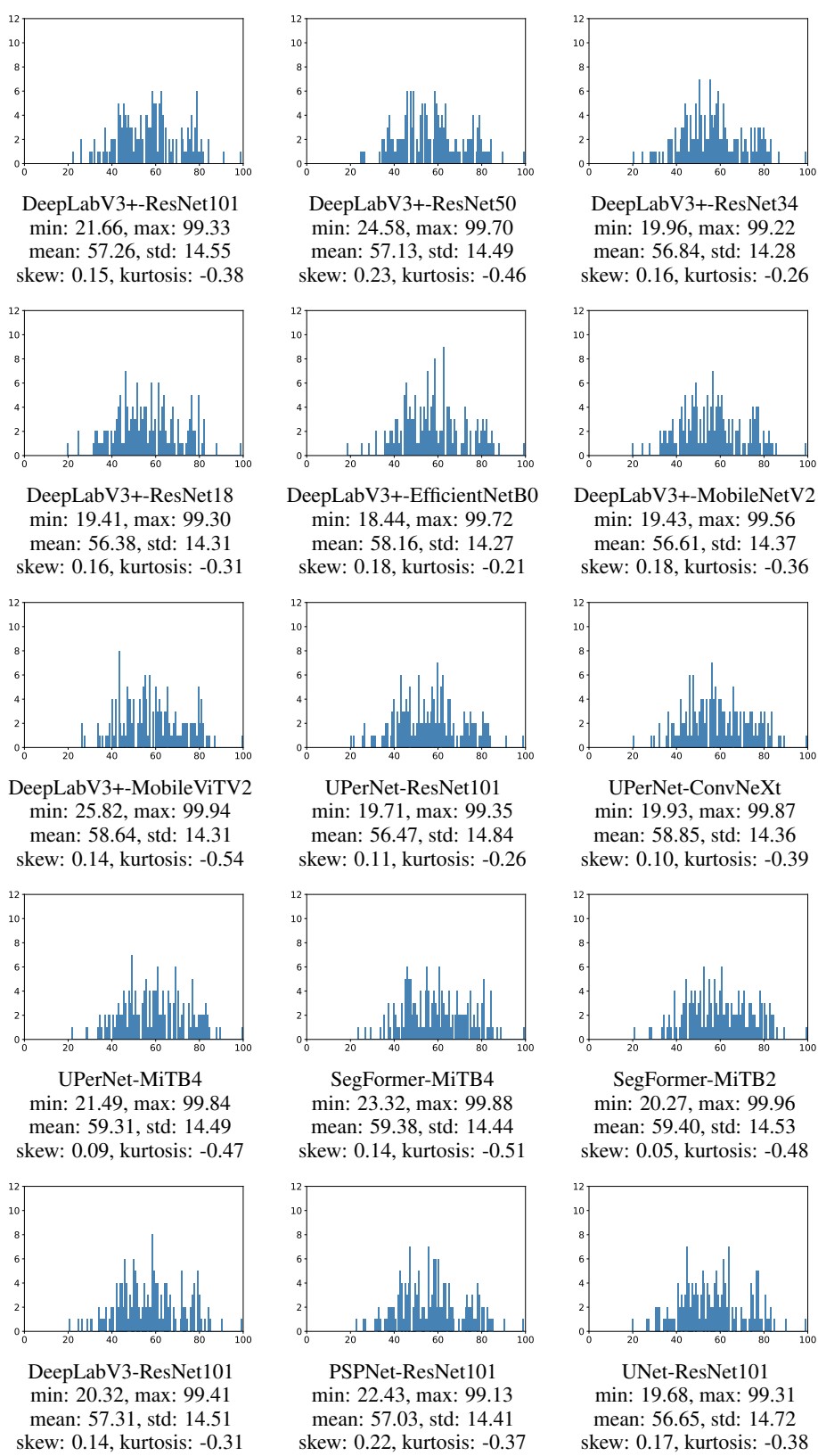

Figure 42: Histograms: DeepGlobe Land.

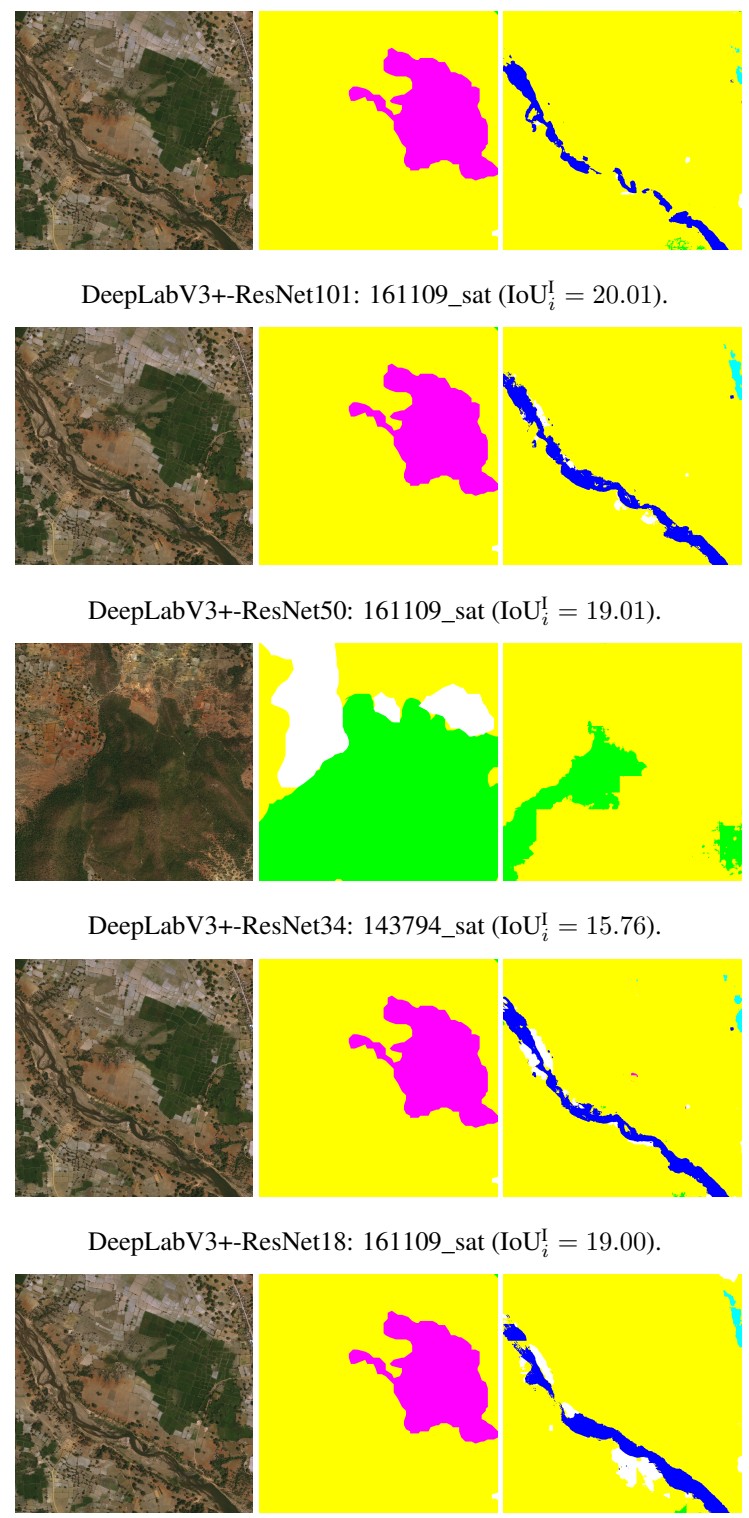

DeepLabV3+-ResNet101: 161109_sat (IoU$_i^I$ = 20.01).

DeepLabV3+-ResNet50: 161109_sat (IoU$_i^I$ = 19.01).

DeepLabV3+-ResNet34: 143794_sat (IoU$_i^I$ = 15.76).

DeepLabV3+-ResNet18: 161109_sat (IoU$_i^I$ = 19.00).

DeepLabV3+-EfficientNetB0: 161109_sat (IoU$_i^I$ = 18.34).

Figure 43: Worst-case images: DeepGlobe Land 1 / 3. Left: image. Middle: label. Right: prediction.

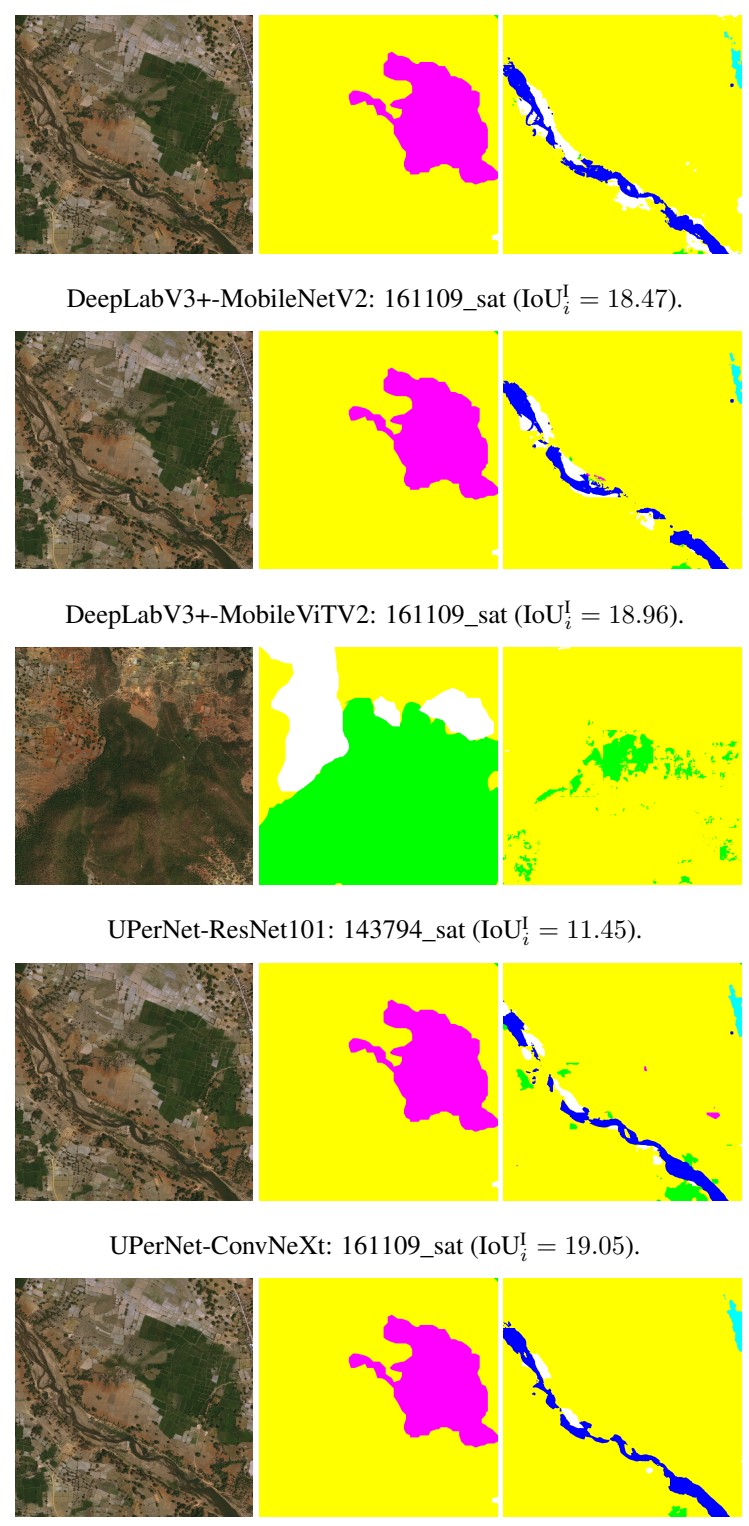

DeepLabV3+-MobileNetV2: 161109_sat ($\text{IoU}_i^I = 18.47$).

DeepLabV3+-MobileViTV2: 161109_sat ($\text{IoU}_i^I = 18.96$).

UPerNet-ResNet101: 143794_sat ($\text{IoU}_i^I = 11.45$).

UPerNet-ConvNeXt: 161109_sat ($\text{IoU}_i^I = 19.05$).

UPerNet-MiTB4: 161109_sat ($\text{IoU}_i^I = 19.22$).

Figure 44: Worst-case images: DeepGlobe Land 2 / 3. Left: image. Middle: label. Right: prediction.

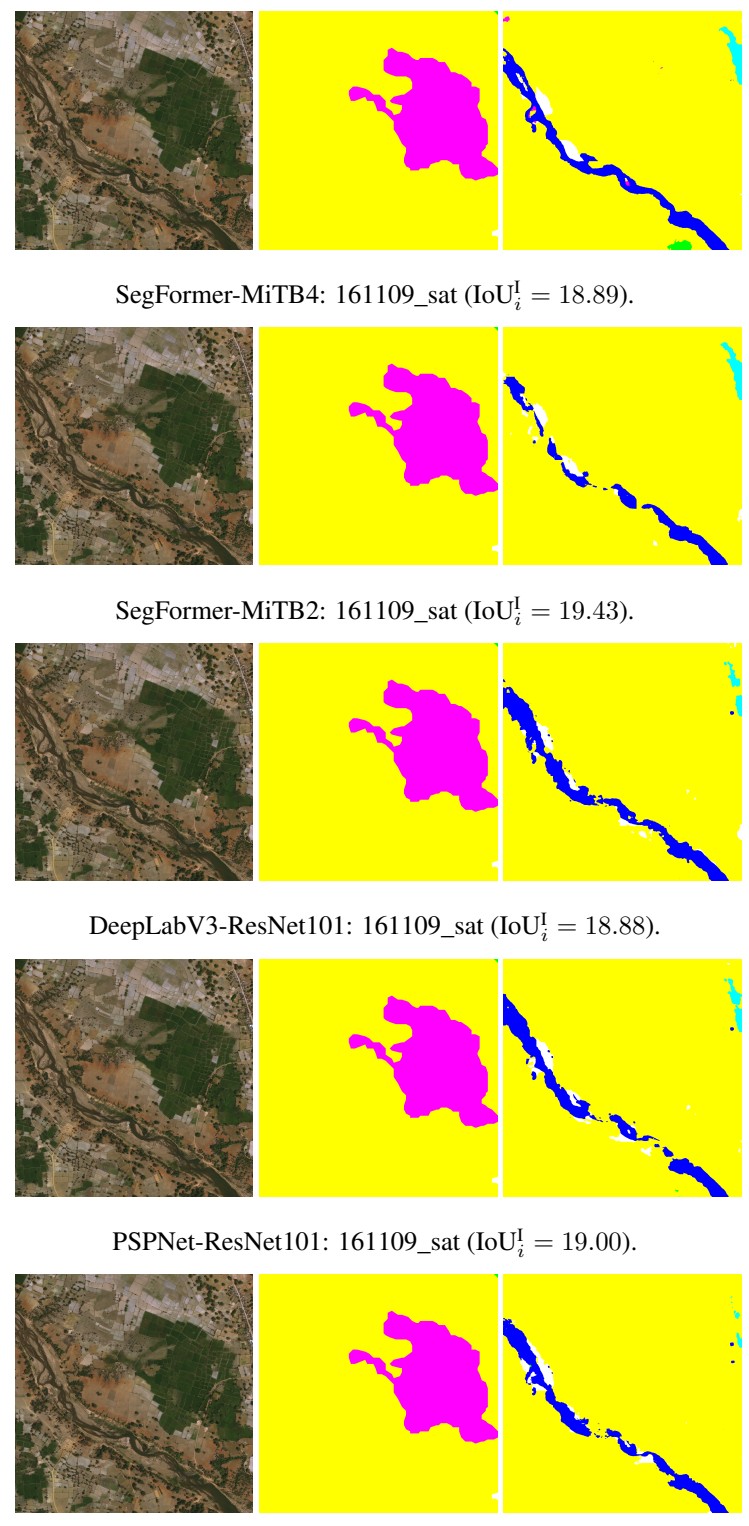

SegFormer-MiTB4: 161109_sat ($IoU_i^I = 18.89$).

SegFormer-MiTB2: 161109_sat ($IoU_i^I = 19.43$).

DeepLabV3-ResNet101: 161109_sat ($IoU_i^I = 18.88$).

PSPNet-ResNet101: 161109_sat ($IoU_i^I = 19.00$).

UNet-ResNet101: 161109_sat ($IoU_i^I = 19.15$).

Figure 45: Worst-case images: DeepGlobe Land 3 / 3. Left: image. Middle: label. Right: prediction.

Table 24: Results: DeepGlobe Road. Red: the best for each metric. Green: the worst for each metric.

| Method | Backbone | $\text{mIoU}^{\text{I}}$ / $\text{mIoU}^{\text{C}}$ / $\text{mIoU}^{\text{D}}$ | $\text{mIoU}^{\text{I}^{\bar{q}}}$ / $\text{mIoU}^{\text{C}^{\bar{q}}}$ / mAcc | $\text{mIoU}^{\text{I}^{5}}$ / $\text{mIoU}^{\text{C}^{5}}$ / Acc | $\text{mIoU}^{\text{I}^{1}}$ / $\text{mIoU}^{\text{C}^{1}}$ |
|---|---|---|---|---|---|
| DeepLabV3+ | ResNet101 | 64.59 ± 0.88 | 53.15 ± 1.19 | 26.09 ± 2.10 | 8.50 ± 1.45 |
|  |  | 64.59 ± 0.88 | 53.15 ± 1.19 | 26.09 ± 2.10 | 8.00 ± 1.63 |
|  |  | 65.60 ± 0.45 | 88.68 ± 0.52 | 98.22 ± 0.01 |  |
| DeepLabV3+ | ResNet50 | 63.86 ± 0.07 | 52.56 ± 0.08 | 25.95 ± 0.36 | 6.67 ± 1.47 |
|  |  | 63.86 ± 0.07 | 52.56 ± 0.08 | 25.95 ± 0.36 | 6.00 ± 1.41 |
|  |  | 64.59 ± 0.08 | 88.34 ± 0.14 | 98.16 ± 0.01 |  |
| DeepLabV3+ | ResNet34 | 63.49 ± 0.17 | 51.77 ± 0.22 | 23.13 ± 0.50 | 4.63 ± 0.97 |
|  |  | 63.49 ± 0.17 | 51.77 ± 0.22 | 23.13 ± 0.50 | 4.00 ± 0.82 |
|  |  | 64.51 ± 0.06 | 88.21 ± 0.11 | 98.16 ± 0.01 |  |
| DeepLabV3+ | ResNet18 | 62.74 ± 0.12 | 50.90 ± 0.22 | 22.52 ± 0.93 | 5.16 ± 1.01 |
|  |  | 62.74 ± 0.12 | 50.90 ± 0.22 | 22.52 ± 0.93 | 4.67 ± 1.25 |
|  |  | 63.87 ± 0.08 | 87.85 ± 0.11 | 98.12 ± 0.00 |  |
| DeepLabV3+ | EfficientNetB0 | 64.29 ± 0.09 | 53.49 ± 0.10 | 28.82 ± 0.11 | 13.88 ± 0.58 |
|  |  | 64.29 ± 0.09 | 53.49 ± 0.10 | 28.82 ± 0.11 | 13.67 ± 0.47 |
|  |  | 65.14 ± 0.07 | 89.16 ± 0.07 | 98.16 ± 0.00 |  |
| DeepLabV3+ | MobileNetV2 | 62.25 ± 0.24 | 50.47 ± 0.33 | 22.58 ± 0.73 | 5.95 ± 0.78 |
|  |  | 62.25 ± 0.24 | 50.47 ± 0.33 | 22.58 ± 0.73 | 5.33 ± 0.47 |
|  |  | 63.47 ± 0.12 | 87.94 ± 0.27 | 98.08 ± 0.00 |  |
| DeepLabV3+ | MobileViTV2 | 64.65 ± 0.13 | 53.58 ± 0.24 | 27.50 ± 0.83 | 12.66 ± 1.71 |
|  |  | 64.65 ± 0.13 | 53.58 ± 0.24 | 27.50 ± 0.83 | 12.33 ± 1.70 |
|  |  | 65.55 ± 0.05 | 88.81 ± 0.10 | 98.21 ± 0.01 |  |
| UPerNet | ResNet101 | 63.06 ± 0.52 | 51.03 ± 1.05 | 22.61 ± 3.41 | 5.12 ± 3.13 |
|  |  | 63.06 ± 0.52 | 51.03 ± 1.05 | 22.61 ± 3.41 | 4.67 ± 3.30 |
|  |  | 64.55 ± 0.26 | 87.68 ± 0.56 | 98.19 ± 0.01 |  |
| UPerNet | ConvNeXt | 66.71 ± 0.04 | 55.82 ± 0.12 | 29.37 ± 0.82 | 10.13 ± 2.14 |
|  |  | 66.71 ± 0.04 | 55.82 ± 0.12 | 29.37 ± 0.82 | 9.67 ± 1.89 |
|  |  | 67.65 ± 0.05 | 89.88 ± 0.10 | 98.33 ± 0.00 |  |
| UPerNet | MiTB4 | 65.05 ± 0.87 | 53.73 ± 1.21 | 26.65 ± 2.79 | 10.51 ± 5.42 |
|  |  | 65.05 ± 0.87 | 53.73 ± 1.21 | 26.65 ± 2.79 | 10.00 ± 5.66 |
|  |  | 66.37 ± 0.64 | 88.90 ± 0.55 | 98.27 ± 0.03 |  |
| SegFormer | MiTB4 | 64.27 ± 1.15 | 52.90 ± 1.59 | 25.69 ± 2.96 | 8.70 ± 4.76 |
|  |  | 64.27 ± 1.15 | 52.90 ± 1.59 | 25.69 ± 2.96 | 8.33 ± 4.50 |
|  |  | 65.61 ± 0.87 | 88.61 ± 0.72 | 98.23 ± 0.04 |  |
| SegFormer | MiTB2 | 64.74 ± 0.02 | 53.81 ± 0.03 | 28.66 ± 0.22 | 13.74 ± 1.37 |
|  |  | 64.74 ± 0.02 | 53.81 ± 0.03 | 28.66 ± 0.22 | 13.00 ± 1.41 |
|  |  | 65.70 ± 0.09 | 89.44 ± 0.12 | 98.19 ± 0.00 |  |
| DeepLabV3 | ResNet101 | 61.82 ± 2.12 | 49.95 ± 2.66 | 22.49 ± 4.08 | 6.34 ± 2.31 |
|  |  | 61.82 ± 2.12 | 49.95 ± 2.66 | 22.49 ± 4.08 | 6.00 ± 2.16 |
|  |  | 63.57 ± 1.53 | 88.16 ± 0.81 | 98.08 ± 0.09 |  |
| PSPNet | ResNet101 | 61.13 ± 2.60 | 49.09 ± 3.45 | 21.76 ± 5.14 | 7.33 ± 3.03 |
|  |  | 61.13 ± 2.60 | 49.09 ± 3.45 | 21.76 ± 5.14 | 6.67 ± 3.09 |
|  |  | 63.19 ± 1.73 | 87.49 ± 1.25 | 98.09 ± 0.08 |  |
| UNet | ResNet101 | 62.88 ± 0.68 | 50.85 ± 0.94 | 21.23 ± 2.00 | 3.29 ± 0.87 |
|  |  | 62.88 ± 0.68 | 50.85 ± 0.94 | 21.23 ± 2.00 | 3.00 ± 0.82 |
|  |  | 64.26 ± 0.56 | 86.77 ± 0.71 | 98.21 ± 0.00 |  |

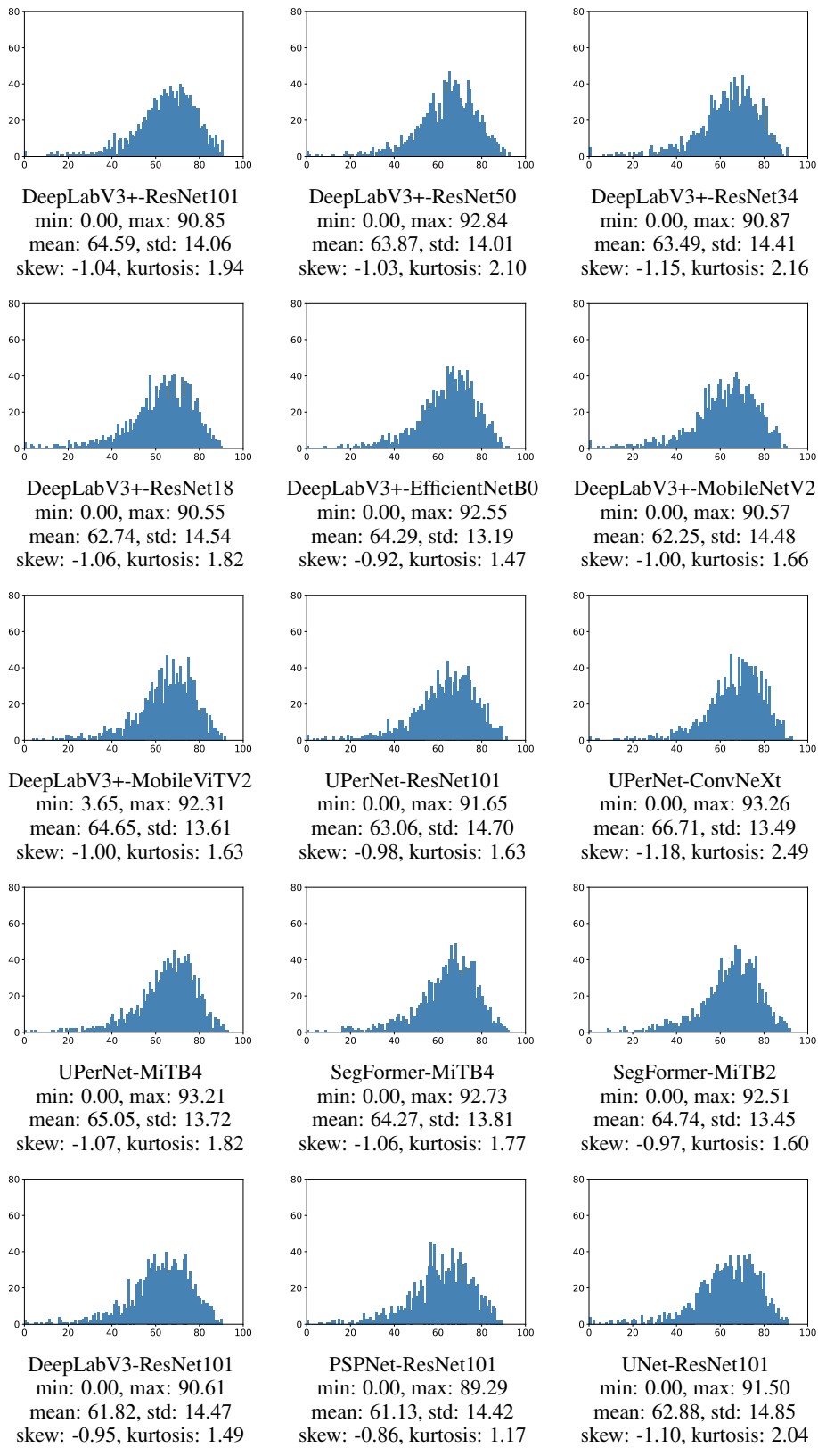

Figure 46: Histograms: DeepGlobe Road.

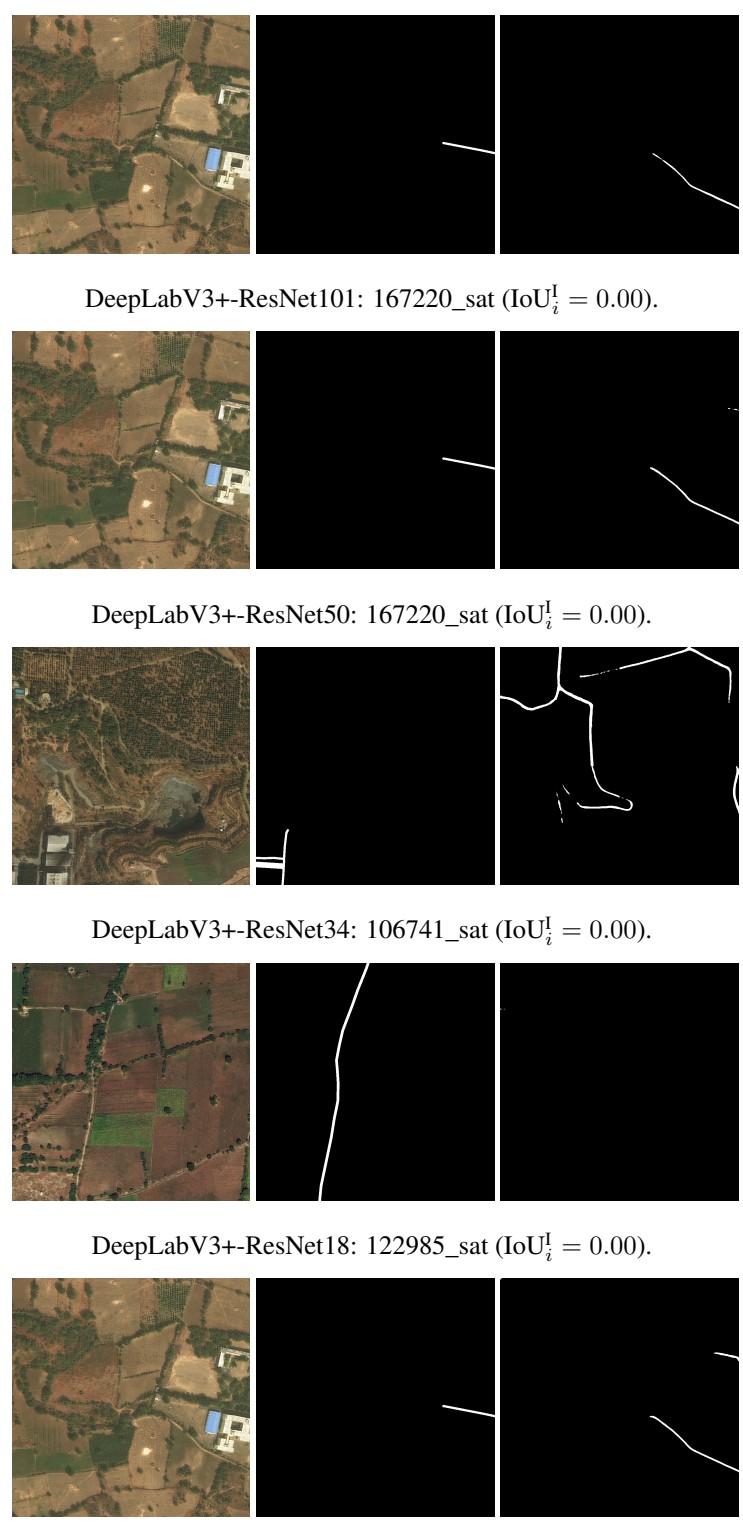

DeepLabV3+-ResNet101: 167220_sat ($IoU_i^I = 0.00$).

DeepLabV3+-ResNet50: 167220_sat ($IoU_i^I = 0.00$).

DeepLabV3+-ResNet34: 106741_sat ($IoU_i^I = 0.00$).

DeepLabV3+-ResNet18: 122985_sat ($IoU_i^I = 0.00$).

DeepLabV3+-EfficientNetB0: 167220_sat ($IoU_i^I = 0.00$).

Figure 47: Worst-case images: DeepGlobe Road 1 / 3. Left: image. Middle: label. Right: prediction.

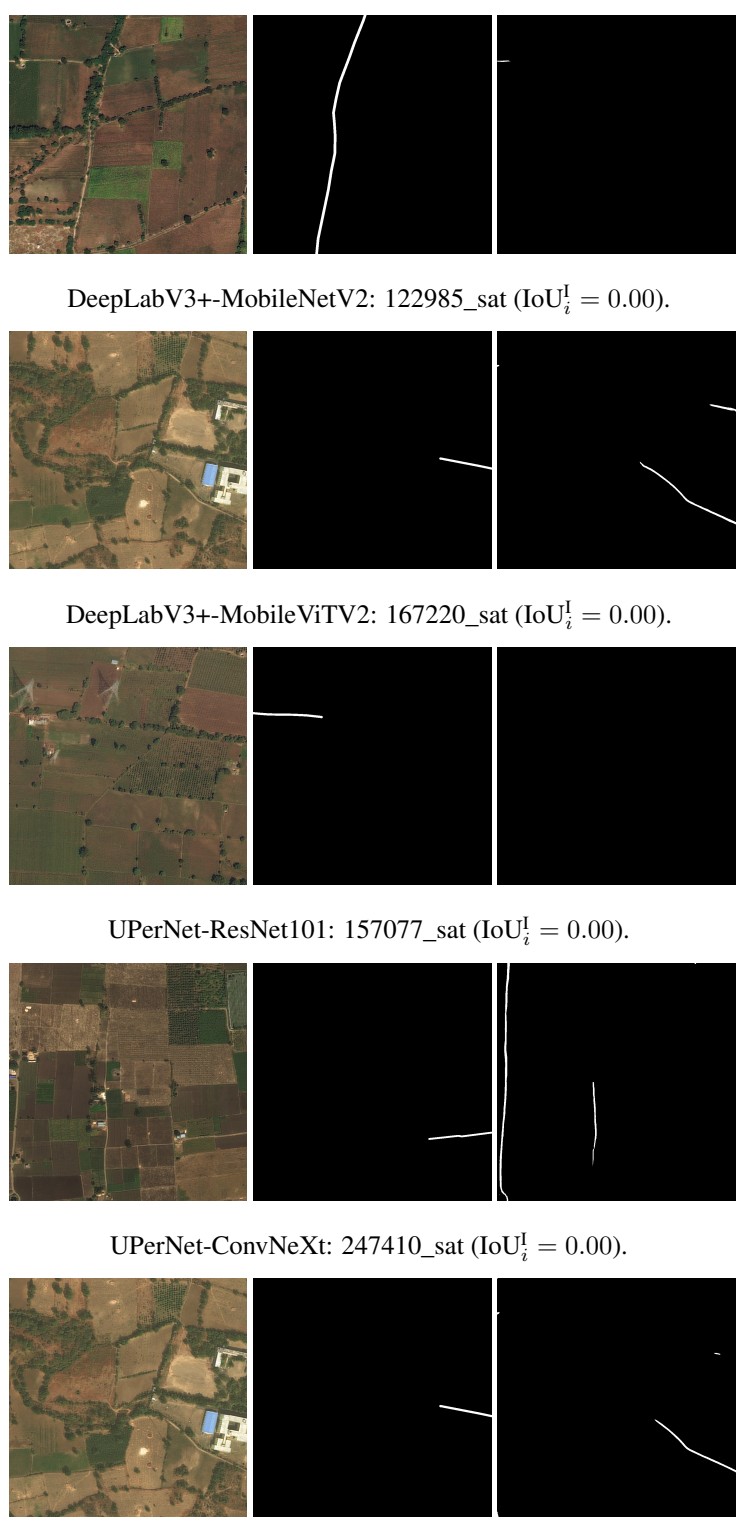

DeepLabV3+-MobileNetV2: 122985_sat (IoU$_i^I$ = 0.00).

DeepLabV3+-MobileViTV2: 167220_sat (IoU$_i^I$ = 0.00).

UPerNet-ResNet101: 157077_sat (IoU$_i^I$ = 0.00).

UPerNet-ConvNeXt: 247410_sat (IoU$_i^I$ = 0.00).

UPerNet-MiTB4: 167220_sat (IoU$_i^I$ = 0.00).

Figure 48: Worst-case images: DeepGlobe Road 2 / 3. Left: image. Middle: label. Right: prediction.

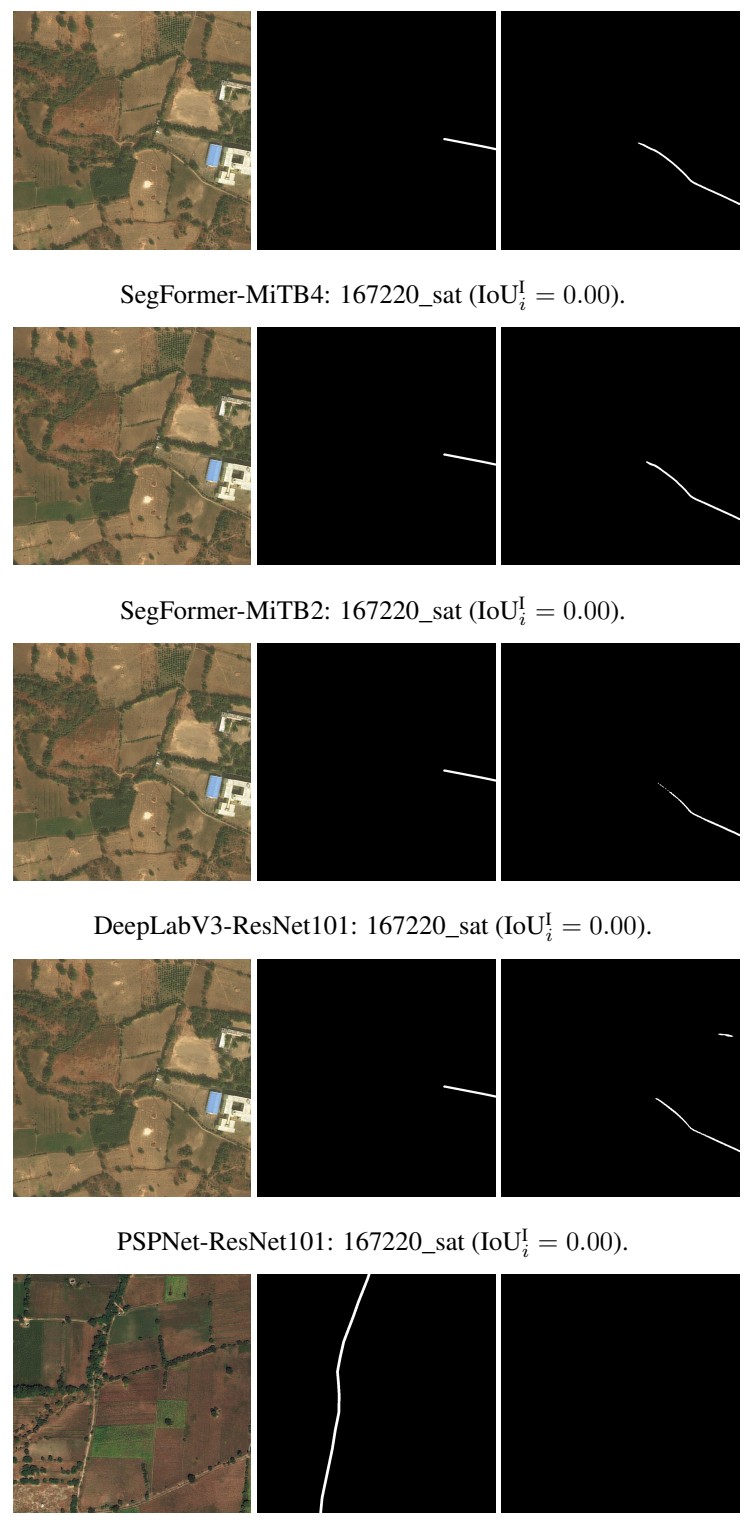

SegFormer-MiTB4: 167220_sat ($IoU_i^I = 0.00$).

SegFormer-MiTB2: 167220_sat ($IoU_i^I = 0.00$).

DeepLabV3-ResNet101: 167220_sat ($IoU_i^I = 0.00$).

PSPNet-ResNet101: 167220_sat ($IoU_i^I = 0.00$).

UNet-ResNet101: 122985_sat ($IoU_i^I = 0.00$).

Figure 49: Worst-case images: DeepGlobe Road 3 / 3. Left: image. Middle: label. Right: prediction.

## G.12 DeepGlobe Building

Table 25: Results: DeepGlobe Building. Red: the best for each metric. Green: the worst for each metric.

| Method | Backbone | mIoU$^I$ / mIoU$^C$ / mIoU$^D$ | mIoU$^{I\bar{q}}$ / mIoU$^{C\bar{q}}$ / mAcc | mIoU$^{I5}$ / mIoU$^{C5}$ / Acc | mIoU$^{I1}$ / mIoU$^{C1}$ |
|---|---|---|---|---|---|
| DeepLabV3+ | ResNet101 | $77.54 \pm 0.18$ 
 $77.54 \pm 0.18$ 
 $80.94 \pm 0.03$ | $59.40 \pm 0.33$ 
 $59.40 \pm 0.33$ 
 $93.66 \pm 0.11$ | $0.12 \pm 0.11$ 
 $0.12 \pm 0.11$ 
 $96.94 \pm 0.01$ | $0.00 \pm 0.00$ 
 $0.00 \pm 0.00$ |
| DeepLabV3+ | ResNet50 | $77.24 \pm 0.19$ 
 $77.24 \pm 0.19$ 
 $80.58 \pm 0.06$ | $59.04 \pm 0.38$ 
 $59.04 \pm 0.38$ 
 $93.44 \pm 0.10$ | $0.15 \pm 0.18$ 
 $0.15 \pm 0.18$ 
 $96.88 \pm 0.02$ | $0.00 \pm 0.00$ 
 $0.00 \pm 0.00$ |
| DeepLabV3+ | ResNet34 | $75.73 \pm 0.30$ 
 $75.73 \pm 0.30$ 
 $79.70 \pm 0.17$ | $55.86 \pm 0.58$ 
 $55.86 \pm 0.58$ 
 $93.29 \pm 0.11$ | $0.00 \pm 0.00$ 
 $0.00 \pm 0.00$ 
 $96.71 \pm 0.04$ | $0.00 \pm 0.00$ 
 $0.00 \pm 0.00$ |
| DeepLabV3+ | ResNet18 | $75.37 \pm 0.45$ 
 $75.37 \pm 0.45$ 
 $79.25 \pm 0.15$ | $55.56 \pm 0.90$ 
 $55.56 \pm 0.90$ 
 $92.98 \pm 0.08$ | $0.00 \pm 0.00$ 
 $0.00 \pm 0.00$ 
 $96.64 \pm 0.03$ | $0.00 \pm 0.00$ 
 $0.00 \pm 0.00$ |
| DeepLabV3+ | EfficientNetB0 | $74.12 \pm 1.04$ 
 $74.12 \pm 1.04$ 
 $80.01 \pm 0.22$ | $52.91 \pm 1.94$ 
 $52.91 \pm 1.94$ 
 $93.61 \pm 0.05$ | $0.00 \pm 0.00$ 
 $0.00 \pm 0.00$ 
 $96.75 \pm 0.04$ | $0.00 \pm 0.00$ 
 $0.00 \pm 0.00$ |
| DeepLabV3+ | MobileNetV2 | $74.66 \pm 1.09$ 
 $74.66 \pm 1.09$ 
 $79.20 \pm 0.04$ | $54.34 \pm 2.08$ 
 $54.34 \pm 2.08$ 
 $93.26 \pm 0.10$ | $0.00 \pm 0.00$ 
 $0.00 \pm 0.00$ 
 $96.61 \pm 0.01$ | $0.00 \pm 0.00$ 
 $0.00 \pm 0.00$ |
| DeepLabV3+ | MobileViTV2 | $74.19 \pm 0.62$ 
 $74.19 \pm 0.62$ 
 $80.58 \pm 0.10$ | $52.89 \pm 1.21$ 
 $52.89 \pm 1.21$ 
 $93.73 \pm 0.03$ | $0.00 \pm 0.00$ 
 $0.00 \pm 0.00$ 
 $96.86 \pm 0.02$ | $0.00 \pm 0.00$ 
 $0.00 \pm 0.00$ |
| UPerNet | ResNet101 | $77.45 \pm 0.14$ 
 $77.45 \pm 0.14$ 
 $80.85 \pm 0.14$ | $59.39 \pm 0.34$ 
 $59.39 \pm 0.34$ 
 $93.46 \pm 0.07$ | $0.15 \pm 0.13$ 
 $0.15 \pm 0.13$ 
 $96.94 \pm 0.02$ | $0.00 \pm 0.00$ 
 $0.00 \pm 0.00$ |
| UPerNet | ConvNeXt | $79.21 \pm 0.22$ 
 $79.21 \pm 0.22$ 
 $81.68 \pm 0.07$ | $62.63 \pm 0.54$ 
 $62.63 \pm 0.54$ 
 $94.04 \pm 0.00$ | $2.83 \pm 0.96$ 
 $2.83 \pm 0.96$ 
 $97.06 \pm 0.02$ | $0.00 \pm 0.00$ 
 $0.00 \pm 0.00$ |
| UPerNet | MiTB4 | $76.73 \pm 1.12$ 
 $76.73 \pm 1.12$ 
 $81.97 \pm 0.16$ | $57.28 \pm 2.40$ 
 $57.28 \pm 2.40$ 
 $94.15 \pm 0.15$ | $0.00 \pm 0.00$ 
 $0.00 \pm 0.00$ 
 $97.11 \pm 0.02$ | $0.00 \pm 0.00$ 
 $0.00 \pm 0.00$ |
| SegFormer | MiTB4 | $77.02 \pm 0.92$ 
 $77.02 \pm 0.92$ 
 $81.60 \pm 0.24$ | $58.21 \pm 2.06$ 
 $58.21 \pm 2.06$ 
 $93.98 \pm 0.18$ | $0.27 \pm 0.39$ 
 $0.27 \pm 0.39$ 
 $97.05 \pm 0.03$ | $0.00 \pm 0.00$ 
 $0.00 \pm 0.00$ |
| SegFormer | MiTB2 | $75.77 \pm 0.03$ 
 $75.77 \pm 0.03$ 
 $81.26 \pm 0.02$ | $55.67 \pm 0.04$ 
 $55.67 \pm 0.04$ 
 $93.91 \pm 0.06$ | $0.00 \pm 0.00$ 
 $0.00 \pm 0.00$ 
 $96.99 \pm 0.01$ | $0.00 \pm 0.00$ 
 $0.00 \pm 0.00$ |
| DeepLabV3 | ResNet101 | $77.21 \pm 0.09$ 
 $77.21 \pm 0.09$ 
 $79.89 \pm 0.36$ | $59.62 \pm 0.56$ 
 $59.62 \pm 0.56$ 
 $93.36 \pm 0.28$ | $1.51 \pm 1.64$ 
 $1.51 \pm 1.64$ 
 $96.74 \pm 0.07$ | $0.00 \pm 0.00$ 
 $0.00 \pm 0.00$ |
| PSPNet | ResNet101 | $77.13 \pm 0.30$ 
 $77.13 \pm 0.30$ 
 $80.32 \pm 0.31$ | $59.09 \pm 0.55$ 
 $59.09 \pm 0.55$ 
 $93.48 \pm 0.33$ | $0.37 \pm 0.28$ 
 $0.37 \pm 0.28$ 
 $96.83 \pm 0.04$ | $0.00 \pm 0.00$ 
 $0.00 \pm 0.00$ |
| UNet | ResNet101 | $76.93 \pm 0.31$ 
 $76.93 \pm 0.31$ 
 $80.87 \pm 0.12$ | $58.11 \pm 0.67$ 
 $58.11 \pm 0.67$ 
 $93.77 \pm 0.13$ | $0.00 \pm 0.00$ 
 $0.00 \pm 0.00$ 
 $96.92 \pm 0.01$ | $0.00 \pm 0.00$ 
 $0.00 \pm 0.00$ |

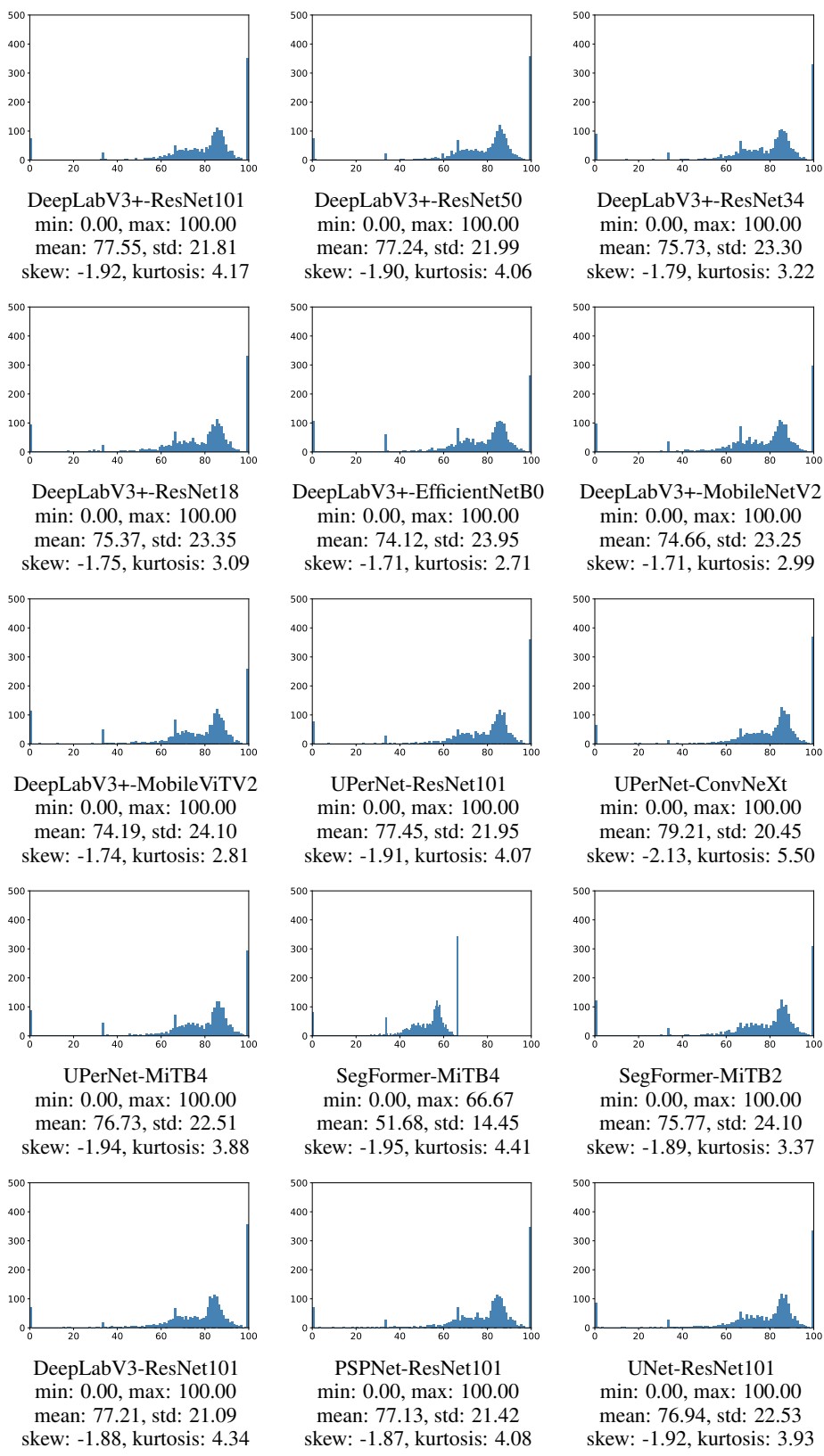

Figure 50: Histograms: DeepGlobe Building.

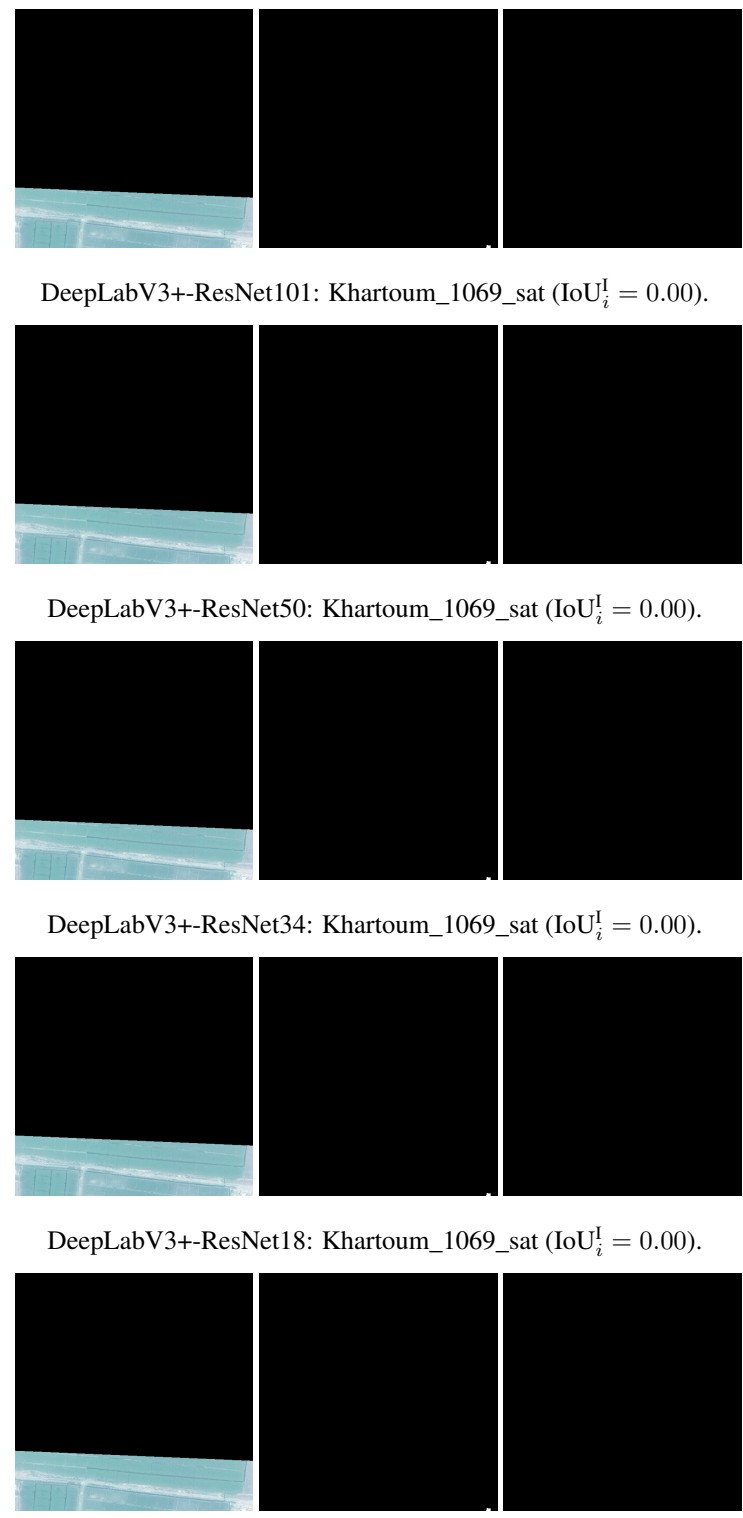

DeepLabV3+-ResNet101: Khartoum_1069_sat ($\text{IoU}_i^{\text{I}} = 0.00$).

DeepLabV3+-ResNet50: Khartoum_1069_sat ($\text{IoU}_i^{\text{I}} = 0.00$).

DeepLabV3+-ResNet34: Khartoum_1069_sat ($\text{IoU}_i^{\text{I}} = 0.00$).

DeepLabV3+-ResNet18: Khartoum_1069_sat ($\text{IoU}_i^{\text{I}} = 0.00$).

DeepLabV3+-EfficientNetB0: Khartoum_1069_sat ($\text{IoU}_i^{\text{I}} = 0.00$).

Figure 51: Worst-case images: DeepGlobe Building 1 / 3. Left: image. Middle: label. Right: prediction.

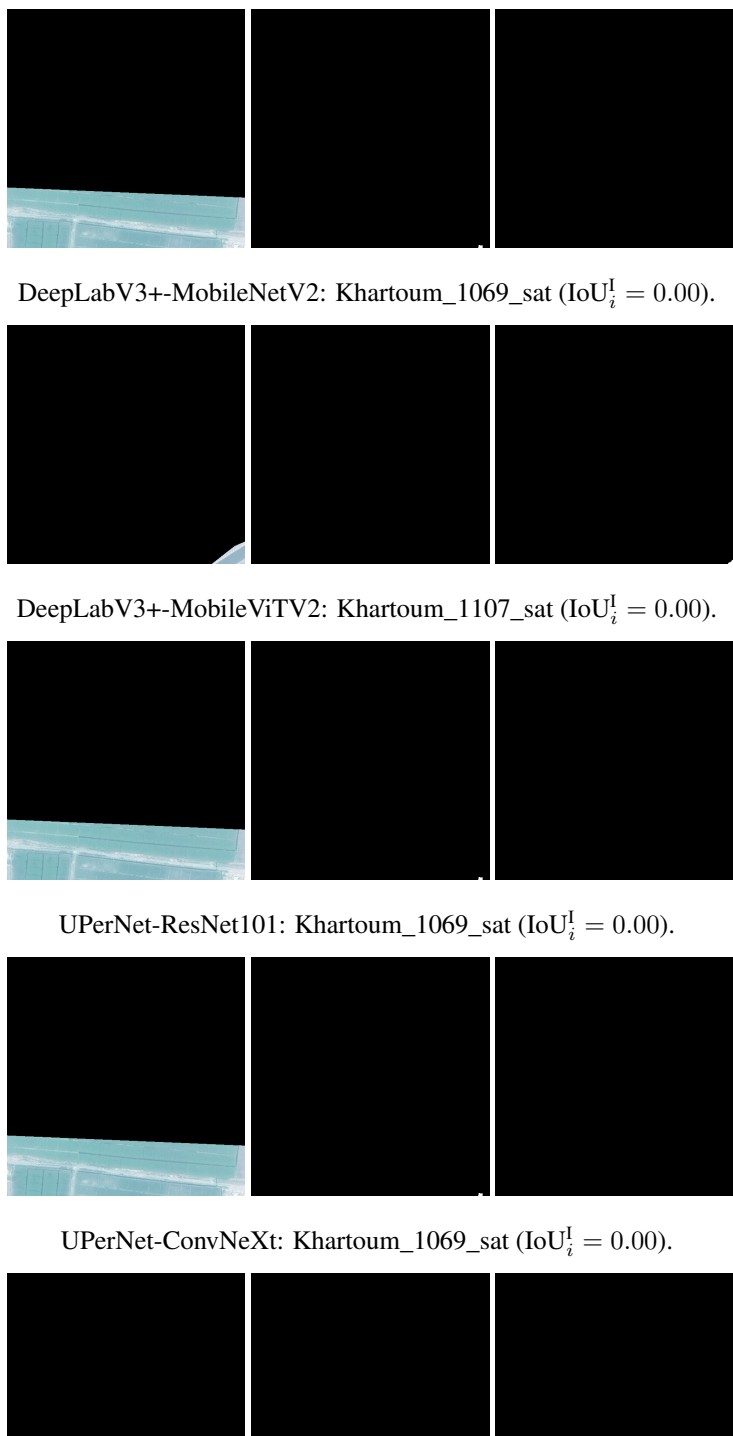

DeepLabV3+-MobileNetV2: Khartoum_1069_sat ($\text{IoU}_i^{\text{I}} = 0.00$).

DeepLabV3+-MobileViTV2: Khartoum_1107_sat ($\text{IoU}_i^{\text{I}} = 0.00$).

UPerNet-ResNet101: Khartoum_1069_sat ($\text{IoU}_i^{\text{I}} = 0.00$).

UPerNet-ConvNeXt: Khartoum_1069_sat ($\text{IoU}_i^{\text{I}} = 0.00$).

UPerNet-MiTB4: Khartoum_1107_sat ($\text{IoU}_i^{\text{I}} = 0.00$).

Figure 52: Worst-case images: DeepGlobe Building 2 / 3. Left: image. Middle: label. Right: prediction.

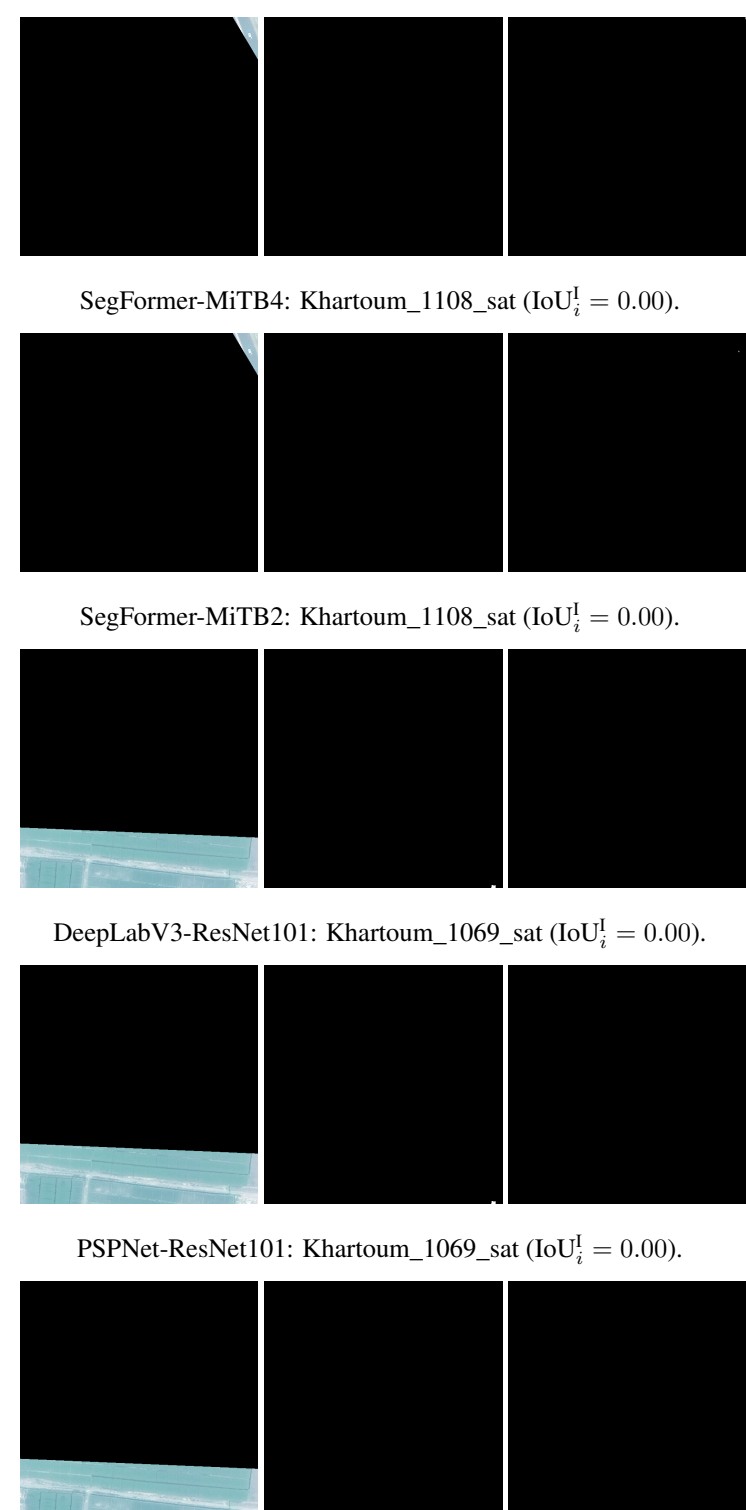

SegFormer-MiTB4: Khartoum_1108_sat (IoU$_i^I$ = 0.00).

SegFormer-MiTB2: Khartoum_1108_sat (IoU$_i^I$ = 0.00).

DeepLabV3-ResNet101: Khartoum_1069_sat (IoU$_i^I$ = 0.00).

PSPNet-ResNet101: Khartoum_1069_sat (IoU$_i^I$ = 0.00).

UNet-ResNet101: Khartoum_1069_sat (IoU$_i^I$ = 0.00).

Figure 53: Worst-case images: DeepGlobe Building 3 / 3. Left: image. Middle: label. Right: prediction.

