# OpenReview forum: "Revisiting Evaluation Metrics for Semantic Segmentation: Optimization and Evaluation of Fine-grained Intersection over Union"
_NeurIPS.cc/2023/Track/Datasets_and_Benchmarks — NeurIPS 2023 Datasets and Benchmarks Poster_

### Official Review · Reviewer_QtsN · 2023-07-20
**A paper about modified mIoU metrics for Semantic Segmentation**

**Rating:** 5
**Confidence:** 5
**Clarity:** Paper is well-written

**Strengths:**

Authors accomplished a comprehensive study for multiple segmentation models on many datasets using the proposed metrics.

**Additional Feedback:**

This paper require a substantial revision. If authors are keen to submit a "benchmark" paper, they need to do it accordingly: propose substantially novel metrics with a certain motivation, provide reproducible code and remove inconsistencies in the story line.

**Correctness:**

Evaluations are performed correctly (however, no code is available to reproduce results at this time).

**Documentation:**

No documentation is available at this time.

**Limitations:**

1. The GitHuB page seems to refer to a completely different project (Jaccard Metric Losses: Optimizing the Jaccard Index with Soft Labels).
2. Paper has minor contributions to research community due to trivial extensions to the conventional mIoU metric.

**Opportunities For Improvement:**

1. The proposed metrics are a trivial modifications of conventional mIoU. Hence, if position paper as the one with novel metrics, more influential ideas are required to convince wider audience to use them.
2. It is not clear what was an actual reason to train models from the scratch. It seems that this paper was initially positioned for an "algorithm" track and, then, converted into a "benchmarking" track with certain inconsistent story line such as unnecessary mass training from the scratch.

**Relation To Prior Work:**

Yes, motivation w.r.t. prior metrics is clear

**Summary And Contributions:**

This paper proposes certain modifications to a commonly-used mIoU metric for semantic segmentation application. Such modifications aim to highlight and overcome a fine-grained shortcomings (e.g. reduce bias towards large objects) of the conventional mIoU metric. Next, authors train multiple models on a number of datasets from the scratch and present results' analysis. In summary, this paper does not propose any new dataset or benchmark. Instead, it proposes minor modifications to mIoU metric with comprehensive evaluations of the trained segmentation models.

---

> ### Author Response · Authors · 2023-08-17
>
> Thank you for acknowledging our comprehensive benchmark study. We appreciate your feedback, and would like to address your concerns regarding relevancy, novelty and reproducibility of our study.
>
> * **This paper does not propose any new dataset or benchmark.** As stated in the abstract (lines 10-13, original version): *"we undertake an extensive benchmark study, where we train and evaluate 15 modern neural network architectures with the proposed metrics on 12 diverse natural and aerial segmentation datasets"*. Therefore, we kindly reject the criticism by the reviewer.
>
> * **This paper has inconsistencies in the story lines as a benchmark paper.** Our paper clearly falls into the scope of NeurIPS Datasets and Benchmarks Track, as specified at *https://neurips.cc/Conferences/2023/CallForDatasetsBenchmarks*:
>     * *"Benchmarks on new or existing datasets, as well as benchmarking tools."* We undertake an extensive benchmark study, where we train and evaluate 15 modern neural networks with the proposed metrics on 12 diverse natural and aerial segmentation datasets. A comprehensive codebase with detailed documentation is also provided along with the submission.
>     * *"Systematic analyses of existing systems on novel datasets yielding important new insight".* We analyze the limitations of conventional mIoU and advocate fined-grained mIoUs due to their advantages in less bias towards large objects and a wealth of statistical information. Our benchmark study over the proposed metrics also yields important new insights regarding best practices to optimize the proposed metrics, e.g. architecture designs and loss functions.
>     * *"Audits of existing datasets, identifying significant problems with existing datasets and their use".* Evaluating with fine-grained mIoUs facilitates data auditing. We identify several mislabels in existing segmentation datasets (lines 154-156, 667-677, original version).
>
> * **The proposed metrics are a trivial modification of conventional mIoU, and more influential ideas are required to convince wider audience to use them.** We firmly believe that the potential research impact of an idea should not be gauged solely by its mathematical complexity. Rather, it is the ability of the idea to address a critical issue in the research community that truly matters. This is particularly the case in a venue like the NeurIPS Datasets and Benchmarks Track, which does not necessarily emphasize algorithmic advances but encourages papers that bring valuable insights to existing systems.
>
>     Indeed, numerous impactful works have emerged from seemingly trivial modifications to prior work. For example, ResNets [1] is a straightforward modification of the Highway Networks [2]; the focal loss [3] is a trivial extension to the cross-entropy loss; layer normalization [4], instance normalization [5] and group normalization [6] are all simple variants of batch normalization [7].
>
>     In the spirit of these examples, the rationale behind our fine-grained metrics is not based on mathematical complexity, but on necessity and relevancy. We identify a critical issue of conventional mIoU -- its bias towards large objects, with an emphasis on safety-critical applications. We propose fine-grained mIoUs due to their advantages over traditional mIoU (lines 134-156, original version), particularly mitigating the bias towards large objects. Other reviewers also recognize the potential research impact of our paper. Notably, we are opening a new research direction by designing novel methods tailored to optimize our proposed metrics.
>
> * **It is not clear what was an actual reason to train models from the scratch.** We have elaborated on the reasons for training models from scratch (lines 167-188, original version), where we discuss aspects such as analysis, completeness, fairness, performance, and statistical significance. Most importantly, we believe that to maintain a fair comparison, it is vital to train different models from scratch under the same training settings, so that we can analyze how specific training strategies, i.e. best practices, can lead to high results on fine-grained metrics.
>
> * **GitHub link is invalid and code is not provided.** As stated in the abstract (lines 17-18, original version), *"the code is available in the appendix and will be made public by the camera-ready deadlines at here"*. Specifically, the GitHub link is still a work in progress, but we ensure that it will be fully functional and updated with the relevant code by the camera-ready deadline. Given the large-scale nature of this paper, we believe the provided comprehensive codebase will be equally valuable to the segmentation community. In the meantime, you can find the source code, as well as detailed documentation, in the supplementary.
>
> We hope our responses can solve your concerns. Please let us know if you have any follow-up comments.

---

> > ### Author Response · Authors · 2023-08-17
> >
> > [1] Kaiming He, Xiangyu Zhang, Shaoqing Ren, and Jian Sun. Deep residual learning for image recognition. CVPR, 2016.
> >
> > [2] Srivastava, Rupesh Kumar, Klaus Greff, and Jürgen Schmidhuber. Highway networks. ICML Workshop, 2015.
> >
> > [3] Tsung-Yi Lin, Priya Goyal, Ross Girshick, Kaiming He, and Piotr Dollár. Focal Loss for Dense Object Detection. TPAMI, 2018.
> >
> > [4] Ba, Jimmy Lei, Jamie Ryan Kiros, and Geoffrey E. Hinton. Layer normalization. NeurIPS Deep Learning Symposium, 2016.
> >
> > [5] Ulyanov, Dmitry, Andrea Vedaldi, and Victor Lempitsky. Improved Texture Networks: Maximizing Quality and Diversity in Feed-forward Stylization and Texture Synthesis. CVPR, 2017.
> >
> > [6] Wu, Yuxin, and Kaiming He. Group normalization. ECCV, 2018.
> >
> > [7] Ioffe, Sergey, and Christian Szegedy. Batch normalization: Accelerating deep network training by reducing internal covariate shift. ICML, 2015.

---

> > ### Comment · Reviewer_QtsN · 2023-08-21
> > **Post-rebuttal comments**
> >
> > Thank you for addressing several of my concerns and overlooked items. Overall, I think, this is a well-written and executed paper. My only left concern is the level of scientific contribution to the community. It seems that the proposed metrics can be potentially valuable to the community (however, not sure if these metrics will be actually widely adopted), but they are inherently limited due reliance on the prior IoU paradigm. Hence, I updated my score to "marginally below acceptance threshold".

---

> > > ### Author Response · Authors · 2023-08-25
> > >
> > > Thank you for your feedback. We appreciate your revised score and the acknowledgment of potential values of our proposed metrics.
> > >
> > > We understand your reservations regarding whether our metrics will gain widespread acceptance, especially given their inherent limitations due to reliance on the prior IoU paradigm. However, we see this reliance as a strength rather than a limitation of our work. We believe science is often progressed step by step, and introducing an entirely new metric (while each metric could have pros and cons [1]) might pose challenges for widespread acceptance within the community. Our metrics, albeit similar to mIoUD, offer clear advantages over it. We hope that this familiarity, combined with the evident benefits, will make our metrics more palatable to the community.
> > >
> > > Indeed, due to the notable shortcoming of mIoUD, there is a recent trend of adopting per-image mIoU in the referring segmentation community [2, 3]. In particular, the influential paper referenced in [3], which was on arXiv only 3 weeks ago, has already gained over 800 GitHub stars. We believe the segmentation community is likely to embrace fine-grained mIoU soon, and our paper is one of the pioneering works that systematically study these metrics.
> > >
> > > Furthermore, a pivotal aspect of our work is highlighting the advantages of fine-grained metrics over conventional dataset-level ones. This focus is not only timely but also adaptable to forthcoming metric innovations. Given the cost and ambiguities associated with pixel-wise labeling [4, 5], there is a clear movement towards unsupervised training in the segmentation domain [6, 7]. In light of this, we anticipate a rising trend where segmentation models will be evaluated in zero-shot contexts. Such evaluations may harness the capabilities of large vision language models, mirroring developments in NLP [8] and image generation [9]. Within this framework, fine-grained metrics, especially the per-image ones, emerge as the most intuitive option.
> > >
> > > We hope our responses can solve your concerns. Please let us know if you have any follow-up comments.
> > >
> > > [1] Lena Maier-Hein et al. Metrics Reloaded: Recommendations for image analysis validation. arXiv, 2023.
> > >
> > > [2] Liu, Chang, Henghui Ding, and Xudong Jiang. GRES: Generalized referring expression segmentation. CVPR, 2023.
> > >
> > > [3] Lai, Xin, Zhuotao Tian, Yukang Chen, Yanwei Li, Yuhui Yuan, Shu Liu, and Jiaya Jia. LISA: Reasoning Segmentation via Large Language Model. arXiv, 2023.
> > >
> > > [4] Bolei Zhou, Hang Zhao, Xavier Puig, Sanja Fidler, Adela Barriuso, and Antonio Torralba. Scene Parsing Through ADE20K Dataset. CVPR, 2017.
> > >
> > > [5] Gerhard Neuhold, Tobias Ollmann, Samuel Rota Bulò, and Peter Kontschieder. The Mapillary Vistas Dataset for Semantic Understanding of Street Scenes. ICCV, 2017.
> > >
> > > [6] Kirillov, Alexander, Eric Mintun, Nikhila Ravi, Hanzi Mao, Chloe Rolland, Laura Gustafson, Tete Xiao et al. Segment anything. ICCV, 2023.
> > >
> > > [7] Zou, Xueyan, Jianwei Yang, Hao Zhang, Feng Li, Linjie Li, Jianfeng Gao, and Yong Jae Lee. Segment everything everywhere all at once. arXiv, 2023.
> > >
> > > [8] Zheng, Lianmin, Wei-Lin Chiang, Ying Sheng, Siyuan Zhuang, Zhanghao Wu, Yonghao Zhuang, Zi Lin et al. Judging LLM-as-a-judge with MT-Bench and Chatbot Arena. arXiv, 2023.
> > >
> > > [9] Chen, Yixiong, Liu, Li, and Ding, Chris. X-IQE: eXplainable Image Quality Evaluation for Text-to-Image Generation with Visual Large Language Models. arXiv, 2023.

---

### Official Review · Reviewer_gpCb · 2023-07-21
**New metric for semantic segmentation tasks with comprehensive studies and extensive experiments**

**Rating:** 7
**Confidence:** 4
**Correctness:** Yes
**Clarity:** Yes

**Strengths:**

+ The motivation of the new metric is clear and easy to follow.
+ This paper provides a comprehensive study of the proposed metric, including its advantages, the benchmark study, and the architecture design as well as loss functions.
+ The experiments are extensive, covering 12 varied natural and aerial datasets and featuring 15 modern neural network architectures, which can verify the claims of the proposed metric.
+ The findings in Sec. 5.1 are interesting, and can possibly guide the design of novel algorithms.


**Additional Feedback:**

This paper is well-organized and written, and the motivation is clear. The benchmark study and corresponding analysis are comprehensive, the experiments are extensive to support the claims. As a result, this paper is above the acceptance bar. But it would be great if the limitation can be discussed.

**Documentation:**

This paper is about a new metric, with the code provided. Thus, there is sufficient detail to support reproducibility.

**Limitations:**

No, please provide more discussion on the limitations. I assume that the negative social impact can be minor as the new metric targets at improving safety-critical applications. But again, to ensure the safety, limitations of the metric should be discussed.

**Opportunities For Improvement:**

1. Does the metric have limitations? Please discuss this.
2. Ln. 198-201, and Ln. 208-214 focus on the fine-grained metrics, so it would be better to put them next to each other.


**Relation To Prior Work:**

Yes

**Summary And Contributions:**

This paper proposes a new mIoU metric to address the class and size imbalance in semantic segmentation datasets. The proposed metric is based on fine-grained mIoUs along with corresponding worst-case metrics. The metric is less bias towards large objects and provides richer statistical information. Extensive experiments are conducted to verify the advantages of the proposed metric.

---

> ### Author Response · Authors · 2023-08-17
>
> We are pleased that you find our paper interesting and acknowledge our extensive benchmark study over the proposed metrics. We answer your question in terms of limitations of the metrics as follows.
>
> * **Does the metrics have limitations?** While we touched upon the limitations of our proposed metrics in section 2 (original version), we recognize the importance of a more comprehensive discussion. Thus, we have added a dedicated section to explore these limitations further. Please refer to lines 293-306 (reversed version).
>
> * **Writing.** We are grateful for your detailed feedback regarding the organization of our contents. We have reorganized the bullet points based on your suggestion. Please refer to 211-224 (revised version).
>
> We hope our responses can solve your concerns. Please let us know if you have any follow-up comments.

---

> > ### Comment · Reviewer_gpCb · 2023-08-26
> > **Concerns are addressed**
> >
> > Thanks for the feedback. My concerns are addressed and I would like to keep my initial rating, "7: Good paper, accept".

---

### Official Review · Reviewer_NFd4 · 2023-07-21
**Well proposed metrics that could improve the evaluation of semantic segmentation**

**Rating:** 7
**Confidence:** 4
**Correctness:** Yes, I am not aware of any error or w…

**Strengths:**

The downside of the current practice and the resulting motivation for fine-grained metrics is well explained. Furthermore, the proposed metrics are clearly described and easy to understand. They cover a wide range of potential use cases, like worst case metrics for safety-critical applications. In the future, researchers may not evaluate only the default mIoU but more specific metrics that better represent their goal for a use case or dataset. Lastly, the experiments demonstrate the benefits of the proposed metrics and enable a more detailed comparison of different models and their architectural approaches. The authors use these insights to provide best practices for training segmentation models.

**Additional Feedback:**

This work does not only cover semantic segmentation but also “panoptic segmentation”. You describe the second task but do not name it. I think it would be easier if you would clearly differentiate between these two tasks: mIoUC for semantic segmentation and mIoUK for panoptic segmentation.

You do not test any medical dataset although semantic segmentation is very relevant in this domain. This would broaden your benchmark which is currently focused on urban driving scenes.
You could compare the mIoUI metric with frequency-weighted IoU rather than the mIoUD because of the class imbalance. Further, your approach for binary segmentation tasks could be compared to IoUpos (dataset wide IoU of the positive class) which is sometimes used for medical datasets.

I currently do not see the benefit of the histogram plots. Did you consider other visualization methods which might enable a better comparison? E.g., a precision-recall curve-like plot.

Overall, the metrics could benefit future evaluations. Providing best practices on how and when to use which metric would further guide your audience.


**Clarity:**

The paper is well written and easy to understand. The metrics and mathematical equations are well described. The overall results in supplementary material are rather difficult to interpret due to the amount of metrics.

**Documentation:**

The code of the experiments is provided in the supplementary material and is going to be released. The code is understandable and sufficiently described to support reproducibility.

**Ethics:**

No ethical concerns.

**Limitations:**

Potential limitations are sparsely discussed (e.g., mIoUI bias towards more frequent classes). However, the proposed metrics have only few relevant limitations that I am aware of but solve the limitations of the current practice. The discussion could include the effect on the computation costs (evaluation time) and the design choice of not considering the non-gt classes.

**Opportunities For Improvement:**

The metrics exclude predicted classes that do not occur in the ground truth mask (differently to related research, see line 79-85). The paper only briefly explains the motivation behind this approach and does not state the effect on the resulting metrics. False positives of non-gt classes are not considered which can lead to an overestimation of the performance. This is especially relevant for datasets with many classes and safety-critical applications as the training may lead to more false positives. The paper would benefit from a more detailed discussion of this decision, potentially including a quantitative analysis.

The benchmark results include 12 newly proposed metrics, additionally to three commonly used ones. Even though each metric may lead to new/other insights, the amount of data is not easy to interpret. The work would benefit from a reduction to the most relevant metrics. Imho, the bias of mIoUI towards more common classes is a big drawback leading to mIoUC as the better option (and mIoUK for panoptic tasks). Similarly, the authors could propose a single metric for the worst-case scenario.

The experiments in the main paper cover only detailed analyses and no overall results. As a reader, I would be interested in how the authors propose to visualize their metrics.


**Relation To Prior Work:**

The history of segmentation metrics is well discussed. The differences to a similar paper with fine-grained metrics are clearly stated. However, the reasons for differing from this paper could be discussed in more detail.

**Summary And Contributions:**

Semantic segmentation models are currently evaluated using the mIoU as the default metric, calculated on a dataset level. However, this metric has downsides like the bias towards large segments. Therefore, this paper develops new metrics. Specifically, they calculate the IoU for each image and class and propose different aggregation methods to calculate multiple fine-grained mIoU metrics (class-wise aggregation, image-wise aggregation, instance-wise aggregation, and worst-case metrics). Further, the work includes a benchmark of multiple models and datasets that highlights the benefits of fine-grained metrics. The proposed metrics enable a more detailed and robust evaluation which clearly benefits the research community.

---

> ### Author Response · Authors · 2023-08-17
>
> Thank you for acknowledging the motivation and potential research impact of our proposed metrics. We appreciate your feedback and address your comments as follows.
>
> * **Excluding predicted classes that do not occur in the ground truth mask.** Our decision to exclude classes in the prediction that do not appear in the ground truth is fundamentally rooted in preventing the metric from over-reacting to minimal mis-predictions, specifically when just a handful of pixels are inaccurately identified as classes not present in the ground truth. It is pivotal to note that these mis-predicted pixels still incur penalties—they are classified as false-negatives relative to their corresponding ground-truth classes. We have added more discussions, as well as quantitative results, in lines 692-703 (revised version).
>
> * **mIoUI.** We agree that the bias of mIoUI towards more frequent classes can be perceived as a significant drawback. However, there are two important aspects to consider:
>     * The bias in mIoUI is towards classes that are more prevalent in terms of image count, which is a milder drawback compared to the bias in Acc, which skews towards classes occupying more pixels.
>     * mIoUI offers several other advantages, such as enabling histograms, highlighting worst-case images, and facilitating dataset auditing. Given these strengths, we argue that mIoUI is particularly valuable for users to analyze the dataset with image-level information, as well as provide a holistic evaluation of various segmentation methods.
>
> * **The benefit of the histogram plots.** Histograms can capture nuances in performance metrics that plain averages tend to miss. For instance, two segmentation methods might yield the same mean score, but differ significantly in how their scores are distributed across individual images. To illustrate, two Gaussian distributions with identical means but varied variances will appear identical in terms of a single average, but can be clearly differentiated using histograms. While our observations indeed indicate that distinct architectures produce quite similar histograms on the same dataset (lines 237-244, revised version), we firmly believe that providing a comprehensive evaluation, rather than only reporting a few averaged scores, is essential for a more profound understanding of the performance of segmentation methods.
>
> * **A single metric for the worst-case scenario.** We concur with your perspective on having a single metric for worst-case scenarios. While utilizing $q=1$ might be suitable for larger datasets like ADE20K, it could introduce significant variance for smaller datasets such as DeepGlobe Land. For the latter, a value of $q=5$ might be more appropriate. With these considerations, we have opted to provide both metrics for the sake of comprehensive evaluation, allowing dataset organizers the flexibility to choose the metric that best fits their context.
>
> * **Providing when to use which metric.** To clarify the context in which each metric should be applied, we have introduced a dedicated section outlining our recommendations. Please refer to lines 307-321 (revised version).
>
> * **The main paper covers only detailed analyses and no overall results.** We have included more discussions of overall results in terms of metrics ranking and worst-case images. Please refer to lines 191-196, 203-210, 231-236 (revised version).
>
> * **Potential limitations are sparsely discussed.** We acknowledge the importance of addressing potential drawbacks, and have expanded our discussion on limitations. Please refer to lines 293-306 (revised version).
>
> * **You do not test any medical dataset.** We concur that the inclusion of medical datasets could broaden the benchmark. However, the methodologies typically employed in medical datasets, ranging from data preprocessing and architectural designs to training hyper-parameters and evaluation metrics, significantly diverge from our current pipeline. Such differences could complicate the narrative and potentially obscure our primary findings. That said, we recognize the value of a more focused exploration, and we are open to the possibility of a dedicated study, perhaps in the form of a workshop paper, that solely focuses on medical datasets.

---

> > ### Author Response · Authors · 2023-08-17
> >
> > * **mIoUC for semantic segmentation and mIoUK for panoptic segmentation.** We would like to clarify a potential misinterpretation regarding the utilization of mIoUK. While we refer to panoptic segmentation, the intent is not to evaluate it with mIoUK, but to leverage instance-level labels from panoptic segmentation (rather than from instance segmentation) to compute mIoUK for evaluating semantic segmentation models. This distinction (panoptic segmentation vs. instance segmentation) is important because, in our research, we adhere to the conventions of panoptic segmentation where each pixel has only one semantic label, i.e. no overlapping instances. In essence, even with instance-level labels derived from panoptic segmentation, our primary objective remains the evaluation of semantic segmentation models. To avoid confusions, we have modified the description of mIoUK. Please refer to lines 99-110 (revised version).
> >
> > We hope our responses can solve your concerns. Please let us know if you have any follow-up comments.

---

> > ### Comment · Reviewer_NFd4 · 2023-08-22
> >
> > Thank you for taking the time to address my feedback and respond to my questions.
> >
> > I like to clarify two points:
> >
> > Excluding null classes: I appreciate your explanation of your chosen approach. However, I still have some reservations about fully supporting your method over the alternative, particularly in light of the notable disparities you’ve presented between these approaches in the new Table. For instance, security-related use cases caught my attention. Some false positive pixels could potentially trigger false alarms, which might occur more frequently given that these pixels are solely treated as false negatives during training and evaluation. My intention isn’t to imply that the alternative approach is better, but rather to convey that I’m not entirely convinced by the argument your provided.
> >
> > Medical datasets: You rightly acknowledge that medical studies often deviate, and you don’t have to extra mention the medical datasets in the paper. I just wanted to highlight that most works focus on these typical categories: common, driving, and aerial. Instead of evaluating the 5th driving dataset, I encourage you to explore a more diverse set of datasets in your future research from domains such as medicine, biology, engineering, or others.
> >
> > Once again, I appreciate your engagement with my feedback.

---

> > > ### Author Response · Authors · 2023-08-25
> > >
> > > Thank you for your in-depth analysis and further insights into our work. In the following, we present some thoughts regrading the two points.
> > >
> > > **Excluding null classes.** The point you raise about false-positives potentially triggering false alarms is indeed relevant. However, we do not want to discriminate between false-positives solely based on whether the predicted class is present in the ground-truth or not. Generally speaking, mispredicting "road" as either "car" or "truck" should incur a similar penalty. Yet, if "car" is present in the ground-truth and "truck" is not, mispredicting "road" as "car" may only result in a mild penalty, whereas mispredicting "road" as "truck" will incur a full penalty.
> > >
> > > Moreover, in safety-critical settings, false-negatives should have higher importance, e.g. overlooking a car present in the ground truth. Meanwhile, the metrics should not be overly sensitive to false-positives, e.g. a few pixels are mispredicted as a car absent from the ground-truth. Overemphasizing such false-positives might inadvertently diminish the significance of false-negatives. A concerning scenario could be when the ground-truth contains only a small number of classes, yet each unrelated class has a few pixels erroneously present in the prediction.
> > >
> > > Although we present our metrics with the belief that they offer advantages in certain scenarios, we understand that each metric will have its pros and cons. It is essential to note that our aim is not to assert that our metrics are the best or only way, but to highlight this distinction. Ultimately, end users could select the version most aligned with their specific requirements.
> > >
> > > **Medical datasets.** We are in complete agreement with you on the importance of exploring a diverse set of datasets. While our current focus is on the more typical categories, your suggestion to delve into domains like medicine, biology, and engineering in future research is well-taken. Such diversity will certainly enhance the breadth and depth of our work.
> > >
> > > Thank you once again for your thoughtful considerations. Please let us know if you have any follow-up comments.

---

> > > > ### Author Response · Authors · 2023-08-26
> > > >
> > > > We've slightly modified the above comment with an additional example of mispredicting "road" as either "car" or "truck". Please see the updated comment.

---

### Official Review · Reviewer_ahvw · 2023-07-21
**Nice addition to evaluation metrics for semantic segmentation**

**Rating:** 7
**Confidence:** 4
**Correctness:** The claim is technically sound.
**Clarity:** The paper is well-written overall.

**Strengths:**

- The paper analyze limitations of existing evaluation protocols thoroughly, and proposes new evaluation metrics that can effectively address the limitations.

- I believe that the proposed fine-grained evaluation metrics are technically sound. This will open a new research direction of semantic segmentation.

- Re-evaluation of existing semantic segmentation methods on various datasets is fruitful for future semantic segmentation research.

**Additional Feedback:**

N/A

**Documentation:**

There is sufficient information for reproducing the metrics.

**Limitations:**

There is no discussion about limitations, but I also do not find any specific limitation about this work.

**Opportunities For Improvement:**

- In terms of presentation, the paper can be more improved. For example, the authors can utilize more visual examples to explain the limitations of previous metrics and strenghts of the proposed protocols. It is also helpful if the authors provide figure of computational pipeline to explain best practices.


**Relation To Prior Work:**

Yes.

**Summary And Contributions:**

The paper considers an evaluation protocol for semantic segmentation. To address the conventional metrics such as mAcc or mIoU, the authors introduce novel fine-grained metrics for evaluating semantic segmentation methods. Specifically, they propose mIoU^I (per-image-per-class), mIoU^C (per-class-per-image), mIoU^K (per-instance) metrics. They also propose the worst-case metrics. With the proposed evaluation protocol, they perform extensive experiments with existing methods and datasets. As a result, there is no method that works well on the proposed fine-grained metrics. The authors also provide guideline for improving performances in terms of fine-grained evaluation metrics.

---

> ### Author Response · Authors · 2023-08-17
>
> Thank you for recognizing the potential impact and fruitful evaluation of our work. We value your feedback and address your suggestions as below.
>
> * **More visual examples to explain the limitations of previous metrics.** To enhance clarity, we have added more discussions, as well as visual examples from Cityscapes and Mapillary Vistas to illustrate the limitation of traditional mIoU. Please refer to lines 33-36 (revised version).
>
> * **Figure of computational pipelines to explain best practices.** While we understand the utility of a visual representation, we believe that our best practices are more about concurrent choices rather than a step-by-step pipelines procedure. To facilitate a clearer presentation, we have added an intro before presenting specific best practices, and we hope it will provide readers with a deeper insight into the best practices. Please refer to lines 250-254 (revised version).
>
> We hope our responses can solve your concerns. Please let us know if you have any follow-up comments.

---

### Author Response · Authors · 2023-08-17

We thank all reviewers for your time and efforts. The questions and suggestions are very valuable for us to improve our work. We have revised the paper based on your comments. Below is a summary of modifications.

* **lines 33-36 (ahvw).** More discussions, as well as visual examples from Cityscapes and Mapillary Vistas to illustrate the limitation of traditional mIoU.

* **lines 99-110 (NFd4).** Revise the description of mIoUK to avoid confusions between semantic segmentation and panoptic segmentation.

* **lines 191-196, 203-210, 231-236 (NFd4).** More discussions of overall results in terms of metrics ranking and worst-case images.

* **lines 251-254 (ahvw).** Introduce best practices for a clearer presentation.

* **lines 293-306 (NFd4, gpCb).** Discuss limitations of this work.

* **lines 307-321 (NFd4).** Discuss suggestions when to use which metric.

* **lines 692-703 (NFd4).** More discussions of null classes in per-image mIoUs.

Please let us know if you have any follow-up comments.

---

### Decision · Program_Chairs · 2023-09-22

**Decision:**

Accept (Poster)

**Comment:**

This work proposes the use of fine-grained mIoUs along with corresponding worst-case metrics, offering a more holistic evaluation of segmentation techniques to address class and size imbalances in the data.

Most reviewers found this benchmark paper interesting and important. They commented that:

+ The paper is well written and easy to understand.
+ The fine-grained evaluation metrics are technically sound, although they are a simple modifications of mIoU.
+ The paper Re-evaluation of existing semantic segmentation methods on various datasets is interesting.

Some improvements were proposed:
+ Potential limitations are not discussed enough.
+ The work would benefit from focusing on to the most relevant metrics.

The low score reviewer did not think the paper is suitable for the Dataset and Benchmark track, but authors seems to convince him. I believe that even if the new metrics are only simple adaptations of the old mIoU, the fact that the paper addresses the class/size imbalances is very important. In fact, using simple mIoU metrics one can play with the results by including/removing classes etc.
This paper highlights this problem and tries to address it.

I therefore suggest accepting this paper if the authors revise according to the reviewers' suggestions.